# General anaesthesia decreases the uniqueness of brain functional connectivity across individuals and species

The human brain is characterized by idiosyncratic patterns of spontaneous thought, rendering each brain uniquely identifiable from its neural activity. However, deep general anaesthesia suppresses subjective experience. Does it also suppress what makes each brain unique? Here we used functional MRI scans acquired under the effects of the general anaesthetics sevoflurane and propofol to determine whether anaesthetic-induced unconsciousness diminishes the uniqueness of the human brain, both with respect to the brains of other individuals and the brains of another species. Using functional connectivity, we report that under anaesthesia individual brains become less self-similar and less distinguishable from each other. Loss of distinctiveness is highly organized: it co-localizes with the archetypal sensory–association axis, correlating with genetic and morphometric markers of phylogenetic differences between humans and other primates. This effect is more evident at greater anaesthetic depths, reproducible across sevoflurane and propofol and reversed upon recovery. Providing convergent evidence, we show that anaesthesia shifts the functional connectivity of the human brain closer to the functional connectivity of the macaque brain in a low-dimensional space. Finally, anaesthesia diminishes the match between spontaneous brain activity and cognitive brain patterns aggregated from the Neurosynth meta-analytic engine. Collectively, the present results reveal that anaesthetized human brains are not only less distinguishable from each other, but also less distinguishable from the brains of other primates, with specifically human-expanded regions being the most affected by anaesthesia.

Consciousness—what is lost during anaesthesia and dreamless sleep and restored upon awakening—is inherently subjective to each individual, as indicated by the near-synonymous use of the terms subjective experience and first-person experience. In other words, each individual's consciousness is unique to them. This raises an intriguing question: if consciousness is what makes each of us unique, do we become more alike when consciousness is lost?

A temporary state of unconsciousness can be induced by anaesthetics. The medical benefits of anaesthesia are well established, but the value of its use as a tool to study the functioning of the brain is also increasing[1–5]. Unlike spontaneous sleep, anaesthetic-induced unconsciousness (indicated by a loss of behavioural responsiveness) is amenable to experimental control: it can be reliably induced, maintained and reversed.

In this Article, we combine loss and recovery of consciousness induced by deep anaesthesia using different anaesthetics—sevoflurane[6,7] and propofol[8,9]—with functional MRI (fMRI) recordings of spontaneous activity in the human brain. We ask: does the human

✉e-mail: al857@cam.ac.uk

brain lose its distinctiveness when unconscious? We attack this question from three conceptual angles. First, we compare brains within and across individuals. Seminal work revealed that the patterns of functional connectivity (FC) between brain regions are reliably different across individuals, enabling brain fingerprinting of individuals based on fMRI scans[10-14]. Therefore, here we use functional connectome fingerprinting to evaluate whether individuals become less distinguishable when under deep anaesthesia, which is presumed to induce unconsciousness (note that this is different from mere sedation, during which participants are still responsive and conscious, albeit sluggish[14]).

Second, we assess how well each individual's brain activity across different levels of anaesthesia corresponds to canonical brain maps of cognitive operations obtained from meta-analytic aggregation of >14,000 neuroimaging experiments[15]. Although our study concerns task-free fMRI, we reasoned that even in the absence of any tasks the brain may still spontaneously engage states pertaining to various cognitive operations[16-22]. In contrast, this should not occur during loss of consciousness, when even intrinsically driven cognition should be abolished. This paradigm is inspired by evidence that the ability to detect brain responses to specific tasks (such as imagine playing tennis or imagine navigating around your house) is a robust marker of consciousness even in individuals who are behaviourally unresponsive due to disorders of consciousness[23-32].

Finally, we ask whether anaesthesia makes the FC of the human brain less distinctive from other species—in other words, reducing the distinctiveness of our species compared with other primates.

## Results

In this Article, we consider resting state fMRI data obtained from $n = 15$ healthy volunteers at baseline and after loss of behavioural responsiveness induced by different levels of the inhalational anaesthetic sevoflurane: at electroencephalogram (EEG) burst suppression and 3 and 2 vol%, as well as during post-anaesthetic recovery of responsiveness[6,33]. We replicate our results in an independent dataset of resting state fMRI results from $n = 16$ healthy volunteers scanned before, during and after loss of behavioural responsiveness induced by the intravenous anaesthetic propofol[8,9].

### Reduced identifiability of the anaesthetized brain
First we tested the hypothesis that anaesthesia abolishes each individual's idiosyncratic patterns of spontaneous neural activity, making the corresponding patterns of FC more difficult to distinguish across individuals. Specifically, we correlated each individual's FC during wakefulness with each individual's FC during either post-anaesthetic recovery of responsiveness or anaesthesia. This produced an identifiability matrix where the rows and columns are individuals and each entry represents their connectomes' similarity (correlation) (Fig. 1a).

Successful brain fingerprinting requires two conditions. The first is persistency: an individual's FC needs to be consistent over time in order to be used to identify the individual. The second is diversity: the FCs of distinct individuals need to be different from each other, to avoid confusing individuals. If FC patterns are all the same, identifiability will be low even though FC persists over time. In contrast, if FC is very variable over time identifiability will be low, even if FC configurations are diverse across individuals. We quantify persistency as self–self correlation across scans (correlation between FC at time 1 and time 2, for the same individual). We quantify (lack of) diversity as the mean self–other correlation: the mean correlation between an individual's FC at time 1 and every other individual's FC at time 2. Finally, differential identifiability is the difference between self–self correlation (persistency) and self–other correlation (lack of diversity).

We observed that when comparing two scans of the same individual while awake, it was easy to distinguish self from other. This identifiability can be discerned as a clear pattern along the matrix diagonal, representing self–self similarity, which is clearly distinguishable from

off-diagonal entries, which represent self–other similarity (Fig. 1a). This was consistent with previous work on functional connectome fingerprinting using test–retest scans[10]. In contrast, the diagonal was barely discernible when awake scans were compared against anaesthetized ones, indicative of low identifiability of individuals (Fig. 1b). Indeed, self–self correlations between awake and anaesthetized brains were significantly diminished compared with awake–recovery self-similarity (Fig. 1c). Likewise, differential identifiability (the difference between self–self correlation and the mean correlation between an individual's scan at time $t_x$ and every other individual's scan at time $t_y$) was significantly reduced when considering anaesthetized brains (Fig. 1d). Collectively, these results demonstrate that anaesthetic-induced loss of responsiveness manifests as decreased distinctiveness of the individual functional connectome. Analogous results were obtained at different depths of sevoflurane anaesthesia (Extended Data Fig. 1). In the remainder of this subsection, we identify the anatomical organization of functional connections that contribute to this change in distinctiveness.

We quantify edgewise identifiability using the intra-class correlation coefficient (ICC), which describes how strongly elements in the same group resemble each other, for a given score. In this context, we obtained an ICC value for each edge (the FC value between two brain regions), which indicates how well the weight of that edge separates within and between individuals[10]. Thus, the higher the ICC of an edge, the higher its identifiability (Methods)[10]. The difference in edgewise identifiability between awake–recovery and awake–anaesthesia therefore indicates the extent to which the identifiability of each functional edge is affected by anaesthesia. In other words, the matrix of ICC differences reflects, for each edge, the gain in identifiability that one obtains with consciousness (that is, how much extra ability to discriminate individuals there is when using the recovery scan instead of the anaesthetized scan).

We found that sevoflurane anaesthesia reduced the contribution to identifiability of virtually all edges (Fig. 2a). This pattern was neither uniform nor random. Rather, the anaesthetic-induced change in identifiability of each functional connection was proportional to its contribution to identifiability during wakefulness (that is, between awake and recovery) (Supplementary Fig. 1). Additionally, the prevalence of FC edges that were capable of reliably identifying individuals (that is, whose confidence interval did not include zero) was drastically diminished under anaesthesia (Supplementary Fig. 2).

Notably, the most affected functional connections are those connecting two regions of transmodal cortex (Fig. 2b; see Supplementary Fig. 3 for alternative comparisons between different edge types). This is noteworthy because regions of transmodal cortex are known to provide the largest contribution to identifiability in the awake resting brain[10,12]. In other words, the more a functional edge contributes to identifiability in awake individuals, the more it is affected by anaesthesia. Functional connections within transmodal cortex are particularly vulnerable to this perturbation, losing their distinctiveness and becoming more similar across individuals.

Next, we localized regional changes in identifiability, quantified as the mean change in edgewise identifiability across each region's edges. The greatest decreases in identifiability occurred in regions of the default mode and fronto-parietal networks, as well as the transmodal cortex more broadly, whereas unimodal (somatomotor and visual) cortices were least affected (Fig. 2c and Supplementary Fig. 4). This regional pattern of unimodal–transmodal distinction was confirmed by a significant spatial correlation with the brain's archetypal sensory–association axis (Spearman's $\rho = 0.67$; $p_{spin} < 0.001$; $n = 200$ regions; Fig. 2d)[34]. This region-wise result was consistent with our observation that the most affected functional connections were those linking transmodal regions (Fig. 2b). Given that the FC of the transmodal cortex exhibits the greatest inter-individual variability in the awake state[35], we sought to test whether the anaesthesia-induced decrease in identifiability preferentially targets these regions. Indeed,

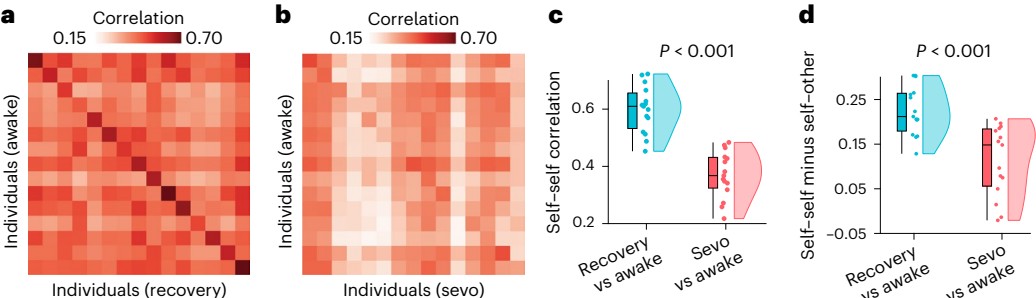

**Fig. 1 | The identifiability of individual functional connectomes is diminished under sevoflurane anaesthesia. a**, Identifiability matrix between wakefulness and post-anaesthetic recovery. The rate of successful identification is 93%. **b**, Identifiability matrix between wakefulness and sevoflurane (sevo) anaesthesia. In **a** and **b**, entries along the diagonal represent self–self similarity (correlation of FC patterns), whereas off-diagonal entries represent self–other similarity. **c**, Self–self similarity is significantly higher between two conscious states than between wakefulness and anaesthesia (for awake versus recovery, mean = 0.60 and s.d. = 0.08; for awake versus anaesthesia, mean = 0.37 and s.d. = 0.08; $t(14)$ = 8.36; $P < 0.001$; effect size (Hedge's $g$) = 2.71; confidence interval (CI) = [2.22, 3.69];

two-sided $t$-test). **d**, Differential identifiability (the difference between self–self correlation and mean self–other correlation for each individual) is significantly higher between two conscious states than between wakefulness and anaesthesia (for awake versus recovery, mean = 0.19, s.d. = 0.06; awake–anaesthesia: mean = 0.07; s.d. = 0.09; $t(14)$ = 4.79; $P < 0.001$; effect size (Hedge's $g$) =1.57; CI = [1.08, 2.35]; two-sided $t$-test). For the box plots in **c** and **d**, the central lines indicate median values, the bounds of the boxes indicate the 25th and 75th percentiles, the whiskers indicate 1.5× the interquartile range and extreme values are shown as individual circles ($n$ = 15 human volunteers). Source data are provided.

patterns of decreased identifiability were correlated with the map of inter-individual variability developed by Mueller and colleagues[35] (Spearman's $\rho$ = 0.63; $p_{spin}$ < 0.001; $n$ = 200 regions; Fig. 2d). The correlations remained significant after regressing out of the regional ICC map, a map of the human brain's regional signal-to-noise ratio of the fMRI signal from ref. 36 (Supplementary Fig. 5). Analogous results were observed at both the edge level and regional level when only including ICC values whose confidence interval did not include zero (Supplementary Fig. 2).

Having observed that changes in regional identifiability are more pronounced where individuals most differ in terms of FC, we further investigated whether a more general phenomenon is at play. Do anaesthetic-induced changes in regional identifiability reflect not only distinctiveness between individuals, but more generally distinctiveness between species? Although we pursue this in more detail in a subsequent section, here we show that changes in regional identifiability correlate with molecular and morphometric markers of phylogenetic cortical differentiation between human and non-human primates. Specifically, anaesthetic-induced changes in regional identifiability are spatially correlated with the cortical map of evolutionary expansion between macaque and human[37] (Spearman's $\rho$ = 0.35; $p_{spin}$ < 0.001; $n$ = 200 regions; Fig. 2d), with greater change in distinctiveness being observed in phylogenetically newer regions. Likewise, we observed a significant spatial correlation between the regional changes in identifiability, and the regional mean expression of genes associated with so-called human accelerated regions (HAR) of the human genome pertaining to brain function and development (HAR–brain genes; Spearman's $\rho$ = 0.42; $p_{spin}$ < 0.001; $n$ = 200 regions; Fig. 2d). These are genes associated with loci that displayed accelerated divergence in the human lineage compared with the chimpanzee, and therefore indicate evolution-related changes in the corresponding regions[38]. In other words, brain regions that exhibit greater change in distinctiveness also exhibit greater expression of human-accelerated genes. Altogether, anaesthesia selectively reduces the identifiability of brain regions that are most distinctive, both between individuals and between humans and non-human primates.

Correlation analysis cannot identify causal determinants of regional changes in identifiability. Nevertheless, multivariate analysis can be helpful in providing insights beyond what is available from multiple individual correlations. Specifically, we can use dominance analysis to assess the relative importance of different canonical brain maps in predicting the regional distribution of identifiability changes[39].

Dominance analysis distributes the fit of the model across predictors such that the contribution of each predictor can be assessed and compared with other predictors, reflecting the proportion of the variance jointly explained by all predictors that can be attributed to each predictor.

Together, the maps of inter-individual variability, archetypal axis, evolutionary expansion and HAR–brain gene expression accounted for 51% of variance in the map of sevoflurane-induced regional changes in identifiability (Supplementary Fig. 6a)—significantly more than was accounted for by null maps with preserved spatial autocorrelation (Supplementary Fig. 6b). Archetypal axis and inter-individual variability were the most important predictors, accounting for 46.5 and 36.0% of the explained variance, respectively, whereas HAR–brain gene expression and evolutionary expansion accounted for 12.4 and 5.0%, respectively (Supplementary Fig. 6a).

### Decreased correspondence with canonical cognitive maps

An influential approach to the investigation of pathological or pharmacological perturbations of consciousness is to determine whether cognitive processes can be algorithmically inferred (or decoded) from neural activity. For example, patients suffering from disorders of consciousness may be asked to imagine playing tennis while in the scanner, to determine whether motion-related regions become reliably activated in response to a command, despite the absence of overt behavioural command following[23,24]. Although this is typically done in the presence of explicit tasks or other stimuli (for example, movie watching[23,24,28]), here we sought to investigate whether covert cognitive processes can be discerned from spontaneous neural activity more generally, through a comprehensive assay based on meta-analytic maps from thousands of neuroimaging experiments[15,40]. Specifically, across different levels of anaesthesia, we assessed how well each individual's neural activity maps at each point in time, corresponded to 123 canonical brain maps obtained from meta-analytic aggregation of >14,000 neuroimaging experiments[15] (Fig. 3a; see Methods for details of how the Neurosynth brain maps were selected and Supplementary Table 1 for the full list of terms included). For simplicity, hereafter we refer to this activity-based reverse inference approach as cognitive matching. Averaging across the scan duration provides, for each individual and each condition, an overall index of the quality of cognitive matching.

We found that as anaesthesia deepens (with an increasing concentration of sevoflurane), the quality of cognitive matching deteriorates: the best spatial correlation between brain activity and meta-analytic

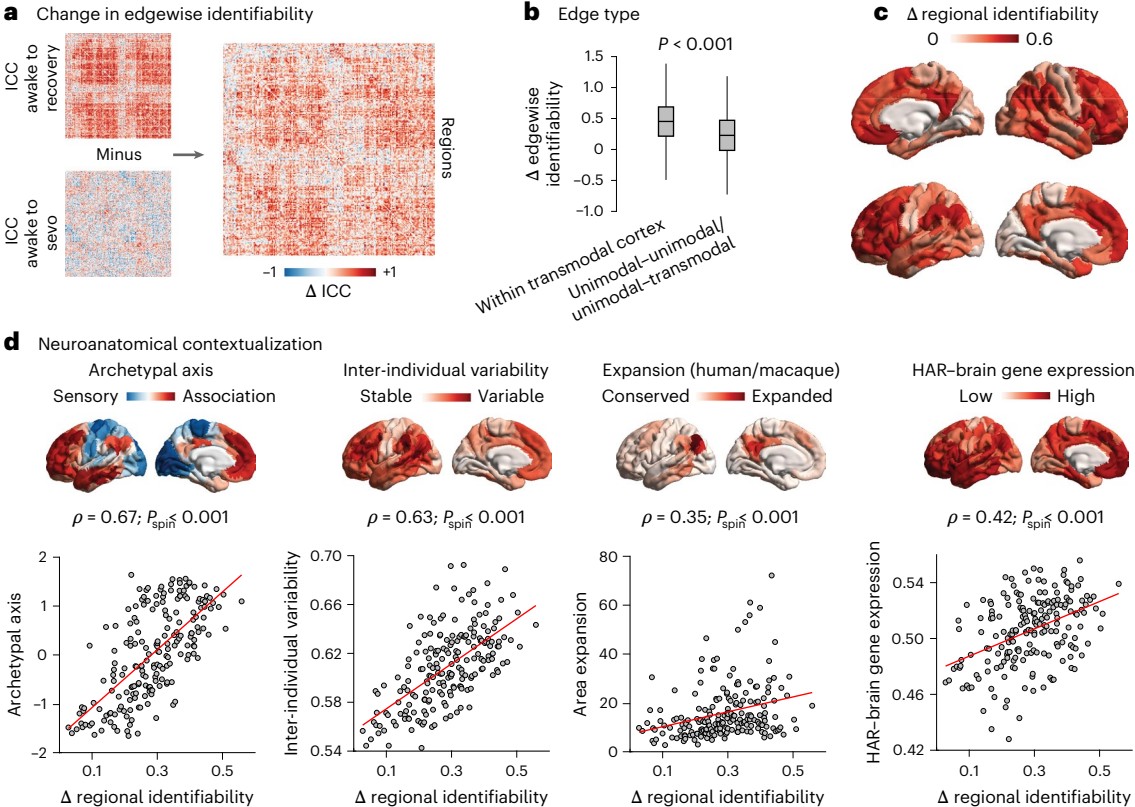

**Fig. 2 | Anatomical characterization of contributions to sevoflurane-induced loss of functional identifiability. a**, Edge-level difference in ICC between awake versus recovery and awake versus sevoflurane. **b**, The anaesthetic-induced loss of ICC is significantly more pronounced for functional connections within the transmodal cortex than those involving unimodal regions. For edges within the transmodal cortex, mean = 0.44 and s.d. = 0.34; for other edges, mean = 0.22 and s.d. = 0.35; $t(39,998)$ = 55.93; $P < 0.001$; effect size (Hedge's $g$) = 0.63; CI = [0.60, 0.65]; two-sided $t$-test). The ICC ranges between −1 and +1. $n$ = 5,580 edges within the transmodal cortex and $n$ = 34,420 unimodal–transmodal and unimodal–unimodal edges. In the box plot, the central lines indicate median values, the bounds of the boxes indicate the 25th and 75th percentiles and the whiskers indicate 1.5× the interquartile range. **c**, Regional distribution of anaesthetic-

induced loss of ICC, projected onto the cortical surface. **d**, The anaesthetic-induced regional loss of ICC is significantly spatially aligned with the archetypal sensory–association axis of cortical organization (Spearman's $\rho$ = 0.67; $p_{spin} < 0.001$; $n$ = 200 regions), the regional distribution of inter-individual variability of FC (Spearman's $\rho$ = 0.63; $p_{spin} < 0.001$; $n$ = 200 regions), the regional distribution of cortical expansion between macaque and human brains (Spearman's $\rho$ = 0.35; $p_{spin} < 0.001$; $n$ = 200 regions) and the regional expression of human-accelerated genes pertaining to brain function and development (that is, HAR–brain genes) (Spearman's $\rho$ = 0.42; $p_{spin} < 0.001$; $n$ = 200 regions). For each brain map, the range of values spanned by the colour bar is displayed on the $y$ axis of the scatter plot directly underneath. Source data are provided.

brain maps from Neurosynth (averaged across the entire scan duration) was lower at deeper levels of anaesthesia (Fig. 3b). This trend was reversed upon recovery of responsiveness (Fig. 3b; full statistics shown in Supplementary Data 1). The anaesthetic-induced decrease in the quality of cognitive matching was more pronounced for Neurosynth maps that loaded onto the higher-order (transmodal/association) end of the brain's archetypal axis (for example, cognitive control or emotion regulation) than for maps that loaded onto the unimodal/sensory end (for example, fixation and movement; Supplementary Fig. 7). In other words, anaesthesia diminishes the extent to which spontaneous brain activity reflects cognitive patterns from the literature, particularly for higher-order cognitive operations, potentially explaining why individual distinctiveness is suppressed by anaesthesia.

### Anaesthesia shifts human FC closer to macaque FC

Finally, we investigated whether anaesthesia changes the similarity between FC of the human brain and FC of the macaque brain. We used fMRI data from $n$ = 10 macaques scanned while awake[41] and processed similarly to human data[42], as well as independently processed fMRI data from The Virtual Brain project[43,44], comprising $n$ = 9 adult macaques that were lightly anaesthetized with 1.0–1.5% isoflurane. To enable comparison between the two species, we parcellated both the macaque and human data according to the regional mapping parcellation of Kötter

and Wanke[45], which was devised to enable inter-species comparisons and has recently been translated between macaque and human brains by ref. 46, such that each cortical region is anatomically matched to its homologue across the two species (see Supplementary Fig. 8).

We then used principal components analysis (PCA) to project all concatenated FC patterns across humans and macaques in a common low-dimensional space (see Methods). PCA is widely used for dimensionality reduction and the visualization of high-dimensional data because it provides a low-dimensional representation of the data while preserving as much of the original variability as possible. This approach enabled us to re-represent each individual's FC as a point in a two-dimensional (2D) plane, where each dimension corresponded to one of the main axes of variation in the data. We could then follow how the location of individuals' FC changed in this low-dimensional space as a function of anaesthesia.

We clearly observed that each condition (awake, recovery and various levels of sevoflurane anaesthesia) tended to occupy a different region of the space spanned by the first two principal components (Fig. 4a). Since principal component 1 (PC1) appeared to primarily reflect the difference between one of our macaque datasets and all other data, we focused our main analysis on PC2, which captured the differences in human states (similar results were observed when considering both PC1 and PC2; Supplementary Figs. 9 and 10). It was

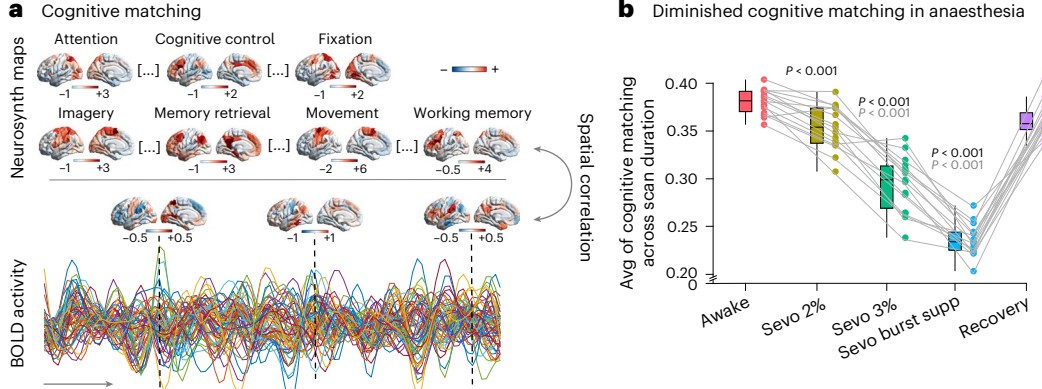

**Fig. 3 | Cognitive matching of brain activity with canonical meta-analytic patterns is diminished under anaesthesia and restored upon recovery.**
**a**, At each point in time, the cognitive matching score was computed as the best spatial correlation between brain activity and 123 Neurosynth meta-analytic maps. For each participant, an overall index of the quality of cognitive matching was then obtained by averaging the cognitive matching scores across the entire scan duration within each condition. **b**, The $y$ axis indicates the mean across time of the cognitive matching score ($n$ = 15 human volunteers). In the box plots,

the central lines indicate median values, the bounds of the boxes indicate the 25th and 75th percentiles, the whiskers indicate 1.5× the interquartile range and extreme values are shown as individual circles. $P$ values were obtained from repeated-measures $t$-tests (two sided) and false discovery rate corrected for multiple comparisons. Significance values for comparisons with the awake data are shown in black and those for comparisons versus the recovery data are shown in grey. See Supplementary Data 1 for full statistical reporting. Source data are provided. Avg, average; supp, suppression.

immediately apparent that as the dose of sevoflurane increased, the FC patterns of our human participants moved progressively further away along PC2 from their initial position during pre-anaesthesia wakefulness (red circles in Fig. 4a,b)—only to then return closest to their initial awake position upon post-anaesthetic recovery of wakefulness (purple circles in Fig. 4a,b). We formally quantified this shift in terms of Euclidean distance along PC2: we found that the human condition with the smallest PC2 distance from the awake human data was recovery, and deeper levels of sevoflurane anaesthesia corresponded to further distance away from awake along PC2 (Fig. 4c). At the same time, we observed that as the human anaesthetized FC patterns moved away from wakefulness, they also reduced their distance to the location of both macaque FC datasets along PC2 (Fig. 4d,e)—with burst suppression (the deepest level of human anaesthesia) being both furthest away from human wakeful FC (Fig. 4c) and closest to macaque FC along PC2 (Fig. 4d,e). Analogous results were observed when considering the space of both of the first two principal components (Supplementary Figs. 9 and 10) or when using cosine distance instead of Euclidean distance (Supplementary Fig. 11). See also Supplementary Fig. 12 for a representation in the space of the first three principal components instead.

Altogether, this low-dimensional representation highlights how anaesthesia shifts the FC of the human brain away from wakefulness and closer to the FC of the non-human primate brain: the distance between human anaesthetized FC and macaque FC is smaller than the distance between human awake FC and macaque FC. This phenomenon is reversed upon post-anaesthetic recovery, whereupon human FC moves back near the original position that it occupied at baseline. These results complement our observation that anaesthetic-induced reduction in regional identifiability is most pronounced in regions of the human brain that are genetically most human specific (Fig. 2d).

### Replication, robustness and sensitivity
**Anaesthesia reduces within-state identifiability of the human functional connectome.** We have observed significantly reduced identifiability between wakefulness and anaesthesia compared with between wakefulness and recovery (Fig. 1d). This means that, given a resting state fMRI scan of an awake individual, it is easier to tell apart a second awake scan of that same individual from other individuals' awake scans, than to tell apart an anaesthetized scan of the same individual against

anaesthetized scans of other people. Do these results tell us anything beyond the observation that anaesthesia deviates from the awake state more than recovery does? Presumably, baseline and recovery should be more similar than baseline and anaesthesia, because baseline and recovery are in some sense the same brain state (that is, wakefulness), whereas baseline and anaesthesia are radically different states. Indeed, this is precisely what our analysis of self–self correlations showed, by indicating reduced self-similarity between awake and anaesthesia than between awake and recovery (Fig. 1c).

However, reduced self–self correlation alone does not logically guarantee reduced identifiability. Identifiability could theoretically stay the same or even increase if the self–other correlations were to exhibit an equivalent or greater reduction than the self–self correlations.

To empirically demonstrate that these results are not simply due to comparing within-state correlations against between-state correlations (with state in this context referring to wakefulness or anaesthesia), we took advantage of the fact that our sevoflurane data included multiple scans obtained under anaesthesia. This allowed us to compare two conscious scans (baseline and recovery) and two anaesthetized scans: either vol 2 versus 3% sevoflurane or vol 3% sevoflurane versus burst suppression (as noted in ref. 47, when multiple scans are available this approach is preferable to comparing two halves of the same scanning session because the latter involves comparing the same person and scan against different people and scans, thereby confounding individual identity and scan identity).

When comparing awake–recovery similarity against the similarity between vol 2 and 3% sevoflurane or between vol 3% sevoflurane and burst suppression, we found exactly the same pattern of results as in our main analysis (Supplementary Fig. 13). Self–self similarity was significantly diminished, not only between wakefulness and anaesthesia (as we previously showed), but also between two anaesthetized scans. Likewise, identifiability was also diminished under anaesthesia, delineating a clear unimodal–transmodal cortical pattern (Supplementary Figs. 13 and 14). Specifically, we found that as anaesthesia deepened, both self-similarity and the difference between self–self and self–other correlations (that is, identifiability) were progressively reduced, with the two distributions increasingly overlapping (Extended Data Fig. 2). This pattern is the reverse of what was recently found by Colenbier et al.[48], who showed that tightly controlled cognitive tasks

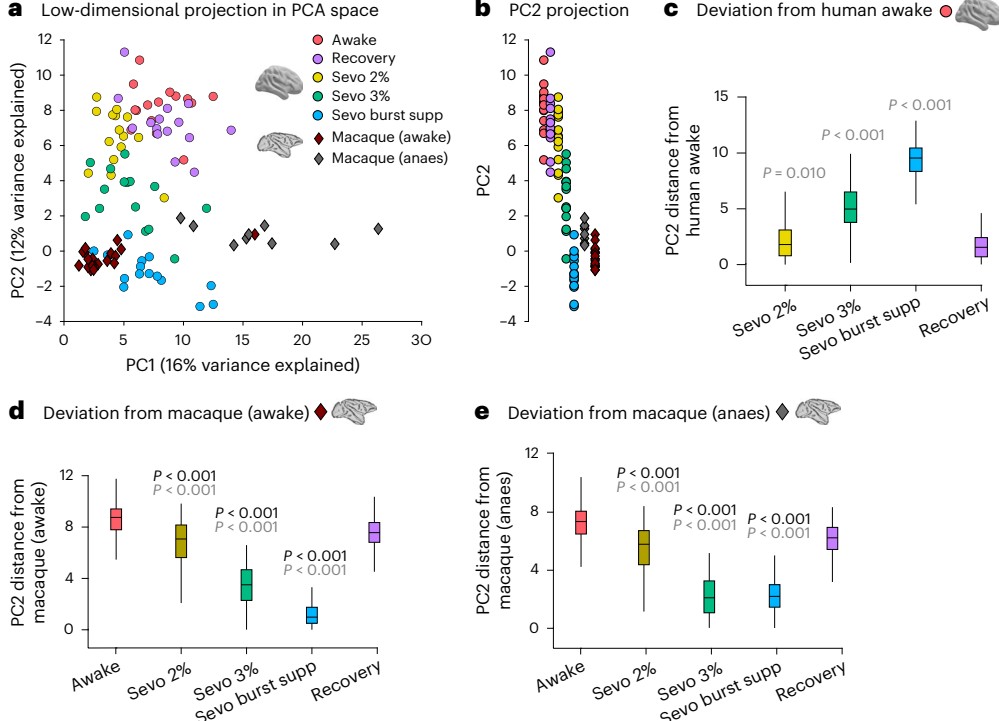

**Fig. 4 | Anaesthesia shifts human FC away from wakefulness and closer to macaque FC. a**, Low-dimensional projection of the human and macaque FC in the space of the first two principal axes of variation from PCA. Each circle represents the FC from one human, with colour reflecting the condition (awake, recovery or different doses of sevoflurane). Each diamond represents FC from one macaque, with colour representing the dataset (awake or anaesthetized (anaes)). **b**, Projection of the data from **a** onto PC2. **c**, Distribution of Euclidean distances from awake humans' FC patterns in the human dataset along PC2, as shown in **b**. *n* = 225 (15 × 15) pairs of data points in each box plot. **d**, Distribution of Euclidean distances between the human data and awake macaques' FC patterns along

PC2. *n* = 285 (15 × 19) pairs of data points. **e**, Distribution of Euclidean distances between the human data and anaesthetized macaques' FC patterns along PC2. *n* = 135 (15 × 9) pairs of data points. In **c**–**e**, significance values are versus the human recovery group (grey) or human awake group (black). For the box plots in **c**–**e**, the central lines indicate median values, the bounds of the boxes indicate the 25th and 75th percentiles and the whiskers indicate 1.5× the interquartile range. The *P* values were obtained from repeated-measures *t*-tests (two sided) and false discovery rate corrected for multiple comparisons. See Supplementary Data 2 for full statistical reporting. Source data are provided.

increase self–other similarity but increase self–self similarity even more, thereby resulting in an overall increase in identifiability.

Overall, since these results were obtained from within-state comparisons (awake–recovery (that is, both conscious) versus anaesthetized–anaesthetized), we can reject the possibility that changes in identifiability are merely a reflection of differences in brain states.

**Robustness to scan duration.** In addition to showing that anaesthesia reduces identifiability between different anaesthetized scans, we also demonstrated that anaesthetic-induced differences in self–self similarity and identifiability were not due to limitations of our scan duration. First, each of our scans had the same duration (see Methods), meaning that we do not need to be concerned about differences in scan duration as a potential confound. Second, this duration (approximately 10 min) is clearly more than sufficient to enable excellent brain fingerprinting, with all but one individual (93%) being identified when conscious in our awake–recovery data. This result is fully consistent with Van De Ville et al.[12], who reported that just over one minute of resting state fMRI is sufficient to achieve over 90% successful identification from brain fingerprinting. Third, we showed that even when an anaesthetized FC was obtained from combining all of the blood oxygenation level-dependent (BOLD) signals acquired across the three sevoflurane conditions (vol 2%, vol 3% and burst suppression, which were temporally concatenated before obtaining correlations between regions), nevertheless identifiability and self–self similarity were still reduced under anaesthesia (Supplementary Fig. 15). Thus, in accordance with the brain fingerprinting literature, our results of anaesthetic-induced

differences in self–self similarity and identifiability cannot be attributed to the scan duration being insufficient for fingerprinting. On the contrary, these results persist even after artificially stacking the deck in favour of anaesthesia by tripling the number of timepoints used for FC estimation, demonstrating their robustness.

**Replication with propofol anaesthesia.** We replicated our main results in a separate dataset of anaesthesia with the intravenous agent propofol[8,9]. Although this dataset did not reach the same depth of anaesthesia as was used in the main analysis, nevertheless the results are broadly consistent with what was observed under sevoflurane anaesthesia. Self–self similarity decreased during anaesthetic-induced loss of behavioural responsiveness (awake–recovery: mean = 0.68, s.d. = 0.07; awake–anaesthesia: mean = 0.48; s.d. = 0.13; *t*(15) = 6.81; *P* < 0.001; effect size (Hedge's *g*) = 1.82; confidence interval = [1.32, 2.74]) and so did differential identifiability (awake–recovery: mean = 0.20, s.d. = 0.06; awake–anaesthesia: mean = 0.08; s.d. = 0.13; *t*(15) = 4.35; *P* < 0.001; effect size (Hedge's *g*) = 1.14; confidence interval = [0.71, 1.77]; Extended Data Fig. 3). Likewise, the regional distribution of propofol-induced changes in identifiability was also more pronounced in the default mode and fronto-parietal than somatomotor and visual cortices, correlating with the sensory–association axis (Spearman's *ρ* = 0.46; *p*spin < 0.001; *n* = 200 regions) (Extended Data Fig. 4). As for sevoflurane, this result was obtained both when considering the ICC of each edge and when only including ICC values whose confidence interval did not include zero (Supplementary Fig. 16). We also replicated the correlation between the propofol-induced

regional change in identifiability and canonical maps of inter-individual variability (Spearman's $\rho$ = 0.43; $p_{spin}$ < 0.001; $n$ = 200 regions), evolutionary cortical expansion (Spearman's $\rho$ = 0.28; $p_{spin}$ = 0.001; $n$ = 200 regions) and expression of HAR–brain genes (Spearman's $\rho$ = 0.30; $p_{spin}$ < 0.001; $n$ = 200 regions) (Extended Data Fig. 4). Dominance analysis showed that the four canonical maps accounted for 27% of variance in the map of propofol-induced changes in regional identifiability, which was significantly more than expected from random spatial autocorrelation-preserving maps ($P$ < 0.001), with archetypal axis and inter-individual variability once again being the most important predictors (Supplementary Fig. 17).

The cognitive matching results from Neurosynth did not indicate statistically significant differences between propofol anaesthesia and either baseline or post-anaesthetic recovery of consciousness in terms of the maximum observed correlation between brain activity and meta-analytic maps (Extended Data Fig. 5 and Supplementary Data 1). However, if instead of only considering the best correlation we considered the average magnitude of correlations between brain activity and all Neurosynth maps, we found significant differences between baseline and anaesthesia, both in the sevoflurane and propofol datasets (Extended Data Fig. 6 and Supplementary Data 1). This latter analysis may be interpreted as the overall ability of meta-analytic patterns to recapitulate patterns of spontaneous brain activity. We also found that propofol anaesthesia, albeit less deep than the sevoflurane anaesthesia from our main analysis, shifted human FC away from wakefulness and towards the location of macaque FC (both awake and anaesthetized) along PC2 of a low-dimensional space obtained from PCA (Extended Data Fig. 7, Supplementary Fig. 18 and Supplementary Data 2).

**Replication of cognitive matching with BrainMap.** The results pertaining to the quality of decoding of brain activity based on the Neurosynth meta-analytic engine were also replicated using 66 unique behavioural domains obtained from an alternative meta-analytic database, BrainMap[49,50]. Whereas Neurosynth has a data-driven bottom-up approach to taxonomy and uses an automated process to identify statistical associations between brain coordinates and studies involving specific cognitive and behavioural terms, BrainMap is expert-curated. Despite the differences between the two databases (for example, Brain-Map explicitly excludes patient studies), we still found that the quality of decoding significantly deteriorated as the level of anaesthesia deepened, and was restored upon recovery of responsiveness (Supplementary Fig. 19 and Supplementary Data 1). This successful replication indicates that our cognitive matching procedure is robust both to the specific choice of which terms to include (which are different between BrainMap and our intersection of Neurosynth and the Cognitive Atlas[51]) and the choice of meta-analytic database more broadly.

**Robustness of the results to the use of different parcellations.** The present results were obtained using the Schaefer functional atlas, which is based on fMRI data of awake individuals[52]. To the best of our knowledge, there has been no report showing that the appropriateness of parcels in the Schaefer (or any other) functional atlas varies as a function of one's state of consciousness. In fact, we and others have successfully used the Schaefer atlas in previous works involving anaesthetic, psychedelic and pathological perturbations of consciousness[53,54], including for brain fingerprinting under altered states of consciousness[47,55]. Nevertheless, to show that our results are not critically dependent on the use of a functional parcellation derived from awake individuals, we replicated our results using an alternative parcellation of the cerebral cortex, the Desikan–Killiany atlas[56] (Supplementary Figs. 20 and 21 and Supplementary Data 1). This atlas is based on anatomical landmarks; therefore, the appropriateness of its parcels cannot be expected to vary under anaesthesia. Likewise, similar results were obtained when including 32 subcortical regions as defined by the recent Tian atlas (Supplementary Fig. 22 and Supplementary Data 1). In particular,

among subcortical structures, we observed an especially high regional contribution of the bilateral globus pallidus to the anaesthetic-induced change in regional identifiability. We also found that the Neurosynth cognitive matching results were robust to the choice of parcellation and inclusion of subcortex (Supplementary Fig. 23). The propofol results can also be replicated using the anatomical Desikan–Killiany atlas (Supplementary Figs. 24 and 25). Overall, we clearly demonstrate that our present results are robust to both parcellation size (from 68 to 200 regions) and type (functional or anatomical).

**Robustness against head motion.** For the cognitive matching analyses, we report the correlation of each contrast with the corresponding difference in mean framewise displacement (Supplementary Data 1). Whereas a correlation was present in the propofol dataset, no significant correlation between cognitive matching results and head motion was found in the main sevoflurane dataset. For the fingerprinting analysis, we observed that the results were not merely driven by the presence of high-motion participants. To demonstrate this, we repeated the analysis after applying a stringent criterion, excluding any participants with a mean framewise displacement of >0.3 in any condition, resulting in the exclusion of $n$ = 3 participants, leaving $n$ = 12 for analysis. The results were essentially unaltered, with anaesthesia significantly reducing both self–self similarity and differential identifiability (Supplementary Fig. 26). The results from the low-dimensional projection of FC in PCA space were also unaltered upon excluding the same three high-motion individuals (Supplementary Fig. 27). Thus, our results were not unduly influenced by head motion in the scanner.

## Discussion

Here we used pharmacological MRI under the effects of sevoflurane and propofol to determine whether anaesthetic-induced unconsciousness diminishes the distinctiveness of the human brain, both with respect to the brains of other individuals and the brain of another species entirely. We found that under deep anaesthesia, individual brains become less self-similar and less identifiable in terms of FC. Spatially, this effect is driven by reduced identifiability in transmodal association cortices.

Specifically, we found that the functional connections whose contributions to identifiability are most affected are those that most contribute to identifiability at baseline (Supplementary Fig. 1), which are also those connecting pairs of transmodal regions (Fig. 2b). These results are consistent with the notion that transmodal cortices, such as the default network and fronto-parietal control network, are particularly susceptible to anaesthesia and loss of consciousness more generally[9,57,58]. In addition, association cortices exhibit the greatest rate of inter-individual variability[35]. This variability is not mere noise, however, since the fronto-parietal and default networks consistently provide the largest contribution to identifiability in conscious individuals[10,12,14], indicating that their variability is individual specific. This may be attributed to the fact that transmodal association cortices have the longest maturation times in the human brain and the highest levels of synaptic plasticity and turnover[34,59,60]. Additionally, they also exhibit the lowest levels of intracortical myelination[34,61], which is known to suppress plasticity both mechanically and chemically[59,62]. As a result, transmodal cortices are relatively unconstrained by the underlying patterns of microstructure and anatomical connectivity[34,63–65] and are thus poised to change and adapt in response to environmental demands during the lifetime of each individual, which would account for their ability to encode individual-specific information in their FC. This individual-specific information in the functional interactions is then temporarily (and reversibly) suppressed by anaesthesia, as the present results indicate. Indeed, this account is consistent with recent evidence that individual differences in the FC and grey matter volume of frontal regions predict individual susceptibility to the behavioural effects of propofol sedation[66].

Indeed, we speculate that anaesthetic-induced suppression of individual-specific differences in FC may be due to the consciousness-suppressing effects of anaesthesia. The default network in particular is well known to engage in reflections about one's own past and future, which by definition are unique to each individual[67–69]. By suppressing the idiosyncratic patterns of spontaneous thought that characterize the human brain even at rest, the present work indicates that anaesthetic-induced unconsciousness diminishes how such patterns are encoded in the macroscale activity and connectivity of the brain. Indeed, we found that as anaesthesia deepens, spontaneous brain activity is increasingly less well-characterized in terms of meta-analytic patterns pertaining to cognitive operations—whether automatically defined or expert curated. This effect is reversed upon recovery, despite the lingering presence of anaesthetic in the bloodstream.

Taken together, the results of diminished cognitive matching and diminished identifiability driven by loss of self-similarity suggest the following tentative account. During wakefulness, brain activity is driven by a combination of spontaneous physiological processes and also the unique stream of consciousness of each individual, which brain activity must reflect. When consciousness is suppressed by anaesthesia, the physiological processes are perturbed, but most importantly the main driver of what makes each person unique is gone, leading to reduced self-similarity and thus reduced identifiability, which is restored upon regaining consciousness.

Our results go beyond confirming that anaesthesia induces FC changes within individuals and reveal that anaesthesia changes the relationships between different brains. Specifically, the anaesthetized brain becomes not only less similar to its awake self, but also less similar to its anaesthetized self, and overall less identifiable from others' brains. In contrast, Deng et al.[66] found that when dividing participants based on their high or low susceptibility to propofol sedation (in terms of reaction time), FC differences between the two groups were amplified during sedation compared with at baseline. However, their results are not in contrast with our own—not only because Deng et al.[66] used a different analytic approach (comparing two groups versus brain fingerprinting of individuals), but also, crucially, because all but three of their participants were still conscious (level 3 of the Ramsay sedation scale), as clearly demonstrated by their ability to perform a behavioural task[66]. In contrast, participants in our datasets reached levels 5 (for propofol) and 6 (for sevoflurane) of the Ramsay sedation scale (that is, the deepest levels, corresponding to full loss of behavioural responsiveness and presumably loss of consciousness[70]), with the sevoflurane anaesthesia achieving surgical depth[6]. Thus, it is entirely possible that sedation (during which individuals are sleepy but still conscious and responsive) could preserve or even amplify individual differences in FC associated with individual differences in behavioural responsiveness, which are then abolished upon full loss of consciousness, whereupon behavioural differences are also abolished.

The same caveat about sedation versus fully fledged deep anaesthesia applies to a previous study of brain fingerprinting, which reported that individuals were still identifiable while under dexmedetomidine sedation (although a reduction in the difference between individuals' whole-brain FC patterns was also observed)[14]. Unlike propofol and sevoflurane (the two anaesthetic agents employed in the present study), dexmedetomidine induces a state that is physiologically analogous to non-rapid eye movement stage 3 sleep[71], preserving the individual's capacity for rapidly recovering oriented responsiveness to external stimulation[72]. The different effects of dexmedetomidine versus sevoflurane and propofol on behaviour, physiology and brain FC may be attributed to differences in their respective molecular mechanisms of action[72]: dexmedetomidine is an alpha-2 adrenergic agonist, whereas propofol and sevoflurane act primarily on GABA-A receptors[73,74]. It is an asset of the present study that we were able to replicate our results in two separate datasets with different anaesthetics, demonstrating that our findings are not drug specific.

In addition to these molecular differences between anaesthetics, participants in the study of Liu and colleagues[14] were at Ramsay level 3–4 and still conscious, as indicated by the fact that they were still able to respond to commands. Therefore, our results (which were replicated in two independent datasets using different anaesthetics) are not in contrast with those of ref. 14; rather, the two studies together suggest that functional brain fingerprints are relatively robust to changes in brain state and only exhibit significant disruption at high doses that also disrupt responsiveness and presumably consciousness.

Intriguingly, transmodal association cortices are not only the most heterogeneous between individuals, but also between species, exhibiting the greatest evolutionary expansion and the greatest expression of brain-related human-accelerated genes[34,37,38,75]. We found that regional contributions to the anaesthetic-induced loss of identifiability are spatially correlated with both evolutionary cortical expansion and regional mean expression of human-accelerated genes. Furthermore, we found that as anaesthesia deepens, it shifts the position of the human functional connectome closer to the macaque functional connectome in a joint low-dimensional space, returning close to the initial position upon recovery.

More broadly, the present results of diminished deviation between humans and macaques under deep anaesthesia are in line with previous work showing diminished deviation between structure and function under anaesthesia. Previous work had shown that across species, the anaesthetized brain's patterns of time-varying FC become more similar to its underlying structural connectivity[76–78] (but see ref. 79 for a report of locally decreased structure–function coupling under propofol anaesthesia). This phenomenon reflects diminished ability of the unconscious brain to engage in unusual patterns of connectivity that go beyond the dictates of anatomy. Intriguingly, psychedelics such as LSD and psilocybin (which induce hallucinations and highly bizarre subjective experiences) were recently found to have the opposite effect on structure–function coupling, making brain activity and connectivity less constrained by the underlying structural connectome[80–82]. Consistent with anaesthesia and psychedelics having opposite effects of structure–function relationships, a recent report suggested that psilocybin increases the idiosyncrasy of FC, resulting in greater differential identifiability[47], which is the opposite of what we found here with different anaesthetics. Of note, decreased idiosyncrasy of FC was instead recently reported with another psychedelic, ayahuasca[55], in ritualistic users of psychedelics (members of the Santo Daime religious community). This result suggests that psychedelics may be able to modulate FC idiosyncrasy in both directions, increasing distinctiveness among strangers but increasing similarity among individuals for whom the psychedelic experience is part of a shared, ritualized cultural experience, which is likely to induce a commonality of mental state among individuals. Indeed, Colenbier et al.[48] and Finn et al.[11] showed that brain identifiability can be modulated by different cognitive tasks. Therefore, although our main result is that brain identifiability is reduced upon anaesthetic-induced loss of consciousness, anaesthesia is clearly not the only way to reduce the distinctiveness of the functional connectome, and as the study of Liu et al.[14] shows, identifiability is relatively robust to anaesthetic exposure, only becoming reduced at high doses that are likely to suppress consciousness itself.

The present results suggest that the anaesthetized human brain is more similar to the brains of other primates, with specifically human-expanded regions being the most affected by anaesthesia. Future research may investigate whether psychedelics have the opposite effect on the human brain, resulting in an even greater difference between humans and macaques, especially since the primary molecular target of classic psychedelics—the 5HT-2A receptor—is particularly prevalent in evolutionarily expanded transmodal cortices[83,84].

This study has a number of limitations that should be borne in mind. First, we followed the common practice in the literature of using loss of behavioural responsiveness as a marker of loss of consciousness,

even though the two are conceptually distinct[85]. Although both sevoflurane and propofol have occasionally been reported to induce dreaming[86,87], such occurrences are rare and so it seems likely that most of our participants were truly unconscious, especially given the depth of anaesthesia employed, which in the sevoflurane dataset went so deep as to induce burst suppression. Nevertheless, future work may fruitfully expand on the present results by using additional methods to assess unconsciousness beyond loss of behavioural responsiveness, such as slow-wave activity saturation of the EEG[88] or the perturbational complexity index based on the combination of EEG and transcranial magnetic stimulation[89].

From the EEG literature, it is also well established that anaesthesia induces so-called anteriorization of the distribution of EEG alpha oscillations (8–12 Hz), with the peak of alpha power shifting from occipital to frontal electrodes[90–92]. Although we found that anaesthesia reduces the overall identifiability of the fMRI connectome, it may still seem counterintuitive that individuals' fMRI FC patterns become less similar to each other under anaesthesia, since the EEG topography is expected to undergo similar reconfigurations across individuals. However, it is essential to realize that it is not inconsistent for two objects A and B to each become more similar to a third object C while at the same time becoming less similar to each other (Supplementary Code 1; see also Supplementary Fig. 28 for illustrations of this phenomenon in 1D and 2D). Additionally, fMRI and EEG reflect different neurobiological processes and operate at different spatial and temporal scales. Magnetoencephalography and EEG co-fluctuation patterns of different frequency bands can look very different from each other and from fMRI, and carry different information for fingerprinting[93–95]. In particular, the phenomenon of EEG alpha anteriorization occurs at a timescale that is several orders of magnitude removed from the fMRI BOLD signal fluctuations studied here (8–12 Hz versus 0.008–0.090 Hz). Last, anteriorization pertains to the behaviour of regions, whereas fingerprinting is predicated on the interactions between different regions. In Supplementary Code 2, we provide an example of two systems that each undergo the same change in the spatial pattern of amplitude of activity of each element while simultaneously decreasing their correlation at the level of edges. Altogether, any of these factors could explain the coexistence of our fMRI results on reduced inter-individual and inter-species distinctiveness, with the phenomenon of anaesthetic-induced EEG anteriorization. Teasing these factors apart with dedicated studies that explicitly investigate magnetoencephalography and EEG brain fingerprinting under anaesthesia represents a promising avenue for future work.

Another clear limitation of the present study is the small sample size, due to the technical and ethical challenges of performing anaesthesia in the scanner. Indeed, we acknowledge that there is a need in the field for larger sample sizes. However, we replicated our results in a separate dataset, demonstrating generalizability across datasets and drugs. We also ensured the robustness of our results to the choice of parcellation (anatomical or functional) and to potential confounds such as head motion. Moreover, and encouragingly, identifiability and self–self similarity were higher between baseline wakefulness and recovery, even though they were the two scans most far apart in time, and everything else being equal, greater intervening time between scans would be expected to diminish identifiability. Additionally, the sevoflurane dataset entirely comprised male participants and the majority of participants in the propofol dataset were also male. We look forward to future replications in sex-balanced datasets as the field expands. We also acknowledge that the mapping of functional activation to psychological terms in Neurosynth does not distinguish activations from deactivations[15]. However, we believe that our replication with meta-analytic maps defined using BrainMap[50] provides reassurance about the validity of our approach. Nonetheless, we note that the effects of cognitive matching from Neurosynth, which were clearly dependent on the depth of anaesthesia (Fig. 3), were only observed for

the sevoflurane dataset, which reached deeper levels of anaesthesia. It will therefore be of particular interest to determine whether this result can be replicated in other datasets. In particular, this approach may prove valuable in datasets of patients with disorders of consciousness, where decoding brain responsiveness to task commands (for example, 'imagine playing tennis') has already enabled the identification of covert consciousness in behaviourally unresponsive patients[23–28,30,31]. However, this paradigm requires patients' ability to understand commands, keep them in working memory and perform them—a non-trivial requirement for individuals who have suffered severe brain damage. To reduce this burden, researchers have also begun using spontaneous brain response to engaging narratives (for example, clips from the movie Taken)[96]. However, this approach still requires language comprehension and working memory to follow the events. Decoding based on the match between meta-analytic maps and spontaneous brain activity without stimuli may further advance this line of research.

Altogether, the present results indicate that regardless of the specific anaesthetic used, anaesthetized human brains are less individually distinctive, both across individuals and even across species, with regions that are most heterogeneous across individuals and across species being especially affected.

## Methods

### Datasets

**Human sevoflurane data.** The sevoflurane data included here have been published before[6,33,97] and we refer the reader to the original publication for details[6]. The ethics committee of the medical school of the Technische Universitat Munchen (Munchen, Germany) approved the current study, which was conducted in accordance with the Declaration of Helsinki. Written informed consent was obtained from volunteers at least 48 h before the study session. Twenty healthy adult men (20–36 years of age; mean = 26 years) were recruited through campus notices and personal contact and compensated for their participation in the study. Before inclusion in the study, detailed information was provided about the protocol and risks and medical histories were reviewed to assess any previous neurologic or psychiatric disorder. A focused physical examination was performed and a resting electrocardiogram was recorded. Further exclusion criteria were the following: physical status other than American Society of Anesthesiologists physical status I, chronic intake of medication or drugs, hardness of hearing or deafness, absence of fluency in German, known or suspected disposition to malignant hyperthermia, acute hepatic porphyria, history of halothane hepatitis, obesity with a body mass index of >30 kg m⁻², gastrointestinal disorders with a disposition for gastroesophageal regurgitation, a known or suspected difficult airway and the presence of metal implants. Data acquisition took place between June and December 2013.

Sevoflurane concentrations were chosen so that participants tolerated artificial ventilation (reached at 2.0 vol%) and that burst suppression was reached in all participants (around 4.4 vol%). To make group comparisons feasible, an intermediate concentration of 3.0 vol% was also used. In the MRI scanner, participants were in a resting state with their eyes closed for 700 s. Since EEG data were simultaneously acquired during MRI scanning[6] (although they were not analysed in the present study), visual online inspection of the EEG was used to verify that participants did not fall asleep during the pre-anaesthesia baseline scan. Sevoflurane mixed with oxygen was administered via a tight-fitting facemask using an fMRI-compatible anaesthesia machine (Fabius Tiro; Dräger). Standard American Society of Anesthesiologists monitoring was performed: concentrations of sevoflurane, oxygen and carbon dioxide were monitored using a cardiorespiratory monitor (DatexaS; General Electric). After administering an end-tidal sevoflurane concentration of 0.4 vol% for 5 min, the sevoflurane concentration was increased in a stepwise fashion by 0.2 vol% every 3 min until the participant became unconscious, as judged by loss of responsiveness to the repeatedly spoken command squeeze my hand two consecutive

times. The sevoflurane concentration was then increased to reach an end-tidal concentration of approximately 3 vol%. When clinically indicated, ventilation was managed by the physician and a laryngeal mask suitable for fMRI (I-gel, Intersurgical) was inserted. The fraction of inspired oxygen was then set at 0.8 and mechanical ventilation was adjusted to maintain end-tidal carbon dioxide at steady concentrations of $33 \pm 1.71$ mmHg during burst suppression, $34 \pm 1.12$ mmHg during 3 vol% and $33 \pm 1.49$ mmHg during 2 vol% (throughout this article, mean $\pm$ s.d.). Norepinephrine was given by continuous infusion ($0.1 \pm 0.01$ μg kg$^{-1}$ min$^{-1}$) through an intravenous catheter in a vein on the dorsum of the hand, to maintain the mean arterial blood pressure close to baseline values (baseline: $96 \pm 9.36$ mmHg; burst suppression: $88 \pm 7.55$ mmHg; 3 vol%: $88 \pm 8.4$ mmHg; 2 vol%: $89 \pm 9.37$ mmHg; follow-up: $98 \pm 9.41$ mmHg). After insertion of the laryngeal mask airway, the sevoflurane concentration was gradually increased until the EEG showed burst suppression with suppression periods of at least 1,000 ms and about 50% suppression of electrical activity (reached at $4.34 \pm 0.22$ vol%), which is characteristic of deep anaesthesia. At that point, another 700 s of EEG and fMRI results were recorded. A further 700 s of data were acquired at steady end-tidal sevoflurane concentrations of 3 and 2 vol%, respectively (corresponding to Ramsay scale level 6, the deepest), each after an equilibration time of 15 min. In a final step, the end-tidal sevoflurane concentration was reduced to twice the concentration at loss of responsiveness. However, most of the participants moved or did not tolerate the laryngeal mask any more under this condition: therefore, this stage was not included in the analysis. Sevoflurane administration was then terminated and the scanner table was slid out of the MRI scanner to monitor post-anaesthetic recovery. The participants were manually ventilated until spontaneous ventilation returned. The laryngeal mask was removed as soon as the patient opened his mouth on command. The physician regularly asked the participant to squeeze their hand: recovery of responsiveness was noted to occur as soon as the command was followed. Fifteen minutes after the time of recovery of responsiveness, the Brice interview was administered to assess for awareness during sevoflurane exposure; the interview was repeated on the phone the next day. After a total of 45 min of recovery time, another resting state combined fMRI–EEG scan was acquired (with eyes closed, as for the baseline scan). When participants were alert, oriented, cooperative and physiologically stable, they were taken home by a family member or a friend appointed in advance.

Although the original study acquired both fMRI and EEG data, in the present work we only considered the fMRI data. Data acquisition was carried out on a 3-Tesla magnetic resonance imaging scanner (Philips Achieva Quasar Dual 3.0T 16CH) with an eight-channel, phased-array head coil. The data were collected using a gradient-echo-planar imaging sequence (echo time = 30 ms, repetition time (TR) = 1.838 s; flip angle = 75°; field of view = 220 × 220 mm$^2$; matrix = 72 × 72; 32 slices; slice thickness = 3 mm; inter-slice gap = 1 mm; acquisition time = 700 s; functional volumes = 350). The anatomical scan was acquired before the functional scan using a T1-weighted magnetization-prepared rapid gradient-echo (MPRAGE) sequence with 240 × 240 × 170 voxels (1 × 1 × 1 mm voxel size) covering the whole brain. A total of 16 volunteers completed the full protocol; one participant was excluded due to high motion, leaving $n = 15$ for analysis.

**Human propofol data.** The propofol data were collected between May and November 2014 at the Robarts Research Institute, Western University, London, Ontario (Canada) and have been published before[8,9]. The study received ethical approval from the Health Sciences Research Ethics Board and Psychology Research Ethics Board of Western University (Ontario, Canada). Healthy volunteers ($n = 19$) were recruited (18–40 years of age; 13 males). Volunteers were right handed, native English speakers and had no history of neurological disorders. In accordance with relevant ethical guidelines, each volunteer provided written informed consent and received monetary compensation for

their time. Due to equipment malfunction or physiological impediments to anaesthesia in the scanner, data from $n = 3$ participants (one male) were excluded from analyses, leaving a total $n = 16$ for analysis.

Resting state fMRI data were acquired at different propofol levels: no sedation (awake), deep anaesthesia (corresponding to a Ramsay score of 5) and also during post-anaesthetic recovery. As previously reported[9], for each condition, fMRI acquisition began after two anaesthesiologists and one anaesthesia nurse independently assessed the Ramsay level in the scanning room. The anaesthesiologists and anaesthesia nurse could not be blinded to the experimental conditions, since part of their role involved determining the participants' level of anaesthesia. Note that the Ramsay score is designed for critical care patients; therefore, participants did not receive a score during the awake condition before propofol administration. Rather, they were required to be fully awake, alert and communicating appropriately. To provide a further, independent evaluation of participants' level of responsiveness, they were asked to perform two tasks: a test of verbal memory recall and a computer-based auditory target detection task. Wakefulness was also monitored using an infrared camera placed inside the scanner. Propofol was administered intravenously using an AS50 auto syringe infusion pump (Baxter Healthcare). An effect site/plasma steering algorithm combined with the computer-controlled infusion pump was used to achieve stepwise sedation increments, followed by manual adjustments as required to reach the desired target concentrations of propofol according to the TIVA Trainer (European Society for Intravenous Anaesthesia) pharmacokinetic simulation program. This software also specified the blood concentrations of propofol, following the Marsh three-compartment model, which were used as targets for the pharmacokinetic model providing target-controlled infusion. After an initial propofol target effect site concentration of 0.6 μg ml$^{-1}$, the concentration was gradually increased by increments of 0.3 μg ml$^{-1}$ and the Ramsay score was assessed after each increment. A further increment occurred if the Ramsay score was <5. The mean estimated effect site and plasma propofol concentrations were kept stable by the pharmacokinetic model delivered via the TIVA Trainer infusion pump. A Ramsay level of 5 was achieved when participants stopped responding to verbal commands, were unable to engage in conversation and were rousable only to physical stimulation. Once both anaesthesiologists and the anaesthesia nurse all agreed that Ramsay sedation level 5 had been reached and participants stopped responding to both tasks, data acquisition was initiated. The mean estimated effect site propofol concentration was 2.48 (1.82–3.14) μg ml$^{-1}$ and the mean estimated plasma propofol concentration was 2.68 (1.92–3.44) μg ml$^{-1}$. The mean total mass of propofol administered was 486.58 (373.30–599.86) mg. These values of variability are typical for the pharmacokinetics and pharmacodynamics of propofol. Oxygen was titrated to maintain SpO$_2$ above 96%. At Ramsay 5 level, participants remained capable of spontaneous cardiovascular function and ventilation. However, the sedation procedure did not take place in a hospital setting; therefore, intubation during scanning could not be used to ensure airway security during scanning. Consequently, although two anaesthesiologists closely monitored each participant, the scanner time was minimized to ensure return to normal breathing following deep sedation. No state changes or movement were noted during the deep sedation scanning for any of the participants included in the study. Propofol was discontinued following the deep anaesthesia scan, and participants reached level 2 of the Ramsay scale approximately 11 min afterwards, as indicated by clear and rapid responses to verbal commands. This corresponds to the recovery period. As previously reported[9], once in the scanner, participants were instructed to relax with closed eyes, without falling asleep. Resting state fMRI results in the absence of any tasks were acquired for 8 min for each participant. A further scan was also acquired during auditory presentation of a plot-driven story through headphones (5 min long). Participants were instructed to listen while keeping their eyes closed. The present analysis focuses on the resting

state data only; the story scan data have been published separately[8] and will not be discussed further here.

As previously reported[9], MRI scanning was performed using a 3-Tesla Siemens Tim Trio scanner (32-channel coil), and 256 functional volumes (echo-planar images (EPI)) were collected from each participant, with the following parameters: slices = 33; inter-slice gap = 25%; resolution = 3 mm isotropic; TR = 2,000 ms; echo time (TE) = 30 ms; flip angle = 75 degrees; matrix size = 64 × 64. The order of acquisition was interleaved, bottom-up. Anatomical scanning was also performed, acquiring a high-resolution T1-weighted volume (32-channel coil; 1 mm isotropic voxel size) with a 3D MPRAGE sequence, using the following parameters: acquisition time (TA) = 5 min, TE = 4.25 ms; matrix size = 240 × 256; flip angle = 9° (ref. 9).

**Functional MRI pre-processing and denoising.** We applied a standard pre-processing pipeline in accordance with our previous publications with anaesthesia data[9,33]. Pre-processing was performed using the CONN toolbox, version 17f (CONN; http://www.nitrc.org/projects/conn)[98], implemented in MATLAB 2016a. The pipeline involved the following steps: removal of the first 10 s to achieve steady-state magnetization; motion correction; slice timing correction; identification of outlier volumes for subsequent scrubbing by means of the quality assurance/artefact rejection software ART (http://www.nitrc.org/projects/artifact_detect); and normalization to Montreal Neurological Institute (MNI-152) standard space (2 mm isotropic resampling resolution), using the segmented grey matter image from each participant's T1-weighted anatomical image, together with an a priori grey matter template.

Denoising was also performed using the CONN toolbox, using the same approach as in our previous publications with pharmaco-MRI datasets[9,33]. Pharmacological agents can induce alterations in physiological parameters (heart rate, breathing rate and motion) or neurovascular coupling. The anatomical component-based noise correction (aCompCor) method removes physiological fluctuations by extracting principal components from regions unlikely to be modulated by neural activity; these components are then included as nuisance regressors[99]. Following this approach, five principal components were extracted from white matter and cerebrospinal fluid signals (using individual tissue masks obtained from the T1-weighted structural MRI images)[98] and regressed out from the functional data together with six individual-specific realignment parameters (three translations and three rotations), as well as their first-order temporal derivatives. This was followed by scrubbing of outliers identified by ART using ordinary least squares regression[98]. Finally, the denoised BOLD signal time series were linearly detrended and bandpass filtered to eliminate both low-frequency drift effects and high-frequency noise, thus retaining frequencies between 0.008 and 0.090 Hz. The step of global signal regression has received substantial attention in the literature as a denoising method[100–102]. However, recent work has demonstrated that the global signal contains behaviourally relevant information[103] and, crucially, information about states of consciousness across pharmacological and pathological perturbations[104]. Therefore, in line with ours and others' previous studies, here we avoided global signal regression in favour of the aCompCor denoising procedure, which is among those recommended.

Finally, denoised BOLD signals were parcellated into 200 cortical regions of interest (ROIs) from the Schaefer atlas[52]. We also replicated our results with the 68-ROI anatomical Desikan–Killiany cortical parcellation[56], as well as with a combined cortical–subcortical atlas comprising 200 cortical ROIs from the Schaefer atlas and an additional 32 subcortical ROIs from the subcortical atlas of Tian and colleagues[105], as previously recommended[106]. For comparison with the macaque data, a human-adapted version of the 82-ROI cortical parcellation of Kötter and Wanke was used[45], as adapted by ref. 46 (see Supplementary Fig. 8). FC was estimated for each individual and each condition as the Pearson correlation between pairs of denoised and parcellated BOLD time series.

**Awake macaque fMRI data from PRIME-DE.** The first dataset of non-human primate MRI data was made available as part of the Primate neuroimaging Data-Exchange (PRIME-DE) monkey MRI data-sharing initiative—a recently introduced open resource for non-human primate imaging[41].

The data pre-processing and denoising followed the same procedures as in a previous publication[42]. We used fMRI data from rhesus macaques (*Macaca mulatta*) scanned at Newcastle University. This sample included 14 exemplars (12 male and two female) with an age distribution of 3.90–13.14 years and a weight distribution of 7.2–18.0 kg (full sample descriptions are available online at http://fcon_1000.projects.nitrc.org/indi/PRIME/files/newcastle.csv and http://fcon_1000.projects.nitrc.org/indi/PRIME/newcastle.html).

*Ethics approval.* All of the animal procedures performed were approved by the UK Home Office and complied with the Animal Scientific Procedures Act (1986) on the care and use of animals in research, as well as the European Directive on the protection of animals used in research (2010/63/EU). We support the Animal Research: Reporting of In Vivo Experiments principles on reporting animal research. All persons involved in this project were Home Office certified and the work was strictly regulated by the UK Home Office. Local Animal Welfare Review Body approval was obtained. Compliance and assessment to ensure that the 3Rs principles were met was conducted by the National Centre for the Replacement, Refinement and Reduction of Animals in Research. Animals in Science Committee (UK) approval was obtained as part of Home Office project license approval.

*Animal care and housing.* All animals were housed and cared for in a group-housed colony and animals performed behavioural training on various tasks for auditory and visual neuroscience. No training took place before MRI scanning.

*Macaque MRI acquisition.* Animals were scanned in a vertical Bruker 4.7T primate dedicated scanner, with single-channel or four- to eight-channel parallel imaging coils used. No contrast agent was used. Optimization of the magnetic field before data acquisition was performed by means of second-order shim with Bruker and custom scanning sequence optimization. Animals were scanned upright, with MRI-compatible head post or non-invasive head immobilization, and working on tasks or at rest (here, only resting state scans were included). Eye tracking, video and audio monitoring were employed during scanning. Resting state scanning was performed for 21.6 min (TR = 2,600 ms; TE = 17 ms; effective echo spacing = 0.63 ms; voxels size = 1.22 × 1.22 × 1.24; phase encoding direction = encoded in columns). Structural scans comprised a T1 structural, MDEFT sequence with the following parameters: TE = 6 ms; TR: = 750 ms; inversion delay = 700 ms; number of slices = 22; in-plane field of view = 12.8 × 9.6 cm²; voxels per grid = 256 × 192; voxel resolution = 0.5 × 0.5 × 2.0 mm³; number of segments = 8.

The macaque MRI data were pre-processed using the recently developed pipeline for non-human primate MRI analysis, Pypreclin, which addresses several specificities of monkey research. The pipeline is described in detail in the associated publication[107]. Briefly, it includes the following steps: (1) slice timing correction; (2) correction for the motion-induced, time-dependent $B_0$ inhomogeneities; (3) reorientation from acquisition position to template (here we used the recently developed National Institute of Mental Health Macaque Template—a high-resolution template of the average macaque brain generated from in vivo MRI of 31 rhesus macaques (*M. mulatta*)[108]); (4) realignment to the middle volume using FSL MCFLIRT function; (5) normalization and masking using Joe's Image Program (JIP-align)

routine (http://www.nmr.mgh.harvard.edu/~jbm/jip/; J. Mandeville, Massachusetts General Hospital, Harvard University), which is specifically designed for preclinical studies (the normalization step aligns (affines) and warps (nonlinear alignment using the distortion field) the anatomical data into a generic template space); (6) $B_1$ field correction for low-frequency intensity non-uniformities present in the data; and (7) coregistration of functional and anatomical images using JIP-align to register the mean functional image (the moving image) to the anatomical image (the fixed image) by applying a rigid transformation. The anatomical brain mask was obtained by warping the template brain mask using the deformation field previously computed during the normalization step. Then, the functional images were aligned with the template space by composing the normalization and coregistration spatial transformations.

*Denoising.* The aCompCor denoising method implemented in the CONN toolbox was used to denoise the macaque fMRI data, to ensure consistency with the human data analysis pipeline. White matter and cerebrospinal fluid masks were obtained from the corresponding probabilistic tissue maps of the high-resolution National Institute of Mental Health Macaque Template (eroded by 1 voxel); their first five principal components were regressed out of the functional data, as well as linear trends and six motion parameters (three translations and three rotations) and their first derivatives. To make human and macaque data comparable, the macaque data were also bandpass filtered in the same 0.008–0.090 Hz range that was used for the human data.

Out of the 14 total animals present in the Newcastle sample, ten had available awake resting state fMRI data; of these ten, all except the first animal had two scanning sessions available. Thus, the total was 19 distinct sessions across ten individual macaques.

**Anaesthetized macaque fMRI data from The Virtual Brain.** The Virtual Brain project provides a dataset of pre-processed macaque fMRI results comprising $n = 9$ adult male rhesus macaques (eight *M. mulatta* and one *Macaca fascicularis*; aged between 4 and 8 years) acquired under light isoflurane anaesthesia[43]. This is the same dataset as was used in a previous publication by some of the authors[44]; for consistency of reporting, we use the same wording. A full description of data acquisition and processing is provided in ref. 43. All surgical and experimental procedures were approved by the Animal Use Subcommittee of the University of Western Ontario Council on Animal Care and were in accordance with the Canadian Council on Animal Care guidelines, as previously reported[43].

Briefly, animals were lightly anaesthetized before their scanning session and anaesthesia was maintained using 1.0–1.5% isoflurane. The scanning was performed on a 7T Siemens MAGNETOM head scanner with the following parameters: structural MRI: sequence = 3D MPRAGE T1 weighted; slices = 128; voxel size = 0.5 mm isotropic; one session of 10 min (600 volumes) resting state fMRI: sequence = 2D multi-band EPI; TR = 1,000 ms; slices = 42; voxel size = $1.0 \times 1.0 \times 1.1$ mm³. As reported in the original publication[43], FSL's FEAT toolbox was used for pre-processing the fMRI data, which included motion correction, high-pass filtering, registration, normalization and spatial smoothing (full width at half maximum = 2 mm). Motion in the fMRI data was minimal, with an average framewise displacement across all animals and all scans of 0.015 mm (range = 0.011–0.019 mm). Global white matter and cerebrospinal fluid signals were linearly regressed using the 'Analysis of Functional NeuroImages' 3dDeconvolve function. The global mean signal was not regressed. The regional fMRI signal was then extracted for each ROI in the regional map parcellation for each resting state fMRI scan.

**Human structural connectome from the Human Connectome Project.** We used diffusion MRI data from the 100 unrelated participants (54 females and 46 males; mean age = 29.1 ± 3.7 years) of the Human Connectome Project (HCP) 900 participants data release[109]. All HCP

scanning protocols were approved by the local Institutional Review Board at Washington University in St. Louis. The diffusion-weighted imaging (DWI) acquisition protocol is covered in detail elsewhere[110]. The diffusion MRI scan was conducted on a Siemens 3T Skyra scanner using a 2D spin-echo single-shot multi-band EPI sequence with a multi-band factor of 3 and monopolar gradient pulse. The spatial resolution was 1.25 mm isotropic (TR = 5,500 ms; TE = 89.50 ms). The *b* values were 1,000, 2,000 and 3,000 s mm⁻². The total number of diffusion sampling directions was 90, 90 and 90 for each of the shells in addition to six $B_0$ images. We used the version of the data made available in DSI Studio-compatible format at http://brain.labsolver.org/diffusion-mri-templates/hcp-842-hcp-1021 (ref. 111).

We adopted previously reported procedures to reconstruct the human connectome from DWI data. The minimally pre-processed DWI HCP data[110] were corrected for eddy current and susceptibility artefact. DWI data were then reconstructed using q-space diffeomorphic reconstruction (QSDR)[112], as implemented in DSI Studio (https://dsi-studio.labsolver.org/). QSDR calculates the orientational distribution of the density of diffusing water in a standard space, to conserve the diffusible spins and preserve the continuity of fibre geometry for fibre tracking. QSDR first reconstructs DWI images in native space and computes the quantitative anisotropy in each voxel. These quantitative anisotropy values are used to warp the brain to a template quantitative anisotropy volume in Montreal Neurological Institute space using a nonlinear registration algorithm implemented in the statistical parametric mapping software. A diffusion sampling length ratio of 2.5 was used and the output resolution was 1 mm. A modified FACT algorithm[113] was then used to perform deterministic fibre tracking on the reconstructed data, with the following parameters[106]: angular cutoff = 55°; step size = 1.0 mm; minimum length = 10 mm; maximum length = 400 mm; spin density function smoothing = 0; and a quantitative anisotropy threshold determined by DWI signal in the cerebrospinal fluid. Each of the streamlines generated was automatically screened for its termination location. A white matter mask was created by applying DSI Studio's default anisotropy threshold (0.6 Otsu's threshold) to the spin distribution function's anisotropy values. The mask was used to eliminate streamlines with premature termination in the white matter region. Deterministic fibre tracking was performed until 1,000,000 streamlines were reconstructed for each individual.

For each individual, their structural connectome was reconstructed by drawing an edge between each pair of regions *i* and *j* from the Schaefer cortical atlas[52] if there were white matter tracts connecting the corresponding brain regions end to end. Edge weights were quantified as the number of streamlines connecting each pair of regions, normalized by ROI distance and size.

A group consensus matrix *A* across participants was then obtained using the distance-dependent procedure of Betzel and colleagues[114] to mitigate concerns about inconsistencies in the reconstruction of individual participants' structural connectomes. This approach seeks to preserve both the edge density and the prevalence and length distribution of inter- and intrahemispheric edge length distribution of individual participants' connectomes and is designed to produce a representative connectome[114,115]. This procedure produces a binary consensus network indicating which edges to preserve. The final edge density was 27%. The weight of each non-zero edge was then computed as the mean of the corresponding non-zero edges across participants.

**Canonical brain maps.** To contextualize our regional pattern of anaesthetic-induced changes in identifiability, we obtained relevant brain maps from the literature using neuromaps (https://netneurolab.github.io/neuromaps/). We fetched and parcellated the map of the sensory–association archetypal axis from ref. 34, the map of cortical expansion between macaques and humans from ref. 37 and the map of inter-individual variability of FC from ref. 35.

Human-accelerated genes are genes associated with so-called human-accelerated regions of the human genome, identified as a set of loci that displayed accelerated divergence in the human lineage by comparing the human genome with that of the chimpanzee (*Pan troglodytes*), one of our closest living evolutionary relatives[116,117]. Among these human-accelerated genes, HAR–brain genes pertain to brain function and development[38]. The map of mean regional expression of HAR–brain gene expression was obtained as follows. First, the list of 415 HAR–brain genes was obtained from ref. 38 (see the original publication for details of how these genes were selected). Then, regional gene expression for each of the 200 cortical regions of the Schaefer atlas was obtained using the abagen toolbox (https://abagen.readthedocs.io/)[118], following abagen's default processing workflow and mirroring data between homologous cortical regions to ensure adequate coverage of both the left (data from six donors) and right hemisphere (data from two donors). Distances between samples were evaluated on the cortical surface with a 2-mm distance threshold. Gene expression data were normalized across the cortex using outlier-robust sigmoid normalization. Of the resulting 15,633 genes, 392 were among the list of HAR–brain genes from Wei and colleagues[38]. Finally, the map of regional mean expression of HAR–brain genes was obtained as the regional mean normalized gene expression across the 392 genes.

**Signal-to-noise ratio map of human fMRI.** To quantify the regional signal-to-noise ratio of the fMRI signal in the human brain, we used the map originally made by ref. 36. Briefly, Shafiei et al.[36] computed the signal-to-noise ratio as the ratio of the regional BOLD time series' mean to its standard deviation for each region of $n = 201$ subjects of the HCP, and subsequently averaged across individuals to obtain a group-representative signal-to-noise ratio map of the human brain. For further details, we refer the reader to ref. 36.

### Brain fingerprinting

Brain fingerprinting refers to using brain-derived metrics (here the FC obtained from resting state fMRI) to discriminate individuals from each other, analogously to how the grooves on one's fingertips may be used to discern one's identity. This requires brain fingerprints (just like conventional fingerprints) to be different across different people (to avoid confusing distinct individuals) but consistent within the same individual (to track identity).

Let $A$ be the identifiability matrix (that is, the square, non-symmetric matrix of similarity between individuals' test and retest scans), such that the size of $A$ is $S$-by-$S$ (with $S$ being the number of individuals in the dataset). Each entry of $A$ is obtained as the correlation between the corresponding individuals' vectorized matrices of parcellated FC. Let $I_{self} = \langle A_{ii} \rangle$ represent the average of the main diagonal elements of $A$, which comprise the Pearson correlation values between scans of same individual. From now on, we refer to this quantity as self-identifiability or $I_{self}$. Similarly, let $I_{others} = \langle A_{ij} \rangle$ define the average of the off-diagonal elements of matrix $A$ (that is, the correlation between scans of different individuals $i$ and $j$). Then, we define the differential identifiability ($I_{diff}$) of the sample as the difference between both terms: $I_{diff} = (I_{self} - I_{others})$, which quantifies the difference between the average within-participant FCs similarity and the average between-participants FCs similarity. The higher the value of $I_{diff}$, the higher the individual fingerprint overall along the population[10].

We can also quantify the edgewise identifiability of individuals by using the ICC[10]. The ICC is a widely used measure in statistics, most commonly to assess the percentage of agreement between units (or ratings or scores) of different groups (or raters or judges)[119–121]. It describes how strongly units in the same group resemble each other. The stronger the agreement between the ratings, the higher its ICC value (as for Pearson's correlation, the ICC ranges between −1 and +1). We use ICC to quantify the extent to which the connectivity value of an edge (the FC value between two brain regions) could separate within and between participants. In other words, the higher the ICC, the higher the identifiability of the connectivity edge[10]. We implemented the ICC analysis using code available at https://github.com/eamico/MEG_fingerprints, as described in ref. 10. In practice, the ICC is estimated using the difference between sample mean squares: ICC = (MSB − MSW)/(MSB + ($k$ − 1) MSW), where MSB is the variability of the group means from the grand mean, MSW is the variability of the individual scores from their respective group means and $k$ is the sample size. The rationale is that if group membership has no relevance, the variability within groups should be the same as the variability between groups (that is, MSB = MSW and the ICC equals 0). However, if there is more variability within groups than between groups, a negative ICC will be observed. In our main analysis, we included all ICC values. However, we also replicated our results using only ICC values whose confidence interval did not include zero.

### Meta-analytic cognitive matching from Neurosynth

Continuous measures of the association between voxels and cognitive categories were obtained from Neurosynth—an automated term-based meta-analytic tool that synthesizes results from more than 14,000 published fMRI studies by searching for high-frequency key words (such as 'pain' and 'attention' terms) that are systematically mentioned in the papers alongside fMRI voxel coordinates (https://github.com/neurosynth/neurosynth)—using the volumetric association test maps[15]. This measure of association strength is the tendency that a given term is reported in the functional neuroimaging study if there is activation observed at a given voxel. Note that Neurosynth does not distinguish between areas that are activated or deactivated in relation to the term of interest, nor the degree of activation, only that certain brain areas are frequently reported in conjunction with certain words. Although more than 1,000 terms are catalogued in the Neurosynth engine, we refined our analysis by focusing on cognitive function and therefore limited the terms of interest to cognitive and behavioural terms. To avoid introducing a selection bias, we opted for selecting terms in a data-driven fashion instead of selecting terms manually. Therefore, terms were selected from the Cognitive Atlas, a public ontology of cognitive science[51], which includes a comprehensive list of neurocognitive terms. This approach totalled to $t = 123$ terms, ranging from umbrella terms (attention and emotion) to specific cognitive processes (visual attention and episodic memory), behaviours (eating and sleep) and emotional states (fear and anxiety) (note that the 123 term-based meta-analytic maps from Neurosynth do not explicitly exclude patient studies). The Cognitive Atlas subdivision has previously been used in conjunction with Neurosynth[122–124], so we opted for the same approach to make our results comparable to previous reports. The full list of terms included in the present analysis is shown in Supplementary Table 1. The probabilistic measure reported by Neurosynth can be interpreted as a quantitative representation of how regional fluctuations in activity are related to psychological processes. As with the resting state BOLD data, voxelwise Neurosynth maps were parcellated into 200 cortical regions according to the Schaefer atlas[52] (or 68 for the replication with the Desikan–Killiany atlas and 232 cortical and subcortical regions for replication with the subcortex included). Code to perform cognitive matching is available at https://github.com/netneurolab/luppi-cognitive-matching.

For each individual, their parcellated BOLD signals at each point in time were spatially correlated against each Neurosynth map, producing one value of correlation per Neurosynth map per BOLD volume. We refer to this operation as cognitive matching. For each volume, the quality of cognitive matching was quantified as the highest value of (positive) correlation across all maps. These values were subsequently averaged across all volumes to obtain a single value per condition per participant. As an alternative, instead of using the highest positive correlation, we also considered the mean magnitude of correlation (regardless of sign) across all maps, subsequently averaging across volumes as described above.

We also repeated the cognitive matching separately for Neurosynth maps exhibiting a positive spatial correlation with the archetypal axis of ref. 34 (see above; that is, those primarily exhibiting positive values in the transmodal association cortex) and for maps exhibiting a negative correlation with the archetypal axis, which primarily comprised activation in unimodal (sensory) cortices.

**Alternative meta-analytic matching from BrainMap.** Whereas Neurosynth is an automated tool, BrainMap is an expert-curated repository. It includes the brain coordinates that are significantly activated during thousands of different experiments from published neuroimaging studies[49,50]. As a result, Neurosynth terms and BrainMap behavioural domains differ considerably. Here we used maps in the Desikan–Killiany anatomical atlas, pertaining to 66 unique behavioural domains (the same as in ref. 123) obtained from 8,703 experiments. The full list of BrainMap terms included in the present analysis is shown in Supplementary Table 2. Experiments conducted on unhealthy participants were excluded, as well as experiments without a defined behavioural domain.

## Low-dimensional representation with PCA
We used PCA to obtain a low-dimensional representation of the human and macaque FC in a common space. PCA re-represents the data in terms of linearly uncorrelated (orthogonal) variables called principal components, which are extracted from the data themselves as the axes of maximum variation[125]. Therefore, PCA is widely used for dimensionality reduction and visualization of high-dimensional data because it provides a low-dimensional representation of the data while preserving as much of the original variability as possible.

To obtain the joint PCA space, we began by vectorizing the upper triangular of each FC matrix for each scan (awake, recovery and different levels of anaesthesia) of each individual in the human dataset. We also did the same for all available scans in the awake and anaesthetized macaque datasets. The vectorized FC patterns were then concatenated, forming separate columns of a matrix $M$ whose rows corresponded to FC edges. The PCA algorithm was then applied to this matrix $M$ to extract linearly orthogonal principal components of maximum variability (ranked in descending order of variance explained). The algorithm also provided weights that associated each column of $M$ with each of the extracted PCs. We used these weights to obtain a low-dimensional representation of each column of $M$ (corresponding to FC patterns) as a point in the space of principal components. In our main analyses, we used the first two principal components to define the low-dimensional space, but we also replicated our results using the first three principal components.

## Statistical analyses
The statistical significance of differences between conditions (here, levels of anaesthesia) was determined with permutation $t$-tests (paired sample and two sided) with 10,000 permutations. The use of permutation tests alleviated the need to assume normality of data distributions (which was not formally tested). All tests were two sided, with an $\alpha$ value of 0.05. The effect sizes were estimated using Hedge's measure of the standardized mean difference, $g$, which was interpreted in the same way as Cohen's $d$, but more appropriate for small sample sizes[126]. The Measures of Effect Size Toolbox for MATLAB (https://github.com/hhentschke/measures-of-effect-size-toolbox) was used[127]. Anaesthesia conditions were compared separately against wakefulness and recovery and the false positive rate against multiple comparisons was controlled using false discovery rate correction[128], separately for these two cases. The spatial correlation between brain maps was quantified with Spearman's correlation coefficient and its statistical significance was assessed non-parametrically via comparison against a null distribution of null maps with preserved spatial autocorrelation[124,129].

**Dominance analysis.** To consider all of the regional correlates together and evaluate their respective contributions, we performed a dominance analysis with all four canonical brain maps as predictors and the regional map of anaesthetic-induced ICC changes as the target. Dominance analysis seeks to determine the relative contribution (the dominance of each independent variable to the overall fit (adjusted $R^2$)) of the multiple linear regression model (https://github.com/dominance-analysis/dominance-analysis)[39]. This is done by fitting the same regression model on every combination of predictors ($2^p - 1$ submodels for a model with $p$ predictors). Total dominance is defined as the average of the relative increase in $R^2$ when adding a single predictor of interest to a submodel, across all $2^p - 1$ submodels. The sum of the dominance of all input variables is equal to the total adjusted $R^2$ of the complete model, making the percentage of relative importance an intuitive method that partitions the total effect size across predictors. Therefore, unlike other methods of assessing predictor importance, such as methods based on regression coefficients or univariate correlations, dominance analysis accounts for predictor–predictor interactions and is interpretable. We established the statistical significance of the dominance analysis model using a non-parametric permutation test (one sided), by comparing the empirical variance explained against a null distribution of $R^2$ obtained from repeating the multiple regression with spatial autocorrelation-preserving null maps[124,129].

### Reporting summary
Further information on research design is available in the Nature Portfolio Reporting Summary linked to this article.

## Data availability
The original pharmacological fMRI data are available from the corresponding authors of the original publications referenced herein. The Allen Human Brain Atlas transcriptomic database is available at https://human.brain-map.org/. Neurosynth is available at https://neurosynth.org/. The list of human-accelerated brain genes is available from the supplementary material of ref. 38. The Newcastle macaque fMRI data are available from the PRIME-DE database (http://fcon_1000.projects.nitrc.org/indi/indiPRIME.html). Macaque fMRI data from The Virtual Brain project[43] are available at https://openneuro.org/datasets/ds001875/versions/1.0.3. Diffusion MRI data for the HCP in DSI Studio-compatible format are available at http://brain.labsolver.org/diffusion-mri-templates/hcp-842-hcp-1021. Source data are provided with this paper.

## Code availability
We have made code for cognitive matching freely available at https://github.com/netneurolab/luppi-cognitive-matching. Code for brain fingerprinting is freely available at https://github.com/eamico/MEG_fingerprints. The code for spin-based permutation testing of cortical correlations is freely available at https://github.com/frantisekvasa/rotate_parcellation. DSI Studio is freely available at https://dsi-studio.labsolver.org/. The CONN toolbox is freely available at http://www.nitrc.org/projects/conn. The Pypreclin code is available at https://github.com/neurospin/pypreclin. The abagen toolbox is available at https://abagen.readthedocs.io/. The neuromaps toolbox is available at https://netneurolab.github.io/neuromaps/. Code for dominance analysis is freely available at https://github.com/dominance-analysis/dominance-analysis. The Measures of Effect Size Toolbox for MATLAB is freely available at https://github.com/hhentschke/measures-of-effect-size-toolbox. Supplementary Codes 1 and 2 are provided as Supplementary Information.

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

## Acknowledgements

We express our gratitude to the PRIMatE Data Exchange (PRIME-DE) initiative, the organizers and managers of PRIME-DE and all of the institutions that contributed to the PRIME-DE database (https://www.fcon_1000.projects.nitrc.org/indi/indiPRIME.html), with special thanks to the Newcastle team. We are also grateful to A. Grigis, J. Tasserie and B. Jarraya for assistance with the Pypreclin code. We are also grateful to K. Shen for sharing the human version of the regional map macaque parcellation. We thank members of the Network Neuroscience Lab for helpful discussion. We acknowledge support from the Natural Sciences and Engineering Research Council of Canada (via a Banting Postdoctoral Fellowship to A.I.L.; funding reference 202209BPF-489453-401636), FRQNT Strategic Clusters Program (2020-RS4-265502—Centre UNIQUE—Union Neuroscience & Artificial Intelligence—Quebec, via the UNIQUE Neuro-AI Excellence Award to A.I.L.), Brain Canada Foundation (through the Canada Brain Research Fund with the support of Health Canada; to D.B.), National Institutes of Health (grant numbers NIH R01 AG068563A and NIH R01 R01DA053301-01A1 to D.B.), Canadian Institutes of Health Research (grant numbers CIHR 438531 and CIHR 470425 to D.B.), Healthy Brains, Healthy Lives initiative (Canada First Research Excellence fund; to D.B.), Google (Research Award and Teaching Award; to D.B.), CIFAR Artificial Intelligence Chairs programme (Canadian Institute for Advanced Research; to D.B.), Stephen Erskine Fellowship of Queens' College,

Cambridge (to E.A.S.), Canada Excellence Research Chairs programme (215063; to A.M.O.), L'Oreal–UNESCO for Women in Science Excellence Research Fellowship (to L.N.), Natural Sciences and Engineering Research Council of Canada (to B.M.), Canadian Institutes of Health Research (to B.M.), Brain Canada Foundation Future Leaders fund (to B.M.), Canada Research Chairs Program (to B.M.), Michael J. Fox Foundation (to B.M.), Healthy Brains, Healthy Lives initiative (to B.M.) and SNSF Ambizione project 'Fingerprinting the brain: network science to extract features of cognition, behaviour and dysfunction' (grant number PZ00P2-185716; to E.A.). The seveoflurane data were acquired with the support of the Technical University of Munich. The human connectome data were provided by the HCP, WU-Minn Consortium (1U54MH091657; principal investigators D. V. Essen and K. Ugurbil) funded by the 16 National Institutes of Health institutes and centres that support the NIH Blueprint for Neuroscience Research, and by the McDonnell Center for Systems Neuroscience at Washington University. The funders had no role in study design, data collection and analysis, decision to publish or preparation of the manuscript.

## Author contributions

A.I.L., E.A. and B.M. conceived of the study idea. A.I.L. performed the formal analysis. D.G., R.I., A.R., D.J., L.N., A.M.O., E.A.S. and D.B. contributed data. E.A.S. and E.A. contributed to the interpretation. A.I.L. and B.M. wrote the manuscript with input from all authors.

## Competing interests

D.B. is a shareholder and advisory board member of MindState Design Labs. The other authors declare no competing interests.

## Additional information

Andrea I. Luppi ⬤ [1,2] ✉, Daniel Golkowski ⬤ [3], Andreas Ranft ⬤ [4], Rudiger Ilg [3,5], Denis Jordan ⬤ [6], Danilo Bzdok [1,7], Adrian M. Owen [8], Lorina Naci ⬤ [9], Emmanuel A. Stamatakis ⬤ [2], Enrico Amico ⬤ [10,11,12] & Bratislav Misic ⬤ [1]

[1]Montréal Neurological Institute, McGill University, Montréal, Québec, Canada. [2]Division of Anaesthesia and Department of Clinical Neurosciences, University of Cambridge, Cambridge, UK. [3]Department of Neurology, Klinikum Rechts der Isar, Technical University of Munich, Munich, Germany. [4]Department of Anesthesiology and Intensive Care, School of Medicine and Health, Technical University of Munich, Munich, Germany. [5]Asklepios Clinic, Department of Neurology, Bad Tölz, Germany. [6]Department of Anaesthesiology and Intensive Care Medicine, Klinikum Rechts der Isar, Technical University of Munich, Munich, Germany. [7]Mila, Quebec Artificial Intelligence Institute, Montréal, Québec, Canada. [8]Western Institute for Neuroscience, Western University, London, Ontario, Canada. [9]Trinity College Institute of Neuroscience, School of Psychology, Trinity College Dublin, Dublin, Ireland. [10]School of Mathematics, University of Birmingham, Birmingham, UK. [11]Centre for Human Brain Health, University of Birmingham, Birmingham, UK. [12]Centre for Systems Modelling and Quantitative Biomedicine, University of Birmingham, Birmingham, UK. ✉e-mail: al857@cam.ac.uk

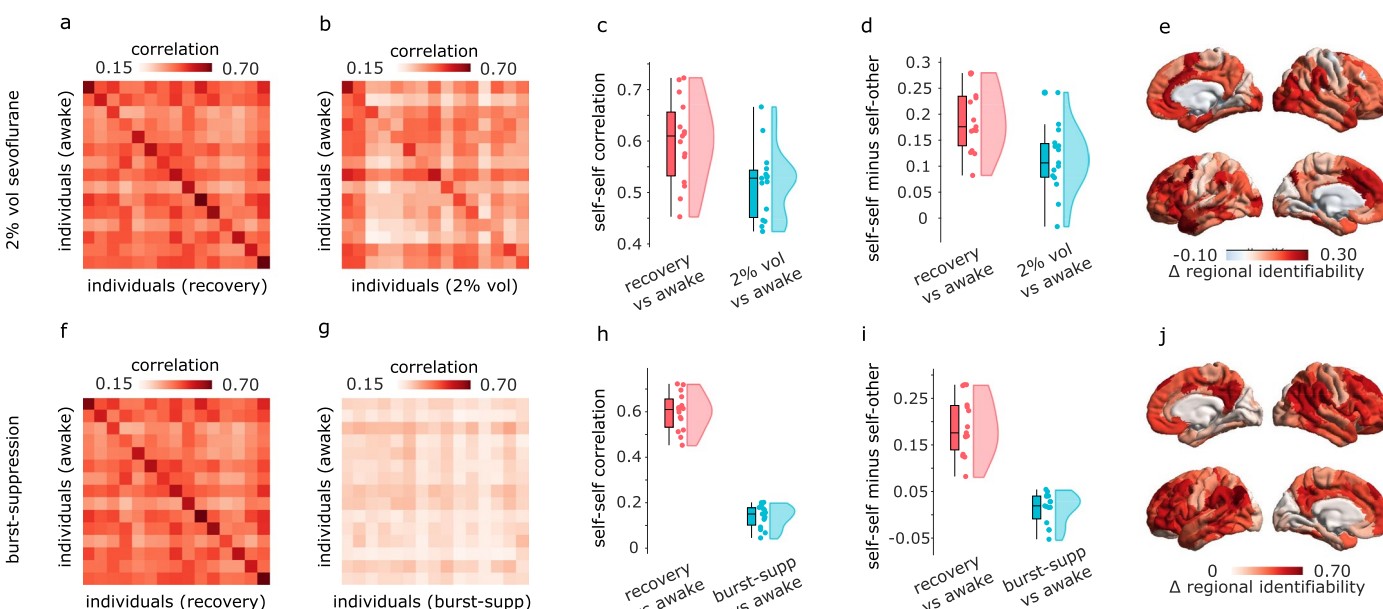

**Extended Data Fig. 1 | Replication of identifiability results at different doses of sevoflurane.** (**a**) Identifiability matrix between wakefulness and post-anaesthetic recovery. (**b**) Identifiability matrix between wakefulness and vol 2% sevoflurane anaesthesia (right). Entries along the diagonal, represent self-self similarity (correlation of FC patterns), whereas off-diagonal entries represent self-other similarity. (**c**) Self-self similarity is significantly higher between two conscious states, than between wakefulness and vol 2% sevoflurane. (**d**) The difference between self-self correlation and mean self-other correlation (differential identifiability) is significantly higher between two conscious states, than between wakefulness and vol 2% sevoflurane. (**e**) The regional distribution of contributions to identifiability (change in intra-class correlation coefficient) is plotted on the cortical surface. It is significantly spatially correlated with the corresponding map obtained with vol 3% sevoflurane: Spearman $\rho$ = 0.61, p$_{spin}$

< 0.001, N = 200 regions. (**f**) Identifiability matrix between wakefulness and post-anaesthetic recovery. (**g**) Identifiability matrix between wakefulness and burst-suppression level of sevoflurane anaesthesia. (**h**) Self-self similarity is significantly higher between two conscious states, than between wakefulness and burst-suppression level of sevoflurane. (**i**) The difference between self-self correlation and mean self-other correlation (differential identifiability) is significantly higher between two conscious states, than between wakefulness and burst-suppression level of sevoflurane. (**j**) The regional distribution of contributions to identifiability (change in intra-class correlation coefficient) is plotted on the cortical surface. It is significantly spatially correlated with the corresponding map obtained with vol 3% sevoflurane: Spearman correlation $\rho$ = 0.80, p$_{spin}$ < 0.001, N = 200 regions. N=15 human volunteers. Source data are provided as source data files.

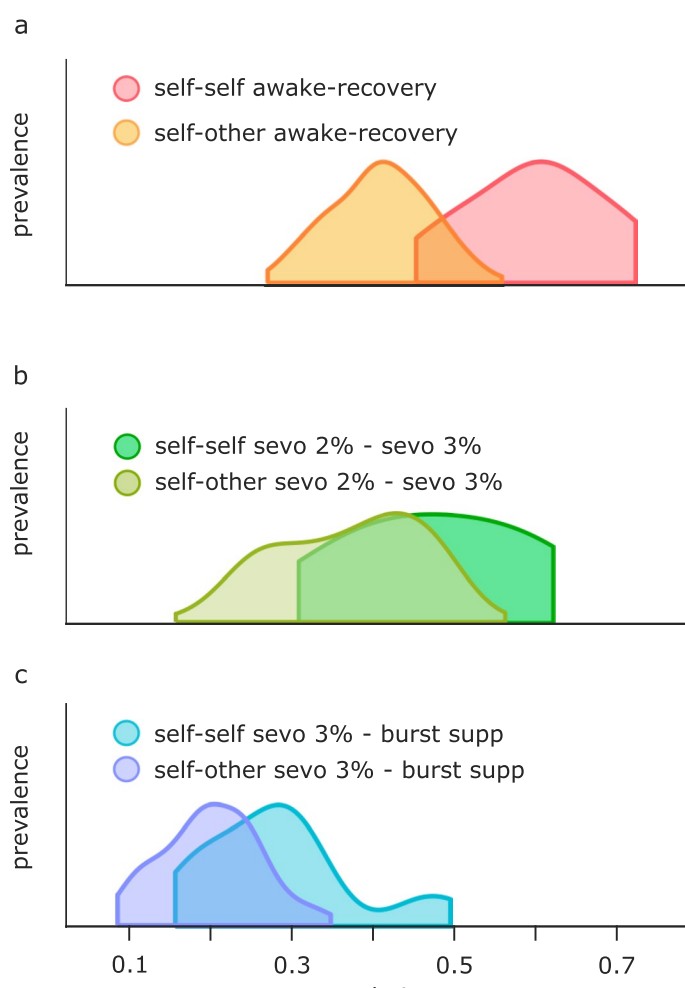

**Extended Data Fig. 2 | Distributions of self-self and self-other correlations for different pairs of conditions.** (**a**) Two conscious states: awake versus post-anaesthetic recovery. (**b**) Vol 2% versus vol 3% sevoflurane. (**c**) Vol 3% versus burst-suppression levels of sevoflurane.

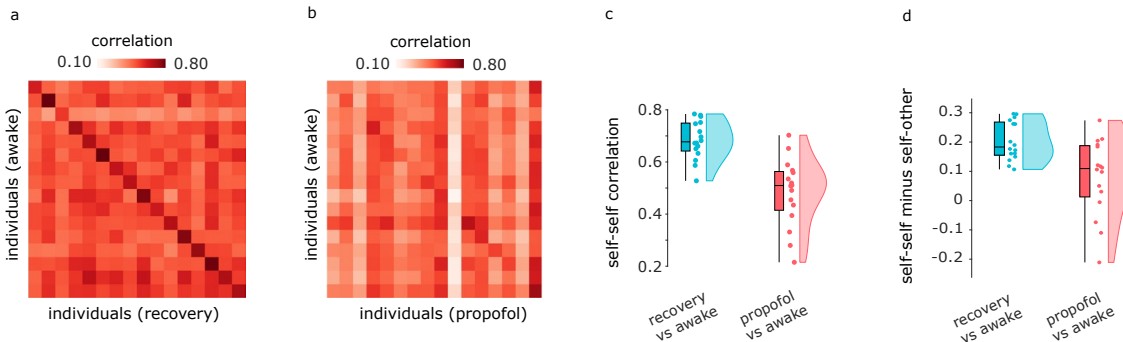

**Extended Data Fig. 3 | Identifiability of individual functional connectomes is diminished under propofol anaesthesia.** (**a**) Identifiability matrix between wakefulness and post-anaesthetic recovery. The rate of successful identification is 100 %. (**b**) Identifiability matrix between wakefulness and propofol anaesthesia (right). Entries along the diagonal, represent self-self similarity (correlation of FC patterns), whereas off-diagonal entries represent self-other similarity. (**c**) Self-self similarity is significantly higher between two conscious states, than between wakefulness and propofol anaesthesia. (**d**) The difference between self-self correlation and mean self-other correlation (differential identifiability) is significantly higher between two conscious states, than between wakefulness and propofol anaesthesia. Box-plot: center line indicates the median; bounds of the box indicate the 25th and 75th percentiles; whiskers indicate 1.5 × interquartile range; extreme values are shown as individual circles. N=16 human volunteers. Source data are provided as a source data file.

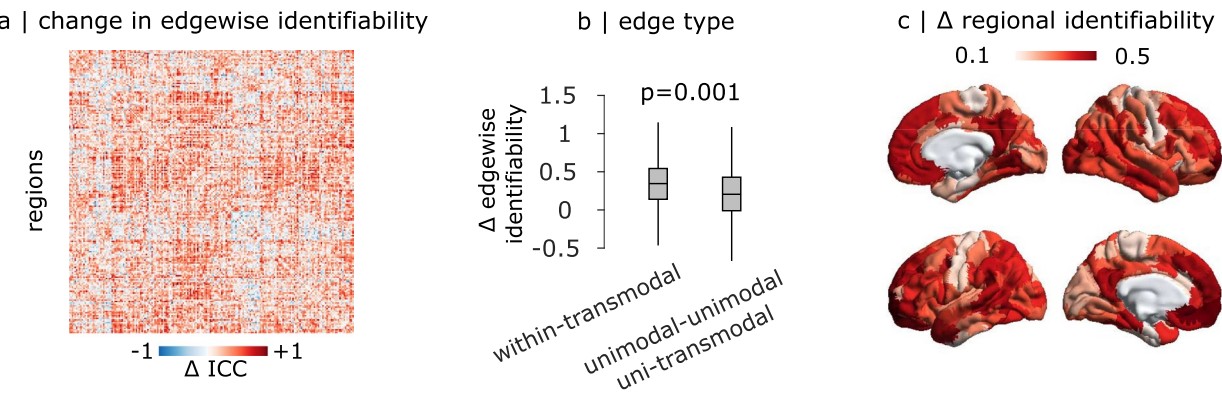

a | change in edgewise identifiability

regions

−1 +1
Δ ICC

b | edge type

p=0.001

Δ edgewise identifiability

within-transmodal

unimodal-unimodal
uni-transmodal

c | Δ regional identifiability

0.1 0.5

d | neuroanatomical contextualisation

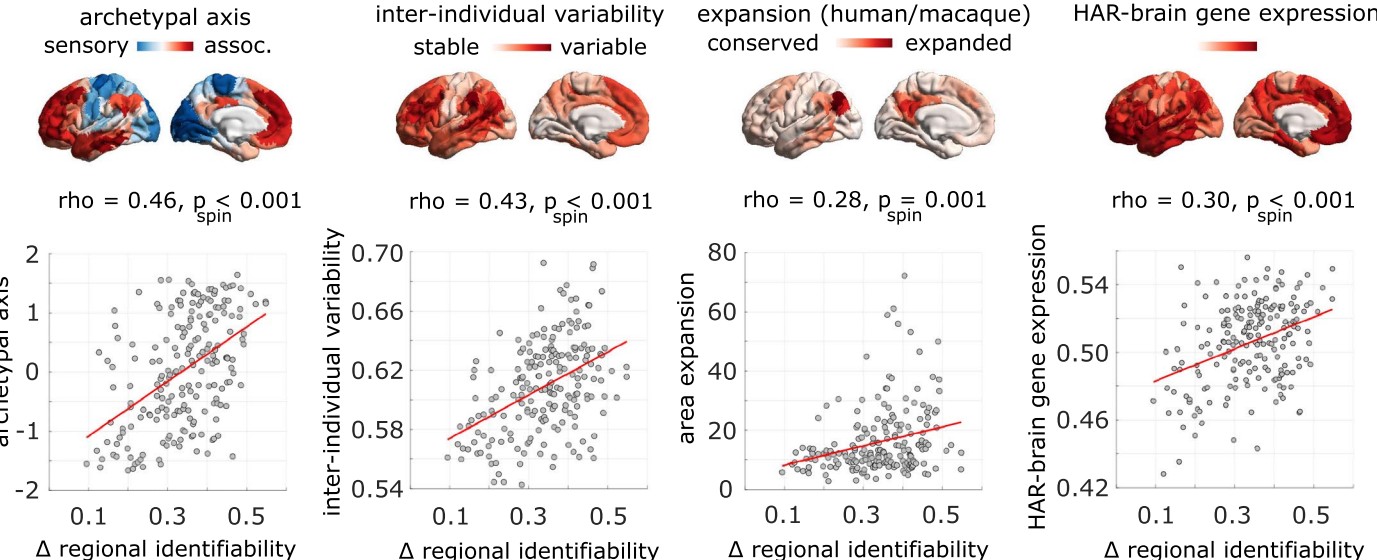

archetypal axis
sensory assoc.

rho = 0.46, $p_{spin}$ < 0.001

archetypal axis

Δ regional identifiability

inter-individual variability
stable — variable

rho = 0.43, $p_{spin}$ < 0.001

inter-individual variability

Δ regional identifiability

expansion (human/macaque)
conserved — expanded

rho = 0.28, $p_{spin}$ = 0.001

area expansion

Δ regional identifiability

HAR-brain gene expression

rho = 0.30, $p_{spin}$ < 0.001

HAR-brain gene expression

Δ regional identifiability

**Extended Data Fig. 4 | Anatomical characterisation of contributions to propofol-induced loss of identifiability.** (**a**) Edge-level difference in intra-class correlation coefficient between awake-recovery and awake-propofol. (**b**) The anaesthetic-induced loss of ICC is significantly more pronounced for functional connections within transmodal cortex, than those involving unimodal regions (two-sided t-test, n=5580 within-transmodal edges and n=34420 unimodal-transmodal and unimodal-unimodal edges). Box-plot: center line indicates the median; bounds of the box indicate the 25th and 75th percentiles; whiskers indicate 1.5 × interquartile range; extreme values are shown as individual circles. (**c**) Regional distribution of propofol-induced loss of ICC, projected onto the cortical surface. It is significantly spatially correlated with the corresponding

map obtained with sevoflurane: Spearman $\rho$ = 0.35, $p_{spin}$ < 0.001, N = 200 regions. (**d**) The propofol-induced regional loss of ICC is significantly spatially aligned with the archetypal sensory-association axis of cortical organisation; the regional distribution of inter-individual variability of functional connectivity; the regional distribution of cortical expansion between macaque and human brains; and the regional expression of human-accelerated genes pertaining to brain function and development ("HAR-brain genes"), as assessed with Spearman correlation across N = 200 regions. For each brain map, the range of values spanned by the color-bar is displayed on the y-axis of the scatter-plot directly underneath. Source data are provided as a source data file.

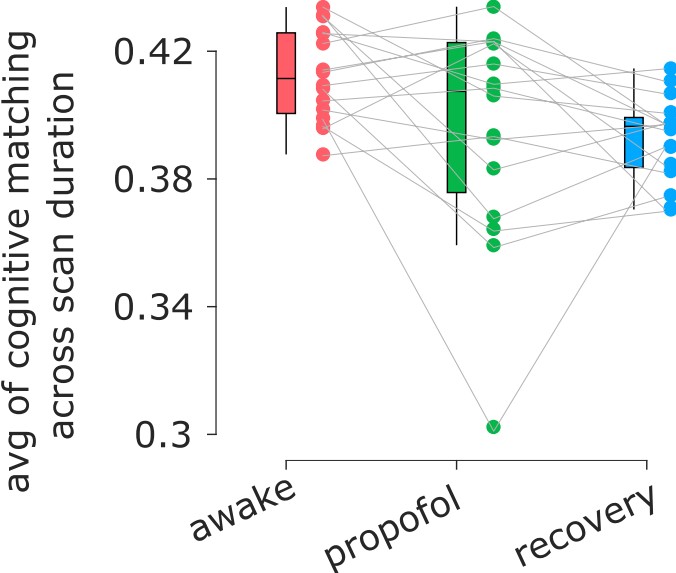

**Extended Data Fig. 5 | Cognitive matching from brain activity under wakefulness and propofol anaesthesia.** Ordinate: mean across time of the best decoding score (maximum spatial correlation between brain activity and 123 NeuroSynth meta-analytic maps). Box-plots: center line indicates the median; bounds of the box indicate the 25th and 75th percentiles; whiskers indicate 1.5 × interquartile range; extreme values are shown as individual circles. N=16 human volunteers. See Supplementary Data 1 for full statistical reporting. Source data are provided as a source data file.

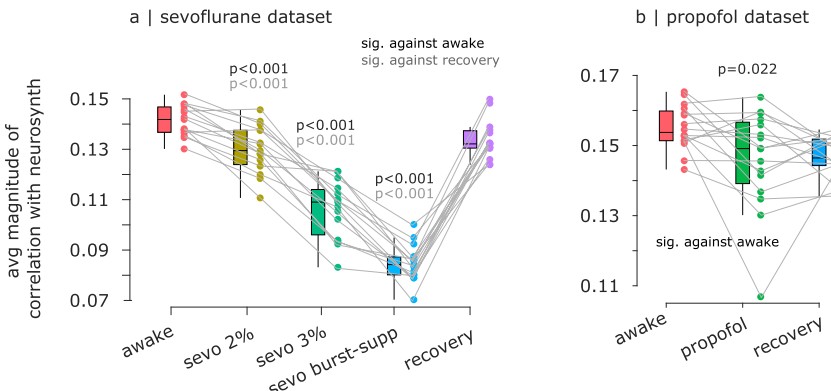

**Extended Data Fig. 6 | Alternative quantification of cognitive matching from brain activity under wakefulness and anaesthesia.** (**a**) Sevoflurane dataset. N=15 human volunteers. (**b**) Propofol dataset. N=16 human volunteers. For both (**a**) and (**b**): Ordinate indicates the temporal average of the mean absolute value of spatial correlation between brain activity and 123 NeuroSynth meta-analytic maps. Box-plots: center line indicates the median; bounds of the box indicate the 25th and 75th percentiles; whiskers indicate 1.5 × interquartile range; extreme values are shown as individual circles. P-values are obtained from repeated-measures t-tests (two-sided) and FDR-corrected for multiple comparisons. See Supplementary Data 1 for full statistical reporting. Source data are provided as a source data file.

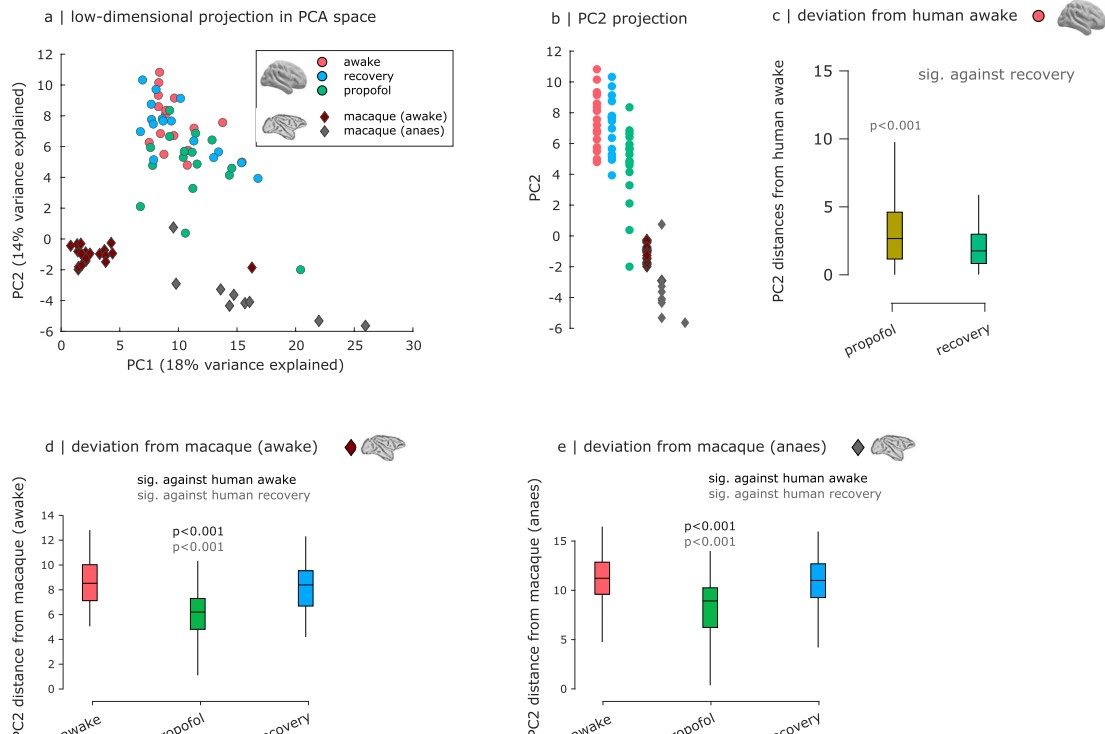

**Extended Data Fig. 7 | Propofol anaesthesia moves human functional connectivity away from wakefulness and closer to macaque functional connectivity in PCA space.** (**a**) Low-dimensional projection of the human (propofol dataset) and macaque functional connectivity, in the space of the first two principal axes of variation from Principal Components Analysis. Each circle represents the FC from one human, with colour reflecting condition (awake, recovery, or propofol anaesthesia). Each diamond represents FC from one macaque, with colour representing the dataset (awake or anaesthetised). (**b**) Projection of the data from (**a**) onto PC2. (**c**) Distribution of Euclidean distances from awake humans' FC patterns, in the human dataset, along PC2 as

shown in (**b**). N = 256 (16 × 16). (**d**) Distribution of Euclidean distances between the human data and awake macaques' FC patterns, along PC2. N = 304 (16 × 19) pairs of data-points. (**e**) Distribution of Euclidean distances between the human data and anaesthetised macaques' FC patterns, along PC2. N = 144 (16 × 9) pairs of data-points. Box-plots: center line indicates the median; bounds of the box indicate the 25th and 75th percentiles; whiskers indicate 1.5 × interquartile range. P-values are obtained from repeated-measures t-tests (two-sided) and FDR-corrected for multiple comparisons. See Supplementary Data 2 for full statistical reporting. Source data are provided as a source data file.

# Reporting Summary

## Statistics

For all statistical analyses, confirm that the following items are present in the figure legend, table legend, main text, or Methods section.

| n/a | Confirmed | |
|---|---|---|
| ☐ | ☒ | The exact sample size (*n*) for each experimental group/condition, given as a discrete number and unit of measurement |
| ☐ | ☒ | A statement on whether measurements were taken from distinct samples or whether the same sample was measured repeatedly |
| ☐ | ☒ | The statistical test(s) used AND whether they are one- or two-sided<br>*Only common tests should be described solely by name; describe more complex techniques in the Methods section.* |
| ☐ | ☒ | A description of all covariates tested |
| ☐ | ☒ | A description of any assumptions or corrections, such as tests of normality and adjustment for multiple comparisons |
| ☐ | ☒ | A full description of the statistical parameters including central tendency (e.g. means) or other basic estimates (e.g. regression coefficient) AND variation (e.g. standard deviation) or associated estimates of uncertainty (e.g. confidence intervals) |
| ☐ | ☒ | For null hypothesis testing, the test statistic (e.g. *F*, *t*, *r*) with confidence intervals, effect sizes, degrees of freedom and *P* value noted<br>*Give P values as exact values whenever suitable.* |
| ☒ | ☐ | For Bayesian analysis, information on the choice of priors and Markov chain Monte Carlo settings |
| ☒ | ☐ | For hierarchical and complex designs, identification of the appropriate level for tests and full reporting of outcomes |
| ☐ | ☒ | Estimates of effect sizes (e.g. Cohen's *d*, Pearson's *r*), indicating how they were calculated |

*Our web collection on statistics for biologists contains articles on many of the points above.*

## Software and code

Policy information about availability of computer code

| | |
|---|---|
| Data collection | The TIVA Trainer (European Society for Intravenous Aneaesthesia, eurosiva.eu) pharmacokinetic simulation program was used to control the propofol infusion. An Alaris PK infusion pump (Carefusion, Basingstoke, UK) was used for propofol infusion. |
| Data analysis | We have made code for cognitive matching freely available at https://github.com/netneurolab/luppi-cognitive-matching<br>The following third-party code was used.<br>Code for brain fingerprinting is freely available at https://github.com/eamico/MEG_fingerprints.<br>The code for spin-based permutation testing of cortical correlations is freely available at https://github.com/frantisekvasa/rotate_parcellation.<br>DSI Studio is freely available at https://dsi-studio.labsolver.org/.<br>The CONN toolbox is freely available at http://www.nitrc.org/projects/conn.<br>The Pypreclin code is available at https://github.com/neurospin/pypreclin.<br>The abagen toolbox is available at https://abagen.readthedocs.io/<br>The neuromaps toolbox is available at https://netneurolab.github.io/neuromaps/.<br>The Measures of Effect Size Toolbox for MATLAB is freely available at https://github.com/hhentschke/measures-of-effect-size-toolbox.<br>Supplementary Code 1 and Supplementary Code 2 are provided as supplementary materials. |

For manuscripts utilizing custom algorithms or software that are central to the research but not yet described in published literature, software must be made available to editors and reviewers. We strongly encourage code deposition in a community repository (e.g. GitHub). See the Nature Portfolio guidelines for submitting code & software for further information.

# Data

Policy information about availability of data

All manuscripts must include a data availability statement. This statement should provide the following information, where applicable:

- Accession codes, unique identifiers, or web links for publicly available datasets
- A description of any restrictions on data availability
- For clinical datasets or third party data, please ensure that the statement adheres to our policy

Source data are provided as a source data file.
The original pharmacological fMRI data are available
from the corresponding authors of the original
publications referenced herein. The Allen Human
Brain Atlas transcriptomic database is available
at https://human.brain-map.org/; NeuroSynth is
available at https://neurosynth.org/. The list of
human-accelerated brain genes is available from the
Supplementary Material of [93]. The Newcastle
macaque fMRI data are available from the PRIMEDE
database (http://fcon_1000.projects.nitrc.org/indi/
indiPRIME.html). Diffusion MRI data for the Human
Connectome Project in DSI Studio-compatible
format are available at http://brain.labsolver.org/
diffusion-mri-templates/hcp-842-hcp-1021. Macaque
fMRI data from TheVirtualBrain project [42] are
available at https://openneuro.org/datasets/ds001875/
versions/1.0.3.

# Research involving human participants, their data, or biological material

Policy information about studies with human participants or human data. See also policy information about sex, gender (identity/presentation), and sexual orientation and race, ethnicity and racism.

| | |
|---|---|
| Reporting on sex and gender | Each dataset had been previously collected. One dataset only included male participants; the other included participants of both sexes. The design is within-subjects, and our focus was not on comparing groups or inter-individual differences but rather on comparing states of anaesthesia. |
| Reporting on race, ethnicity, or other socially relevant groupings | No grouping by race, ethnicity, or socieconomic status was performed. |
| Population characteristics | See Life sciences reporting. |
| Recruitment | Sevoflurane dataset: Data acquisition took place between June and December 2013. Participants approaches the research team to seek participation. healthy adult men were recruited through campus notices and personal contact, and compensated for their participation in the study. Further exclusion criteria were the following: physical status other than American Society of Anesthesiologists physical status I, chronic intake of medication or drugs, hardness of hearing or deafness, absence of fluency in German, known or suspected disposition to malignant hyperthermia, acute hepatic porphyria, history of halothane hepatitis, obesity with a body mass index more than 30 kg/m2, gastrointestinal disorders with a disposition for gastroesophageal regurgitation, known or suspected difficult airway, and presence of metal implants.

propofol dataset: we recruited participants with posters around campus as per ethics protocol. Participants approaches the research team to seek participation, and there were no specific selection biases. Participants were required to be healthy, right-handed, native English speakers with no history of neurological disorders, and no contraindications to MRI scanning.

HCP dataset: Detailed information about the recruitment, acquisition and imaging is provided in the dedicated HCP publications. |
| Ethics oversight | All HCP scanning protocols were approved by the local Institutional Review Board at Washington University in St. Louis. Sevoflurane dataset: The ethics committee of the medical school of the Technische Universitat Munchen (Munchen, Germany) approved the current study. propofol dataset: The study received ethical approval from the Health Sciences Research Ethics Board and Psychology Research Ethics Board of Western University (Ontario, Canada). |

Note that full information on the approval of the study protocol must also be provided in the manuscript.

# Field-specific reporting

Please select the one below that is the best fit for your research. If you are not sure, read the appropriate sections before making your selection.

☒ Life sciences ☐ Behavioural & social sciences ☐ Ecological, evolutionary & environmental sciences

For a reference copy of the document with all sections, see nature.com/documents/nr-reporting-summary-flat.pdf

# Life sciences study design

All studies must disclose on these points even when the disclosure is negative.

| | |
|---|---|
| Sample size | This study used previously collected data. No power analysis was performed prior to data collection; the sample sizes are within the range reported in the literature.<br>sevoflurane dataset: n=20 participants were recruited .<br>propofol dataset: n=19 participants were recruited . |
| Data exclusions | Sevoflurane dataset: A total of 16 volunteers completed the full protocol and were included in our analyses; one participant was excluded due to high motion, leaving N=15 for analysis.<br>Propofol dataset: Due to equipment malfunction<br>or physiological impediments to anaesthesia in the scanner,<br>data from n=3 participants (1 male) were excluded<br>from analyses, leaving a total n=16 for analysis. |
| Replication | The propofol dataset was used to replicate all results obtained from the sevoflurane dataset. We also replicated the NeuroSynth results using the BrainMap meta-analytic dataset, and we replicated our results using a different brain parcellation (Desikan-Killiany). |
| Randomization | all participants were run in both conditions (awake and anaesthetised) since this was a repeated measures design. |
| Blinding | No blinding is possible, since anaesthetised state is tested by means of behavioural responsiveness. |

# Reporting for specific materials, systems and methods

We require information from authors about some types of materials, experimental systems and methods used in many studies. Here, indicate whether each material, system or method listed is relevant to your study. If you are not sure if a list item applies to your research, read the appropriate section before selecting a response.

### Materials & experimental systems

| n/a | Involved in the study |
|---|---|
| ☒ | ☐ Antibodies |
| ☒ | ☐ Eukaryotic cell lines |
| ☒ | ☐ Palaeontology and archaeology |
| ☐ | ☒ Animals and other organisms |
| ☒ | ☐ Clinical data |
| ☒ | ☐ Dual use research of concern |
| ☒ | ☐ Plants |

### Methods

| n/a | Involved in the study |
|---|---|
| ☒ | ☐ ChIP-seq |
| ☒ | ☐ Flow cytometry |
| ☐ | ☒ MRI-based neuroimaging |

# Animals and other research organisms

Policy information about studies involving animals; ARRIVE guidelines recommended for reporting animal research, and Sex and Gender in Research

| | |
|---|---|
| Laboratory animals | Functional MRI data were obtained from 10 exemplars of Macaca Mulatta, out of 14  (12 male, 2 female); Age distribution: 3.9-13.14 years; Weight distribution: 7.2-18 kg (full sample description available online: http://fcon_1000.projects.nitrc.org/indi/PRIME/files/newcastle.csv and http://fcon_1000.projects.nitrc.org/indi/PRIME/newcastle.html).<br><br>TheVirtualBrain project provides a dataset of preprocessed macaque fMRI comprising N=9 adult male rhesus macaques (8 Macaca mulatta, 1 Macaca fascicularis, aged between 4 and 8 years) acquired under light isoflurane anaesthesia. |
| Wild animals | The study did not involve wild animals. |
| Reporting on sex | Newcastle: Animals of both sexes were included.<br>TheVirtualBrain: male animals were used. |

| Field-collected samples | The study did not involve samples collected from the field. |
|---|---|
| Ethics oversight | Newcastle dataset: All of the animal procedures performed were approved by the UK Home Office and comply with the Animal Scientific Procedures Act (1986) on the care and use of animals in research and with the European Directive on the protection of animals used in research (2010/63/EU). We support the Animal Research Reporting of In Vivo Experiments (ARRIVE) principles on reporting animal research. All persons involved in this project were Home Office certified and the work was strictly regulated by the U.K. Home Office. Local Animal Welfare Review Body (AWERB) approval was obtained. The 3Rs principles compliance and assessment was conducted by National Centre for 3Rs (NC3Rs). Animal in Sciences Committee (UK) approval was obtained as part of the Home Office Project License approval.<br><br>TheVirtualBrain dataset: All surgical and experimental procedures were approved by the Animal Use Subcommittee of the University of Western Ontario Council on Animal Care and were in accordance with the Canadian Council of Animal Care guidelines, as previously reported [42]. |

Note that full information on the approval of the study protocol must also be provided in the manuscript.

# Plants

| Seed stocks | No plants used |
|---|---|
| Novel plant genotypes | No plants used |
| Authentication | No plants used |

# Magnetic resonance imaging

## Experimental design

| Design type | Resting-state fMRI |
|---|---|
| Design specifications | Sevoflurane dataset: five scanning sessions: awake, 2 vol%, 3 vol% burst-suppression, and recovery.<br>propofol dataset: four scanning sessions: awake, mild sedation (not considered here), deep anaesthesia, and recovery. |
| Behavioral performance measures | Loss of behavioural responsiveness was used to determine depth of anaesthesia.<br>Sevoflurane dataset: loss of consciousness was judged by the loss of responsiveness (LOR) to the repeatedly spoken command "squeeze my hand" two consecutive times.<br>For the propofol dataset, failure to perform two computerised tasks (a computerised auditory target-detection task and a memory test of verbal recall) was used to evaluate the level of wakefulness in the anaesthesia condition independently of the assessors, who also evaluated participants' level of behavioural responsiveness based on the Ramsay scale. |

## Acquisition

| Imaging type(s) | Functional and anatomical |
|---|---|
| Field strength | 3T for all datasets |
| Sequence & imaging parameters | Sevoflurane dataset: data were collected using a gradient echo planar imaging sequence (echo time = 30 ms, repetition time (TR) = 1.838 s, flip angle = 75 deg, field of view = 220 × 220 mm2, matrix = 72 × 72, 32 slices, slice thickness = 3 mm, and 1 mm interslice gap; 700-s acquisition time, resulting in 350 functional volumes). The anatomical scan was acquired before the functional scan using a T1-weighted MPRAGE sequence with 240 × 240 × 170 voxels (1 × 1 × 1 mm voxel size) covering the whole brain.<br><br>propofol dataset: MRI scanning was performed using a 3-Tesla Siemens Tim Trio scanner (32-channel coil), and 256 functional volumes (echo-planar images, EPI) were collected from each participant, with the following parameters: slices = 33, with 25% inter-slice gap; resolution = 3mm isotropic; TR = 2000ms; TE = 30ms; flip angle = 75 degrees; matrix size = 64x64. The order of acquisition was interleaved, bottom-up. Anatomical scanning was also performed, |

acquiring a high-resolution T1- weighted volume (32-channel coil, 1mm isotropic voxel size) with a 3D MPRAGE sequence, using the following parameters: TA = 5min, TE = 4.25ms, 240x256 matrix size, 9 degrees flip angle.

Area of acquisition | Whole-brain

Diffusion MRI ☒ Used ☐ Not used

Parameters | The dMRI data were from the HCP dataset. The spatial resolution was 1.25 mm isotropic. TR=5500ms, TE=89.50ms. The b-values were 1000, 2000, and 3000 s/mm2. The total number of diffusion sampling directions was 90, 90, and 90 for each of the shells in addition to 6 b0 images.

## Preprocessing

Preprocessing software | Preprocessing of the functional MRI data for both datasets followed the same standard workflow as in our previous studies, and was implemented in the CONN toolbox (http://www.nitrc.org/projects/conn), version 17f [81].

Normalization | Direct normalisation to MNI space (nonlinear) using the segmented grey matter image from each volunteer's high-resolution T1-weighted image, together with an a priori grey matter template.

Normalization template | MNI-152 volumetric template, 2x2x2mm isotropic resolution.

Noise and artifact removal | Denoising followed the anatomical CompCor (aCompCor) method of removing cardiac and motion artifacts, by regressing out of each individual's functional data the first 5 principal components corresponding to white matter signal, and the first 5 components corresponding to cerebrospinal fluid signal, as well as six subject-specific realignment parameters (three translations and three rotations) and their first- order temporal derivatives, and nuisance regressors identified by the software ART 82. The subject-specific denoised BOLD signal time-series were linearly detrended and band-pass filtered between 0.008 and 0.09 Hz to eliminate both low-frequency drift effects and high-frequency noise.

Volume censoring | the artifact rejection tool (ART), implemented in the CONN toolbox, was used to identify and regress out outlying volumes, as part of the CompCor denoising procedure described above. The default CONN settings of 5 global signal z-values and 0.9mm were used.

## Statistical modeling & inference

Model type and settings | We used correlation against an autocorrelation-preserving null distribution to test the spatial association between regional identifiability and canonical maps of interest. We used non-parametric permutation t-tests (repeated-measures), with 10,000 permutations to compare identifiability, cognitive matching , and similarity to the macaque regional connectivity between levels of anaesthesia,

Effect(s) tested | We tested whether anaesthesia influences identifiability; we tested whether regional identifiability is spatially associated with canonical maps of interest; we tested whether the anaesthetised FC is more similar to the macaque FC than awake FC; and we tested whether meta-analytic matching is affected by anaesthesia.

Specify type of analysis: ☒ Whole brain ☐ ROI-based ☐ Both

Statistic type for inference

(See Eklund et al. 2016) | t-tests: permutation-based with 10,000 permutations. Correlation: Spearman's rank-based correlation coefficient, with p-value assessed against a null distribution of spatial autocorrelation-preserving maps.

Correction | Anaesthesia conditions were compared separately against wakefulness and against recovery, and the false positive rate against multiple comparisons was controlled using the false discovery rate (FDR) correction [10], separately for these two cases.

## Models & analysis

n/a | Involved in the study
☐ ☒ Functional and/or effective connectivity
☒ ☐ Graph analysis
☐ ☒ Multivariate modeling or predictive analysis

Functional and/or effective connectivity | Functional connectivity was obtained as the Pearson correlation between timeseries.

Multivariate modeling and predictive analysis | To consider all regional correlates together and evaluate their respective contributions, we performed a dominance analysis with all four canonical brain maps as predictors, and the regional map of anaesthetic-induced ICC changes as target.

