## [Peer Review File · Nature Human Behaviour]

General anaesthesia reduces the uniqueness of brain functional connectivity across individuals and across species

Corresponding Author: Dr Andrea Luppi

Version 0:

Decision Letter:

10th May 2024

Dear Dr Luppi,

Thank you once again for your manuscript, entitled "General anaesthesia reduces the uniqueness of brain functional connectivity across individuals and across species," and for your patience during the peer review process.

Your manuscript has now been evaluated by 3 reviewers, whose comments are included at the end of this letter. In the light of their advice, I regret that we cannot offer to publish your manuscript in Nature Human Behaviour.

While the reviewers find your work of some interest, they raise concerns about the strength and specificity of the novel conclusions that can be drawn at this stage as well as the robustness of the data. We feel that these reservations are sufficiently important as to preclude publication of this work in Nature Human Behaviour.

I am sorry that we cannot be more positive on this occasion but hope that you will find our reviewers' comments helpful when preparing your paper for submission elsewhere.

Sincerely,

Giacomo Ariani
Editor
Nature Human Behaviour

Reviewer expertise:

Reviewer #1: Human consciousness, fMRI brain fingerprinting, NHPs research

Reviewer #2: Human consciousness, fMRI brain fingerprinting

Reviewer #3: Human consciousness, fMRI brain fingerprinting

Reviewers' Comments:

Reviewer #1:

Remarks to the Author:

The manuscript investigated how anesthesia affects the uniqueness of functional connectivity of human brains. It finds that anesthesia reduces individual identifiability and makes the human brain more similar to macaques, especially at deeper anesthesia levels. The manuscript is well-written and clear. However, several points need to be addressed including the methods and underlying mechanisms that account for the observed reduction in uniqueness between awake and anesthetized states. Below are detailed comments.

- The study demonstrates that individual brain distinctiveness diminishes when comparing within- and between-individual similarity from awake to anesthetized states. However, the analysis does not fully address the question about individual uniqueness within an unconscious state. Specifically, what would be the similarity and identification comparing both within- and between-individual under anesthesia? It's possible that the identifiability of individuals is preserved in an unconscious state, albeit altered from the awake state. In previous fingerprinting studies (e.g. Finn et al. 2015), a significant drop in individual identification across different awake states (tasks) was observed, from ~0.9 to ~0.6, highlighting state dependency of the individual uniqueness. Such findings imply that the observed differences in uniqueness are primarily state-dependent.

- In the current study, the authors compared the individual identifiability between sevoflurane at 2% and 3%, which showed higher identifiability than comparisons across awake and sevoflurane state, although not as high as within the awake state. This finding supports the above idea.
- A related point, the quantity of the time in such comparisons is also crucial. Previous studies indicate that over 20-30min of resting-state data are necessary to accurately characterize an individual FC profile. This suggests that more data may be needed when comparing FC across different states to ensure reliable identification of self. Any comparisons or comments?
- The use of Schaefer parcellations, which were generated based on awake human data, may not optimally reflect the state of anesthetized brain data.
- What's the underlying mechanisms for the reduced uniqueness between awake and anesthetized states? Could the authors elaborate on the physiological or neurobiological processes that underlie this change?
- The findings indicate that human FC under anesthesia, particularly during the sevoflurane burst-suppression state, showed the highest similarity to the macaque awake FC. It would be informative to also compare human FC with anesthetized macaque FC. If the similarity between anesthetized humans and anesthetized macaque is lower than that between anesthetized humans and awake macaques, it would further support the conclusion. However, if the similarity is greater, it may challenge the interpretation that anesthesia makes human FC more like an awake macaque. This additional comparison could provide insights into the specific effects of anesthesia on interspecies FC similarity.
- The result shows a positive correlation in edge-wise FC between humans and macaques, yet this relationship becomes negative when FC is averaged into regional strength (Fig S7). Can the authors provide an explanation to understand this result? And how does the regional FC strength vary across regions (Fig 4a)? How did the authors rule out the impact of fMRI noise (e.g. signal inhomogeneity, SNR, etc.)?
- Relatedly, since the cross-species comparison was conducted using regional strength rather than edge-wise FC, and given the opposing relationships observed (positive for edge-wise FC but negative for regional averages), are the cross-species comparison replicable for edge-wise FC (Fig 4b)?
- How do we understand the result that on one hand, human brains when unconscious are more similar to the macaque awake brains, yet simultaneously, they become less similar to each other? This seems paradoxical. Is this also true for edge-wise FC?
- Could the authors provide visualizations of the matched parcellations between the two species? Additionally, how well these cross-species parcellations capture homologous regions between macaque and human brains?
- The authors stated that "Correlation analysis cannot identify causal determinants" and further performed the dominance analysis. However, dominance analysis also does not infer causality.
- There are >900 cognitive concepts listed in the Cognitive Atlas. What criteria or processes were used here to select 123 terms in the final analysis?

Minor:

- Fig 2a, ICC should be in a range of 0-1. It is noted that the ICC matrix from awake to sevoflurane includes negative (blue) values?
 - Many of the figures include the color maps without absolute values on the colorbar. Can the authors provide the absolute values although the comparisons between metrics were relative across regions?
- Fig2b The authors have merged the edge identifiability for the within-unimodal and uni-transmodal edges. What was the rationale behind combining these two types of edges?
- Remove the duplicate "with" in "Replication of cognitive matching with with BrainMap"

Reviewer #2:

Remarks to the Author:

Luppi et al. utilized fMRI while subjects were under general anesthesia to explore its effects on human brain uniqueness. Their findings indicate that anesthesia reduces the self-similarity and individual distinctiveness in brain function, particularly in areas linked to sensory-association. This phenomenon was observed consistently across various anesthetics, was reversible after the subjects recovered from anesthesia, and led to a closer resemblance between the functional connectivity patterns of human and macaque brains. The study suggests that anesthesia not only blurs the unique features between individual human brains but also makes the human brain functionally more similar to that of other primates, especially in regions that are more developed in humans.

My detailed comments are provided below.

1. While the author suggests that sevoflurane anesthesia diminishes the identifiability of individual connectomes, the comparisons used—'recovery vs awake' and 'sevo vs awake'—primarily demonstrate deviations from the awake state rather than a definitive loss of identifiability. To more effectively support this conclusion, it would be beneficial to include a 'sevo vs sevo' comparison alongside the 'awake vs awake' scenario. Additionally, the analysis concerning the ICC could be expanded to compare 'sevo vs sevo' with 'awake vs awake.' It's important to note, however, that previous research suggests that under anesthesia, brain functional networks might align more closely with structural connections, which remain highly individualized. Therefore, ICC values for 'sevo vs sevo' under anesthesia might not necessarily be low, potentially indicating maintained individual distinctiveness despite the anesthetic state.

2. Figure 4 posits a closer resemblance between human and macaque cortical connectivity under anesthesia than in wakefulness. Yet, the methods for this comparison remain unspecified. Moreover, the negative correlation reported for awake connectivity diverges from prior research suggesting homology in these networks. It would be beneficial for the author to introduce additional awake-state data to contextualize and clarify this unexpected correlation.

3. The author, through sophisticated analyses, claims that anesthesia reduces the uniqueness of brain functional connectivity among individuals and across species, primarily using a resting-state paradigm. The main finding is that the intrasubject correlation for 'sevo vs awake' is significantly lower than that for 'recovery vs awake.' However, this result is not necessarily specific to the state of unconsciousness; previous study showed that the awake resting state possesses the highest fingerprinting capacity, with task states generally exhibiting weaker correlation than the resting state (Emily et al, 2015). The author should discuss or conduct additional experiments to elucidate the specificity of these results.

Reviewer #3:

Remarks to the Author:

Luppi et al., showed that anesthesia reduces uniqueness of functional connectivity (FC) across individuals and species. The "fingerprint" of individual FC in resting eye-closed state was diminished under anesthesia and recovered along with return of consciousness. The authors further found that the greatest decrease in identifiability was in transmodal cortex. Furthermore, similarity between fMRI BOLD signal and cognitive map was substantially reduced in anesthesia.

I found the question the authors ask, i.e., does anesthesia suppress uniqueness of functional connectivity?, is interesting and the corresponding data were nicely visualized as boxplot, distributions and individual points, which may help readers understand the manuscript.

There are a few questions and comments regarding manuscripts:

1)

To my understanding, uniqueness requires both diversity across individuals and persistency over time. For example, if FC configurations are all the same, identifiability will be low even though FC persists over time. On the other hand, although FC configurations are diverse enough across individuals, identifiability will be low if FC varies much over time. Considering this, it is unclear how the identifiability was reduced by anesthetics. Is it because of (a) anesthesia makes all FC configurations very similar each other (reduced diversity)? Or (b) FCs are still diverse but their persistency is decreased by anesthesia?

2)

The Fig S16 suggests that FC in anesthesia is more diverse and probably less structured ((b) in the above), because the off-diagonal components of the matrix in Fig S16 b and g is low. That is, there may be large interindividual variation in FC configuration under anesthesia. If so, I have following questions.

In EEG/MEG studies, it has been well known that anesthetized brain has stereotypic FC configuration (Cimenser et al., 2011; Purdon et al., 2013; Blain-Moraes et al., 2017). For example, anesthesia results in a stereotypical FC pattern, so-called "anteriorization". Coherence and power distribution becomes very structured, and moreover, it is consistent across individuals. How would you reconcile your result (Fig. S16 b and g) with this seemingly opposite result of EEG/MEG?

3)

If anesthesia does not make FC configuration stereotypical (Fig. S16 b and g), how do FCs of most individuals under anesthesia become highly correlated with macaque's FC?

4)

I highly recommend the authors to add an additional figure (a subplot or in supplementary), visualizing the FC configurations in the 2d or 3d space. For example, scatter or schematic plot of individual FCs in low-dimensional 2d or 3d space (after dimensionality reduction by t-SNE or UMAP). By doing so, the readers would be able to understand how anesthesia reconfigures FC structure: e.g., does FC become stereotypical (all individual FC under anesthesia are clustered or distant each other?) by anesthetics?, Why and how identifiability was reduced?, visualization of distances between FC of consciousness vs FC of anesthesia vs FC of macaque, etc.

5)

Regarding page 9 first paragraph (FC vs. SC and reference 63~65).

One study reported that FC of anesthetized brain was dissimilar to structural connectivity (SC), compared to that of conscious brain (Tagliazucchi et al., 2016, "Large-scale signatures of unconsciousness are consistent with a departure from critical dynamics").

Why do you think that there are seemingly opposite results?

6)

To my understanding, in Fig. 1d, "self-self minus self-others" represents either $\langle a_{ii} - a_{ij} (j \neq i) \rangle$ or $\langle a_{ij} - a_{ji} (j \neq i) \rangle$. But it is not clear it is the former or latter. Could you describe in more details about the definition of y-axis in the Fig. 1d?

7)

Typo:

Two "with" in the sub-title "Replication of cognitive matching with with Brain-Map" in Results section

Following suitable revisions, you may want to consider transferring your manuscript. Although we cannot offer to publish your

manuscript, I suggest that you consider Communications Psychology as a suitable venue for this work. To transfer your manuscript, please use our manuscript transfer portal. You will not have to re-supply manuscript metadata and files, unless you wish to make modifications. For more information, please see our [manuscript transfer FAQ](http://www.nature.com/authors/author_resources/transfer_manuscripts.html?WT.mc_id=EMI_NPG_1511_AUTHORTRANSF&WT.ec_id=AUTHOR) page.

Version 1:

Decision Letter:

Dear Dr Luppi,

Thank you for your correspondence asking us to reconsider our decision on your Article, "General anaesthesia reduces the uniqueness of brain functional connectivity across individuals and across species". After careful consideration we have decided that we would be willing to consider a revised version of your manuscript.

Along with your revised manuscript, you should also submit a separate point-by-point response to all of the concerns raised by the referees, in each case describing what changes have been made to the manuscript or, alternatively, if no action has been taken, providing a compelling argument for why that is the case. If we feel that a substantial attempt has been made to address the referees' comments, this response will be sent back to the referees - along with the revised manuscript - so that they can judge whether their concerns have been addressed satisfactorily or otherwise.

I should stress, however, that we would be reluctant to trouble our referees again unless we thought that their comments had been addressed in full.

- ensure it complies with our format requirements as set out in our [Guide to Authors](http://www.nature.com/nathumbehav/info/gta).
- state in a cover note the length of the text, methods and figure legends; the number of references and the number of display items.

Please ensure that all correspondence is marked with your Nature Human Behaviour reference number in the subject line.

Please use the following link to submit your revised manuscript:

Link Redacted

We hope to receive your revised paper within four weeks. If you cannot send it within this time, please let us know so that we can close your file. In this event, we will still be happy to reconsider your paper at a later date so long as nothing similar has been accepted for publication at Nature Human Behaviour or published elsewhere in the meantime. Should you miss the four-week deadline and your paper is eventually published, the received date will be that of the revised, not the original, version.

I look forward to hearing from you soon.

Best regards,

Giacomo Ariani
Editor
Nature Human Behaviour

Version 2:

Decision Letter:

11th November 2024

Dear Dr Luppi,

Thank you once again for your revised manuscript, entitled "General anaesthesia reduces the uniqueness of brain functional connectivity across individuals and across species," and for your patience during the re-review process.

Your manuscript has now been evaluated by Reviewers #1 and #2 from the original round of review. We invited a third reviewer but they declined to comment on the revised manuscript. All reviewer feedback is included at the end of this letter. Although the reviewers found your manuscript to have improved during revision, Reviewer #1 maintains some outstanding concerns. We remain very interested in the possibility of publishing your study in Nature Human Behaviour, but would like to consider your response to these outstanding concerns in the form of a revised manuscript before we make a decision on publication.

In sum, we invite you to revise your manuscript taking into account all reviewer and editor comments. We are committed to providing a fair and constructive peer-review process. Do not hesitate to contact us if there are specific requests from the reviewers that you believe are technically impossible or unlikely to yield a meaningful outcome.

We hope to receive your revised manuscript within 4-8 weeks. I would be grateful if you could contact us as soon as possible if you foresee difficulties with meeting this target resubmission date.

- Include a "Response to the editors and reviewers" document detailing, point-by-point, how you addressed each editor and referee comment. If no action was taken to address a point, you must provide a compelling argument. This response will be used by the editors and reviewers to evaluate your revision.
- Highlight all changes made to your manuscript or provide us with a version that tracks changes.

Link Redacted

We look forward to seeing the revised manuscript and thank you for the opportunity to review your work. Please do not hesitate to contact me if you have any questions or would like to discuss these revisions further.

Sincerely,

Giacomo Ariani, PhD
Senior Editor
Nature Human Behaviour

Reviewer expertise:

Reviewer #1: Human consciousness, fMRI brain fingerprinting, NHPs research

Reviewer #2: Human consciousness, fMRI brain fingerprinting

REVIEWER COMMENTS:

Reviewer #1 (Remarks to the Author):

This manuscript examines the effects of general anesthesia agents like sevoflurane and propofol on brain connectivity. Under anesthesia, individual distinctions in human brain connectivity become less pronounced, and there is reduced involvement of cognitive function. The authors have included additional analyses that robustly support this finding, which has made the conclusions clear and compelling.

Only one aspect of the results regarding the analysis comparing the similarity or distance between human and macaque connectivity is somewhat unclear. The authors attempted to conclude that anesthesia shifts human functional connectivity closer to that of macaques. However, in figure 4b, the distance between the macaque functional connectivity and the awake human FC (brown and grey bars) appears to be greater than any distance within human FC states. This observation seems to contradict the authors' conclusion.

In figure 4a, if I understand correctly, a joint PCA was applied by concatenating the human and macaque data (if so, please clarify this in the methods section). It appears that PC2 captures the differences in human states, while PC1 captures state differences in macaques. Given that PCs are orthogonal, does this imply that the distinction between awake and anesthetized states differs between humans and macaques? Could the authors elaborate on how we should interpret this species-state interaction?

I noticed a comment regarding the ICC range. In theory, intra-class correlation is defined as the proportion of variance that is attributable to differences between classes. In other words, it is the ratio of between-subject variance to the total variance. Since both the numerator and denominator in this calculation are variances, which are inherently non-negative, the ICC should be non-negative by definition. While in rare cases in practice, the estimation of ICC can fall slightly below 0. Usually, a negative ICC generally suggests model or estimation anomalies. Therefore, I suspect there might be some misunderstanding.

Reviewer #2 (Remarks to the Author):

Many thanks for the revisions or clarifications regarding all my concerns. I have no further comments.

Version 3:

Decision Letter:

Our ref: NATHUMBEHAV-24031050C

11th December 2024

Dear Dr Luppi,

Thank you for submitting your revised manuscript "General anaesthesia reduces the uniqueness of brain functional connectivity across individuals and across species" (NATHUMBEHAV-24031050C). It has now been seen by one of the original referees and their comments are below. As you can see, the reviewer finds that the paper has improved in revision. We will therefore be happy in principle to publish it in Nature Human Behaviour, pending minor revisions to comply with our editorial and formatting guidelines.

We are now performing detailed checks on your paper and will send you a checklist detailing our editorial and formatting requirements within two weeks. Please do not upload the final materials and make any revisions until you receive this additional information from us.

Sincerely,

Giacomo Ariani, PhD
Senior Editor
Nature Human Behaviour

Reviewer #1 (Remarks to the Author):

The authors have thoroughly addressed all the comments and I have no further comments.

Version 4:

Decision Letter:

Dear Dr Luppi,

We are pleased to inform you that your Article "General anaesthesia reduces the uniqueness of brain functional connectivity across individuals and across species", has now been accepted for publication in Nature Human Behaviour.

Authors may need to take specific actions to achieve <https://www.springernature.com/gp/open-research/funding/policy-compliance-faqs> compliance with funder and institutional open access mandates. If your research is supported by a funder that requires immediate open access (e.g. according to <https://www.springernature.com/gp/open-research/plan-s-compliance>) then you should select the gold OA route, and we will direct you to the compliant route where possible. For authors selecting the subscription publication route, the journal's standard licensing terms will need to be accepted, including <https://www.springernature.com/gp/open-research/policies/journal-policies> self-archiving policies. Those licensing terms will supersede any other terms that the author or any third party may assert apply to any version of the manuscript.

Acceptance of your manuscript is conditional on all authors' agreement with our publication policies (see

<http://www.nature.com/nathumbehav/info/gta>). In particular your manuscript must not be published elsewhere and there must be no announcement of the work to any media outlet until the publication date (the day on which it is uploaded onto our web site).

To assist our authors in disseminating their research to the broader community, our ShareIt initiative provides you with a unique shareable link that will allow anyone (with or without a subscription) to read the published article. Recipients of the link with a subscription will also be able to download and print the PDF.

With best regards,

Giacomo Ariani, PhD
Senior Editor
Nature Human Behaviour

P.S. Click on the following link if you would like to recommend Nature Human Behaviour to your librarian
<http://www.nature.com/subscriptions/recommend.html#forms>

** Visit the Springer Nature Editorial and Publishing website at http://editorial-jobs.springernature.com?utm_source=ejp_NHumB_email&utm_medium=ejp_NHumB_email&utm_campaign=ejp_NHumB for more information about our career opportunities. If you have any questions please click [here](mailto:editorial.publishing.jobs@springernature.com).

From: Bratislav Mistic <bratislav.mistic@mcgill.ca>
Sent: Wednesday, May 15, 2024 7:01 PM
To: Giacomo Ariani <giacomo.ariani@springernature.com>
Cc: A. Luppi <al857@cam.ac.uk>
Subject: Re: Decision on Nature Human Behaviour manuscript NATHUMBEHAV-24031050

Diese Nachricht stammt von einem externen Absender

Diese Nachricht kam von ausserhalb Ihrer Organisation.

Verdächtigen Inhalt melden

Dear Dr. Ariani,

Thank you for conveying the peer reviewers' feedback for our manuscript "General anaesthesia reduces the uniqueness of brain functional connectivity across individuals and across species" (NATHUMBEHAV-24031050).

Although your final decision was rejection, we are reaching out because we are hoping that you may be willing to allow us to revise our manuscript for further consideration at *Nature Human Behaviour*. Briefly, we believe that all the peer reviewers' comments are easily addressable: indeed, many are already addressed in our manuscript. For example:

1. Both Reviewer #1 and Reviewer #2 ask for comparisons under anaesthesia (R1: "what would be the similarity and identification comparing both within- and between-individual under anesthesia"; R2: "it would be beneficial to include a 'sevo vs sevo' comparison alongside the 'awake vs awake' scenario"). In fact,

this comparison is already present in our manuscript: Figures S16a-j and Figure S17a-c. We would be happy to draw more attention to these results in a revised manuscript.

2. R1 asks whether the cross-species comparison is replicable for edge-wise FC. In our manuscript, we already show evidence that this is indeed the case, in both sevoflurane and propofol datasets (Figure S8 and Figure S15).
3. R2 asks how the cross-species comparison is performed; this is already described visually in Figure 4a and its caption, and further explored in Figure S7. R1 asks how we chose the 123 NeuroSynth terms. This is extensively described in the dedicated section of the Methods (Meta-analytic cognitive matching from NeuroSynth); our BrainMap analysis (Figure S18) further shows that results are not dependent on the specific selection. We would be happy to make these points even clearer upon revision.
4. R1 is concerned that results obtained with the Schaefer atlas may be biased, because this atlas was derived from functional data of awake individuals. We had already taken into account this concern: Figure S19 shows that our results can be replicated using the Desikan-Killiany parcellation, which is an anatomical atlas.
5. R1 is concerned that 20-30min of resting-state data may be necessary to accurately characterize an individual FC profile. We can easily prove that this concern is unfounded: we already reach > 90% success rate at identifiability for awake individuals. This is consistent with our own previous published work (Van De Ville et al., 2021 Science Advances), which showed that ~5min of rs-fMRI is sufficient (here, ~10min are available).
6. R3 asks whether anaesthesia makes all FC configurations very similar to each other (reduced diversity), or FCs are still diverse but their persistency is decreased by anaesthesia. This question is addressed in Figure S17: persistency (self-self correlation) is reduced, and diversity is increased (reduced self-other correlation), but the reduction in persistency is greater than the increase in diversity, leading to overall less identifiability. We would gladly elaborate more on this point in a revised manuscript.
7. R3 and R1 both ask how human brains when unconscious can become more

similar to the macaque awake brain, yet simultaneously, become less similar to each other. We acknowledge that this can seem paradoxical, and we can provide results and code for a simulation showing how this phenomenon may occur.

We are confident that all remaining points raised by the reviewers can be addressed in a straightforward manner, and we would be tremendously grateful for the opportunity to submit a revised manuscript for your evaluation.

Sincerely,

Bratislav Mistic & Andrea Luppi

—

Bratislav Mistic, PhD
Montreal Neurological Institute
<https://netneurolab.github.io/>

Response to reviewer comments for manuscript NATHUMBEHAV-24031050A-Z

We thank the Reviewers for their feedback on our manuscript. Please find below our point-to-point responses to each comment. For ease of reading, reviewers' feedback is provided in **bold**; and quoted passages from the revised manuscript are shown as indented text.

Reviewer #1:

Remarks to the Author:

The manuscript investigated how anesthesia affects the uniqueness of functional connectivity of human brains. It finds that anesthesia reduces individual identifiability and makes the human brain more similar to macaques, especially at deeper anesthesia levels. The manuscript is well-written and clear. However, several points need to be addressed including the methods and underlying mechanisms that account for the observed reduction in uniqueness between awake and anesthetized states. Below are detailed comments.

- The study demonstrates that individual brain distinctiveness diminishes when comparing within- and between-individual similarity from awake to anesthetized states. However, the analysis does not fully address the question about individual uniqueness within an unconscious state. Specifically, what would be the similarity and identification comparing both within- and between-individual under anesthesia? It's possible that the identifiability of individuals is preserved in an unconscious state, albeit altered from the awake state. In previous fingerprinting studies (e.g. Finn et al. 2015), a significant drop in individual identification across different awake states (tasks) was observed, from ~0.9 to ~0.6, highlighting state dependency of the individual uniqueness. Such findings imply that the observed differences in uniqueness are primarily state-dependent.

The main message of our article is that the identifiability of the functional connectome is dependent on one's state of consciousness. Certainly, many other factors can also influence identifiability of the functional connectome. Task performance is one of these factors. Indeed, in our Results we explicitly discussed the effect of tasks, addressing evidence that tasks can both increase or decrease identifiability vis-a-vis rest, depending on how tightly controlled they are:

"We find that as anaesthesia deepens, both self-similarity and the difference between self-self and self-other correlations (i.e., identifiability) are progressively reduced, with the two distributions becoming increasingly overlapping (original Fig. S17). This pattern is the reverse of what was recently found by [44], who showed that tightly controlled cognitive tasks increase self-other similarity, but increase self-self similarity even more, thereby resulting in an overall increase in identifiability"

The reviewer is concerned that “the identifiability of individuals could be preserved in an unconscious state, albeit altered from the awake state”. This is a valid concern, and we have taken the following steps to address it. In Fig. S20 (originally Fig. S16) we show that identifiability between two unconscious conditions (sevoflurane anaesthesia at vol 3% and sevoflurane anaesthesia at burst-suppression) is significantly lower than between two conscious conditions (pre-anaesthetic baseline, and post-anaesthetic recovery). As shown in Fig. S22 (originally S17), this reduction occurs because as anaesthesia deepens, self-other similarity is reduced, but self-self similarity is reduced even more, leading to an overall reduction in identifiability (which is computed as the difference between self-self similarity and self-other similarity).

Figure S20. Replication of identifiability results using within-state comparisons | (a) Identifiability matrix between wakefulness and post-anaesthetic recovery. (b) Identifiability matrix between vol 2% sevoflurane and vol 3% sevoflurane anaesthesia. Entries along the diagonal, represent self-self similarity (correlation of FC patterns), whereas off-diagonal entries represent self-other similarity. (c) Self-self similarity is significantly higher between two conscious states, than between vol 2% and vol 3% sevoflurane. (d) The difference between self-self correlation and mean self-other correlation (differential identifiability) is significantly higher between two conscious states, than between vol 2% and vol 3% sevoflurane. (e) The regional distribution of contributions to identifiability (change in intra-class correlation coefficient) is plotted on the cortical surface. (f) Identifiability matrix between wakefulness and post-anaesthetic recovery. (g) Identifiability matrix between vol 3% and burst-suppression level of sevoflurane anaesthesia. (h) Self-self similarity is significantly higher between two conscious states, than between vol 3% and burst-suppression level of sevoflurane. (i) The difference between self-self correlation and mean self-other correlation (differential identifiability) is significantly higher between two conscious states, than between vol 3% and burst-suppression level of sevoflurane. (j) The regional distribution of contributions to identifiability (change in intra-class correlation coefficient) is plotted on the cortical surface. *, $p < 0.05$; ***, $p < 0.001$.

Figure S22. Distributions of self-self and self-other correlations for different pairs of conditions | (a) Two conscious states: awake versus post-anaesthetic recovery. (b) Vol 2% versus vol 3% sevoflurane. (c) Vol 3% versus burst-suppression levels of sevoflurane.

Finally, in the new Fig. S21 we show that regional changes in identifiability also replicate when two conscious conditions are contrasted with two unconscious conditions (sevoflurane anaesthesia at vol 3% and sevoflurane anaesthesia at burst-suppression, or vol 2% versus 3%).

Figure S21. Replication of edge-wise and regional identifiability results using within-state comparisons | (a) Matrix of edge-level difference in intra-class correlation coefficient between awake-recovery and vol 2% - vol 3% sevoflurane anaesthesia. (b) Regional distribution of propofol-induced loss of ICC (awake-recovery versus 2%-3% sevoflurane). (c) The sevoflurane-induced regional loss of ICC (awake-recovery versus 2%-3% sevoflurane) is significantly spatially aligned with the archetypal sensory-association axis of cortical organisation; the regional distribution of inter-individual variability of functional connectivity; the regional cortical expansion between macaque and human; and the regional expression of human-accelerated genes pertaining to brain function and development (“HAR-brain genes”). (d) Matrix of edge-level difference in intra-class correlation coefficient between awake-recovery and vol 3% - burst-suppression levels of sevoflurane anaesthesia. (e) Regional distribution of propofol-induced loss of ICC (awake-recovery versus 3% sevoflurane - burst-suppression). (f) The sevoflurane-induced regional loss of ICC (awake-recovery versus 3% sevoflurane - burst-suppression) is significantly spatially aligned with the archetypal sensory-association axis of cortical organisation; the regional distribution of inter-individual variability of functional connectivity; the regional cortical expansion between macaque

and human; and the regional expression of human-accelerated genes pertaining to brain function and development (“HAR-brain genes”). For all scatter-plots, N = 200 regions from the Schaefer atlas.

- In the current study, the authors compared the individual identifiability between sevoflurane at 2% and 3%, which showed higher identifiability than comparisons across awake and sevoflurane state, although not as high as within the awake state. This finding supports the above idea.

Please see our response to the previous comment. Figures S20-S21-S22 (which expand on the original Figures S16 and S17) provide a comparison of two awake states (awake and recovery) against two anaesthetised states with different levels of sevoflurane. This is reported in the main *Results* section, subsection *Within-state identifiability*:

“To empirically demonstrate that our results are not just due to comparing within-state correlations against between-state correlations (with ‘state’ in this context referring to wakefulness or anaesthesia), we take advantage of the fact that our sevoflurane data include multiple scans obtained under anaesthesia. This allows us to compare two conscious scans (baseline and recovery) and two anaesthetised scans: either vol 2% and 3% sevoflurane, or vol 3% sevoflurane and burst-suppression.

When comparing awake-recovery similarity against the similarity between vol 2% and 3% sevoflurane, or between vol 3% sevoflurane and burst-suppression, we find exactly the same pattern of results as in our main analysis (Fig. S20). Self-self similarity is significantly diminished, not only between wakefulness and anaesthesia (as we previously showed), but also between two anaesthetised scans. Likewise, identifiability is also diminished under anaesthesia, delineating a clear unimodal-transmodal cortical pattern (Fig. S20).”

- A related point, the quantity of the time in such comparisons is also crucial. Previous studies indicate that over 20-30min of resting-state data are necessary to accurately characterize an individual FC profile. This suggests that more data may be needed when comparing FC across different states to ensure reliable identification of self. Any comparisons or comments?

We appreciate this concern. However, we are confident that we can put this concern to rest, because we already reach > 90% success rate at identifiability for awake individuals (93%, corresponding to all but one participants, for the main dataset, and 100% for the replication dataset; this is now reported in the corresponding figures’ legends).

This observation in our data is fully consistent with Finn et al. (2015 *Nature Neuroscience*) who show that >90% of successful identification rate can be achieved with 300 time-points; and also with Van De Ville et al. (2021 *Science Advances*), who showed that in fact, just over 1 minute (72s) of rs-fMRI is sufficient to achieve >90% success rate at identifiability in resting

state HCP data; (our main analysis comprises ~10 min of scan time for each of the 5 conditions).

So our main result that identifiability is low under anaesthesia cannot simply be attributed to scan duration, since the same scan duration is perfectly sufficient for excellent identifiability in the same individuals, when they are conscious.

To further demonstrate this point, in our revised manuscript we have added a new analysis (see Results and Figure S12), where we perform fingerprinting against FC obtained from concatenating all the anaesthetised scans (vol 2%, vol 3% and burst-suppression, amounting to ~30min of rs-fMRI), thereby 'stacking the deck' in favour of anaesthesia producing better fingerprinting than recovery by tripling the number of timepoints used for FC estimation. Nevertheless, we still observe reduced self-self similarity and identifiability of FC under anaesthesia.

We have added the following text in the Results:

“Robustness to scan duration. In addition to showing that anaesthesia reduces identifiability between different anaesthetised scans, we also demonstrate that anaesthetic-induced differences in self-self similarity and identifiability are not due to limitations of our scan duration. First, each of our scans had the same duration (see Methods), which means that we do not need to be concerned about differences in scan duration as a potential confound. Second, this duration (approximately 10 minutes) is clearly more than sufficient to enable excellent brain fingerprinting, with all but one individual (93%) being correctly identified when conscious, in our awake-recovery data. This results is fully consistent with Van de Ville (2021), who reported that just over one minute of rs-fMRI is sufficient to achieve over 90% of successful identification from brain fingerprinting. Third, we show that even when anaesthetised FC is obtained from combining all the BOLD signals acquired across the three sevoflurane conditions (vol 2%, vol 3%, and burst-suppression, which were temporally concatenated prior to obtaining correlations between regions), nevertheless identifiability and self-self similarity are still reduced under anaesthesia (Fig. S12). Thus, in accordance with the brain fingerprinting literature, our results of anaesthetic-induced differences in self-self similarity and identifiability cannot be attributed to scan duration being insufficient for fingerprinting. On the contrary, these results persist even after artificially 'stacking the deck' in favour of anaesthesia by tripling the number of timepoints used for FC estimation, demonstrating their robustness.”

Figure S12. Replication of identifiability results when all anaesthesia levels are concatenated | (a) Identifiability matrix between wakefulness and post-anaesthetic recovery. (b) Identifiability matrix between wakefulness and the FC obtained after concatenating the BOLD timeseries from all three anaesthetic levels (vol 2%, vol 3%, and burst-suppression sevoflurane anaesthesia). Entries along the diagonal, represent self-self similarity (correlation of FC patterns), whereas off-diagonal entries represent self-other similarity. (c) Self-self similarity is significantly higher between two conscious states, than between wakefulness and the combined anaesthesia data. (d) The difference between self-self correlation and mean self-other correlation (differential identifiability) is significantly higher between two conscious states, than between wakefulness and the combined anaesthesia data. N=15 human volunteers.

- The use of Schaefer parcellations, which were generated based on awake human data, may not optimally reflect the state of anesthetized brain data.

We used the Schaefer parcellation for our main analyses because several publications have shown that its use is optimal on multiple desirable dimensions, including being representative of other atlases (Luppi & Stamatakis, 2021 *Network Neuroscience*) and optimising test-retest reliability (Jiang et al., 2023 *Network Neuroscience*). We and others have also successfully used the Schaefer atlas in several previous works involving multiple anaesthetics, psychedelics, and pathological perturbations of consciousness (Luppi et al., 2023 *Science Advances*; Huang et al., 2023 *Nature Communications*) including for recent investigations of brain fingerprinting under psychoactive substances (Tolle et al., 2023 *Network Neuroscience*; Mallaroni et al., 2024 *NeuroImage*). Therefore, there is robust consensus in the field that use of the Schaefer atlas is appropriate for studying anaesthetised fMRI brain data.

Crucially, however, we have also replicated our results using the Desikan-Killiany atlas, which is anatomically rather than functionally derived: this is reported in section *Replication, robustness and sensitivity*, subsection *Results are robust to use of different parcellations*, and Figure S24 (S19 in the original submission).

We now also added corresponding results for the regional ICC delta and its correlations with canonical brain maps, in Figure S25; results for Cognitive Matching from NeuroSynth are further shown in Figure S29, where we also show replication of our results when subcortex is added.

Overall, these extensive control analyses demonstrate that our results are robust to both atlas type (functional or anatomical) and size (from 68 to 232 regions, with or without subcortex).

In our revised manuscript, we now draw the reader's attention more explicitly to this important validation of the robustness of our results:

The present results were obtained using the Schaefer functional atlas, which is based on functional MRI data of awake individuals. To the best of our knowledge, there has been no report showing that the appropriateness of parcels in the Schaefer (or any other) functional atlas varies as a function of one's state of consciousness. In fact, we and others have successfully used the Schaefer atlas in previous works involving anaesthetic, psychedelic, and pathological perturbations of consciousness (Luppi et al., 2023 *Science Advances*; Huang et al., 2023 *Nature Communications*) including for brain fingerprinting under altered states of consciousness (Tolle et al., 2023 *Network*

Neuroscience; Mallaroni et al 2024 Neurolmage). Nevertheless, to show that our results are not critically dependent on the use of a functional parcellation derived from awake individuals, we replicate our results using an alternative parcellation of the cerebral cortex, the Desikan-Killiany atlas (Fig. S24 and Fig. S25). This atlas is based on anatomical landmarks; therefore the appropriateness of its parcels cannot be expected to vary under anaesthesia.

[...]

The propofol results can also be replicated using the anatomical Desikan-Killiany atlas (Fig. S26 and Fig. S27).

[...]

Overall, we clearly demonstrate that our present results are robust to both parcellation size (from 68 to 200 regions) and type (functional or anatomical).

Figure S24. Identifiability under sevoflurane anaesthesia is robust to parcellation choice | (a) Identifiability matrix between wakefulness and post-anaesthetic recovery. (b) Identifiability matrix between wakefulness and sevoflurane anaesthesia (right). Entries along the diagonal, represent self-self similarity (correlation of FC patterns), whereas off-diagonal entries represent self-other similarity. (c) Self-self similarity is significantly higher between two conscious states, than between wakefulness and sevoflurane anaesthesia. (d) The difference between self-self correlation and mean self-other correlation (differential identifiability) is significantly higher between two conscious states, than between wakefulness and sevoflurane anaesthesia. (e) The regional distribution of contributions to identifiability (change in intra-class correlation coefficient) is plotted on the cortical surface for the 68 ROIs of the Desikan-Killiany atlas. Box-plot: center line, median; box limits, upper and lower quartiles; whiskers, 1.5x interquartile range. ***, $p < 0.001$. N=15 human volunteers.

Figure S25. Anatomical characterisation of contributions to sevoflurane-induced loss of identifiability, in the anatomical Desikan-Killiany cortical atlas | (a) Matrix of edge-level difference in intra-class correlation coefficient between awake-recovery and awake-sevoflurane. (b) Regional distribution of propofol-induced loss of ICC, for the Desikan-Killiany anatomical atlas. (c) The sevoflurane-induced regional loss of ICC is significantly spatially aligned with the archetypal sensory-association axis of cortical organisation; the regional distribution of inter-individual variability of functional connectivity; and the regional expression of human-accelerated genes pertaining to brain function and development (“HAR-brain genes”); N = 68 regions from the Desikan-Killiany atlas.

Figure S29. Cognitive matching from meta-analytic patterns is robust to parcellation choice and inclusion of subcortex | (a) Results for Desikan-Killiany anatomical parcellation. (b) Results for the combined Schaefer-200 cortical atlas and Tian 32-ROI subcortical atlas. Ordinate: mean across time of the best decoding score (maximum spatial correlation between brain activity and 123 NeuroSynth meta-analytic maps). ***, p < 0.001 against wakefulness (FDR-corrected); *** (gray), p < 0.001 against recovery (FDR-corrected). Box-plot: center line, median; box limits, upper and lower quartiles; whiskers, 1.5x interquartile range. N=15 human volunteers.

Figure S28. Identifiability under anaesthesia is robust to inclusion of subcortex | (a) Identifiability matrix between wakefulness and post-anaesthetic recovery. (b) Identifiability matrix between wakefulness and sevoflurane anaesthesia (right). Entries along the diagonal, represent self-self similarity (correlation of FC patterns), whereas off-diagonal entries represent self-other similarity. (c) Self-self similarity is significantly higher between two conscious states, than between wakefulness and sevoflurane anaesthesia. (d) The difference between self-self correlation and mean self-other correlation (differential identifiability) is significantly higher between two conscious states, than between wakefulness and sevoflurane anaesthesia. (e) The regional distribution of contributions to identifiability (change in intra-class correlation coefficient) is plotted on the cortical surface for the 200-ROIs of the Schaefer atlas, and plotted in volumetric space for the 32 ROIs of the Tian subcortical atlas. Box-plot: center line, median; box limits, upper and lower quartiles; whiskers, 1.5 \times interquartile range. ***, $p < 0.001$. $N=15$ human volunteers.

- What's the underlying mechanisms for the reduced uniqueness between awake and anesthetized states? Could the authors elaborate on the physiological or neurobiological processes that underlie this change?

Identifiability reflects the difference between self-self similarity and self-other similarity. Figure S.22 (originally S17) suggests that as anaesthesia deepens, individuals' similarity to others is reduced, which on its own would increase identifiability. However, self-self similarity is reduced even more, resulting in an overall *decrease* of identifiability.

We also find that reduced identifiability is specifically localised in regions of transmodal association cortex. In the awake brain, these regions have the highest inter-individual variability across individuals, and are the ones that contribute the most to identifiability - meaning that their variability across individuals is not random, but rather it reflects individual identity. Indeed, transmodal regions have prolonged maturation times and high levels of synaptic plasticity, making them poised to change and adapt in response to environmental demands during the lifetime of each individual, which would account for their ability to encode individual-specific information in their functional connectivity. This is then temporarily suppressed by anaesthesia, as the present results indicate.

We speculate that anaesthetic-induced suppression of individual-specific differences in functional connectivity may be due to the consciousness-suppressing effects of anaesthesia. The default network in particular is well known to engage in both reflections about one's own past and future, which by definition are unique to each individual. By suppressing the idiosyncratic patterns of spontaneous thought that characterise the human brain even at rest, the present work indicates that anaesthetic-induced unconsciousness diminishes how such patterns are encoded in the macroscale activity and connectivity of the brain.

Our “cognitive matching” can shed light on this phenomenon: our results indicate that in the anaesthetised brain there is less prevalence of brain patterns corresponding to cognitive operations, compared with the conscious brain (awake/recovery). This is consistent with the idea that anaesthesia suppresses the unique “stream of consciousness” of each person, so that activity and connectivity in the anaesthetised brain become more driven by random fluctuations instead.

In the Discussion of our revised manuscript we have now expanded on our previous explanation, which now reads as follows:

“Specifically, we found that the functional connections whose contribution to identifiability is most affected, are those that most contribute to identifiability at baseline (Fig. S2), which are also those connecting pairs of transmodal regions (Fig. 2b). These results are consistent with the notion that transmodal cortices, such as the default network and fronto-parietal control network, are particularly susceptible to anaesthesia, and loss of consciousness more generally [8, 55, 56]. In addition, association cortices exhibit the greatest rate of inter-individual variability [35]. This variability is not mere noise, however, since the fronto-parietal and default networks consistently provide the largest contribution to identifiability in conscious individuals [9, 11, 13], indicating that their variability is individual-specific. This may be attributed to the fact that transmodal association cortices have the longest maturation times in the human brain, and the highest levels of synaptic plasticity and turnover [34, 57, 58]. Additionally, they also exhibit the lowest levels of intracortical myelination [34, 59], which is known to suppress plasticity both mechanically and chemically [57, 60]. As a result, transmodal cortices are relatively unconstrained by the underlying patterns of microstructure and anatomical connectivity [34, 61–63], and thus poised to change and adapt in response to environmental demands during the lifetime of each individual, which would account for their ability to encode individual-specific information in their functional connectivity. This individual-specific information in the functional interactions is then temporarily (and reversibly) suppressed by anaesthesia, as the present results indicate. Indeed, this account is consistent with recent evidence that individual differences in the functional connectivity and grey matter volume of frontal regions predict individual susceptibility to the behavioural effects of propofol sedation [64].

Indeed, we speculate that anaesthetic-induced suppression of individual-specific differences in functional connectivity may be due to the consciousness-suppressing effects of anaesthesia. The default network in particular is well known to engage in both reflections about one’s own past and future, which by definition are unique to each individual [65–67]. By suppressing the idiosyncratic patterns of spontaneous thought that characterise the human brain even at rest, the present work indicates that anaesthetic-induced unconsciousness diminishes how such patterns are encoded in the macroscale activity and connectivity of the brain. Indeed, we found that as anaesthesia deepens, spontaneous brain activity is increasingly less well characterised in terms of meta-analytic patterns pertaining to cognitive operations – whether automatically-defined or expert-curated. This effect is reversed upon recovery, despite the lingering presence of anaesthetic in the bloodstream.

Taken together, the results of diminished cognitive matching and diminished identifiability driven by loss of self-similarity suggest the following tentative account. During wakefulness, brain activity is driven by a combination of spontaneous physiological processes, and also the unique stream of consciousness of each individual, which brain activity must reflect. When consciousness is suppressed by anaesthesia, the physiological processes are perturbed, but most importantly, the main driver of what makes each person unique is gone, leading to reduced self-similarity and thus reduced identifiability - which is restored upon regaining consciousness.”

- The findings indicate that human FC under anesthesia, particularly during the sevoflurane burst-suppression state, showed the highest similarity to the macaque awake FC. It would be informative to also compare human FC with anesthetized macaque FC. If the similarity between anesthetized humans and anesthetized macaque is lower than that between anesthetized humans and awake macaques, it would further support the conclusion. However, if the similarity is greater, it may challenge the interpretation that anesthesia makes human FC more like an awake macaque. This additional comparison could provide insights into the specific effects of anesthesia on interspecies FC similarity.

We now provide a new analysis of proximity between human and macaque FC, from two different macaque databases (the original awake dataset and a new dataset acquired under isoflurane anaesthesia), in the joint space of two principal components from PCA.

It is clear to see that as anaesthesia deepens, the FC of the human brain moves closer to the location of macaque FC (Fig. 4a), especially for awake but also for anaesthetised macaques, which are similarly distant from awake human FC (Fig. 4b). We conclude that anaesthesia shifts human FC closer to macaque FC, regardless of whether the macaques are awake or anaesthetised.

Further supporting analyses (including additional comparisons; a different distance metric; and 3D instead of 2D representation) are provided in Supplementary Figures S9, S10, and S11.

From our revised Results section:

“We can now use Principal Components Analysis (PCA) to project all FC patterns across humans and macaques in a common low-dimensional space (see Methods). PCA is widely used for dimensionality reduction and visualisation of high-dimensional data, because it provides a low-dimensional representation of the data while preserving as much of the original variability as possible. This approach enables us to re-represent each individual's FC as a point in a 2-dimensional plane, where each dimension corresponds to one of the main axes of variation in the data. We can then follow how the location of individuals' FC changes in this low-dimensional space, as a function of anaesthesia. We clearly see that each condition (awake, recovery, and various levels of sevoflurane anaesthesia) tends to occupy a different region of the space spanned by the first two principal components (Fig.4a). It is also immediately apparent that as the dose of sevoflurane increases, individuals' FC patterns move progressively further away from the initial position of pre-anaesthesia wakefulness

data - only to then return closest to their initial awake position upon post-anaesthetic recovery of wakefulness. We formally quantify this shift in terms of Euclidean distance in the 2D space of the first two PCs: the average Euclidean distance between pairs of data-points from wakefulness and from other conditions is significantly lower for recovery than for any anaesthetised condition, and increases with increasing depth of anaesthesia (Fig.4b). Notably, as the human anaesthetised FC patterns move away from wakefulness, they exhibit a clear shift towards the location of the macaque FC datasets -- with burst-suppression (the deepest level of anaesthesia) and the two macaque datasets being similarly distant from human wakefulness FC. [...]

Thus, anaesthesia shifts the functional connectivity of the human brain closer to the functional connectivity of the non-human primate brain, and this shift is reversed upon recovery. This result complements our observation that anaesthetic-induced reduction in regional identifiability is most pronounced in regions of the human brain that are genetically most human-specific.”

Figure 4. Anaesthesia shifts human functional connectivity closer to macaque functional connectivity | (a) Low-dimensional projection of the human and macaque functional connectivity, in the space of the first two principal axes of variation from Principal Components Analysis. Each circle represents the FC from one human, with colour reflecting condition (awake, recovery, or different doses of sevoflurane). Each diamond represents FC from one macaque, with colour representing the dataset (awake or anaesthetised). (b) Distribution of Euclidean distances from awake humans' FC patterns, in the 2D space from (a). N = 225 (15×15) pairs of data-points for human data; N = 285 (15×19) pairs of data-points for the awake macaque data; N = 135 (15×9) pairs of data-points for the anaesthetised macaque data. *** (gray), $p < 0.001$ against recovery (FDR-corrected). Box-plots: center line, median; box limits, upper and lower quartiles; whiskers, 1.5× interquartile range. See Supplementary Data 2 for full statistical reporting. Source data are provided as a source data file.

- The result shows a positive correlation in edge-wise FC between humans and macaques, yet this relationship becomes negative when FC is averaged into regional strength (Fig S7). Can the authors provide an explanation to understand this result? And how does the regional FC strength vary across regions (Fig 4a)? How did the authors rule out the impact of fMRI noise (e.g. signal inhomogeneity, SNR, etc.)?

The reviewer asks why inter-species correlations are negative at the node level, yet positive at the edge level. While this can seem unintuitive, it is not mathematically inconsistent: grouping observations (in this case, by averaging all edges belonging to the same node, in order to obtain node-wise FC) can change the direction of a correlation. Please also note that our results of greater inter-species similarity under anaesthesia were observed both at the node level and at the edge level, in both datasets (Figure S8 and Figure S15 in the original submission).

However, since this analysis generated confusion for several reviewers and took the focus away from our main result of reduced identifiability under anaesthesia, after careful deliberation we have decided to remove it. Instead, in the new Figure 4 we have followed the suggestion of Reviewer #3, showing distances in a joint space of two principal components from PCA.

Figure 4. Anaesthesia shifts human functional connectivity closer to macaque functional connectivity | (a) Low-dimensional projection of the human and macaque functional connectivity, in the space of the first two principal axes of variation from Principal Components Analysis. Each circle represents the FC from one human, with colour reflecting condition (awake, recovery, or different doses of sevoflurane). Each diamond represents FC from one macaque, with colour representing the dataset (awake or anaesthetised). (b) Distribution of Euclidean distances from awake humans' FC patterns, in the 2D space from (a). N = 225 (15×15) pairs of data-points for human data; N = 285 (15×19) pairs of data-points for the awake macaque data; N = 135 (15×9) pairs of data-points for the anaesthetised macaque data. *** (gray), $p < 0.001$ against recovery (FDR-corrected). Box-plots: center line, median; box limits, upper and lower quartiles; whiskers, 1.5× interquartile range. See Supplementary Data 2 for full statistical reporting. Source data are provided as a source data file.

The reviewer also asks about the potential impact of fMRI noise. We followed rigorous denoising procedures to account for motion and physiological artefacts (please see *Methods*), and the Results section of revised manuscript includes a subsection *Robustness against head motion*, where we provide details of numerous checks that we have implemented.

Additionally, in the new Results and Figure S6, we now show that our spatial correlations replicate after regressing out a map of the human brain's regional temporal signal-to-noise ratio, from the map of anaesthetic-induced regional ICC differences.

Figure S6. Anatomical characterisation of contributions to sevoflurane-induced loss of identifiability, after regressing out regional signal-to-noise ratio of the BOLD timeseries (tSNR) | (Left) Map of the human brain's signal-to-noise ratio of regional functional MRI signal, from [32]. (Right) The sevoflurane-induced regional loss of ICC is significantly spatially aligned with the archetypal sensory-association axis of cortical organisation; the regional distribution of inter-individual variability of functional connectivity; the regional cortical expansion between macaque and human; and the regional expression of human-accelerated genes pertaining to brain function and development ("HAR-brain genes"), even after regressing out the tSNR map using linear regression; $N = 200$ regions from the Schaefer atlas.

- Relatedly, since the cross-species comparison was conducted using regional strength rather than edge-wise FC, and given the opposing relationships observed (positive for edge-wise FC but negative for regional averages), are the cross-species comparison replicable for edge-wise FC (Fig 4b)?

Yes, the cross-species comparison results successfully replicate with edge-wise FC: this was shown in Figure S8 and Figure S15 of the original manuscript. However, as detailed above, please note that we have now removed these analyses from our revised manuscript.

- How do we understand the result that on one hand, human brains when unconscious are more similar to the macaque awake brains, yet simultaneously, they become less similar to each other? This seems paradoxical. Is this also true for edge-wise FC?

Yes, this is also true for edge-wise FC (see previous answer; Figure S8 and Figure S15 of the original submission).

We acknowledge that our results of reduced similarity between individuals, but increased similarity to the macaque connectome, can seem counterintuitive when taken together. However, please see the new Figure S32 for simple one-dimensional and two-dimensional examples showing that in fact both facts can be true together, without inconsistency. In both cases, at $T=2$ objects A and B become each closer to C than at $T=1$, while the distance between A and B themselves increases.

Figure S32. Simple visual examples of how two objects can become more similar to a third, while also becoming less similar to each other | For both the 1D example (left) and 2D example (right), proximity reflects similarity. At $T=2$, both A and B are closer to C than they were at $T=1$; however, the distance between them has increased. This phenomenon is not constrained to low-dimensional scenarios: Supplementary Code 1 provides MATLAB code for a toy example of two matrices that decrease their correlation with each other, while each increasing their correlation with a third matrix C.

Importantly, this phenomenon is not constrained to low-dimensional scenarios. To prove this point, in addition to these low-dimensional examples we now provide MATLAB code (Supplementary Code 1) for a toy model showing an example of how two matrices can each become more similar to a third matrix, while also becoming less similar to each other.

Output of Supplementary Code 1. At time 2, both A and B are more correlated with C than they were at time 1; simultaneously, however, the similarity between A and B at time 2 is lower than it was at time 1.

- Could the authors provide visualizations of the matched parcellations between the two species? Additionally, how well these cross-species parcellations capture homologous regions between macaque and human brains?

We appreciate this suggestion, and we now provide a figure showing the correspondence between human and macaque versions of the Regional Mapping atlas (Fig. S8). Please note that we did not make these parcellations ourselves: rather, Kotter and Wanke (2005 *PTRSB*) made the macaque parcellation with the specific aim of enabling inter-species comparison, and Bezgin et al (2017 *Human Brain Mapping*) recently translated it to human MNI space in a dedicated publication. These authors' explicit purpose was to provide a common atlas to map the two species' brains onto each other. We refer the reader to the original publications for further details.

- The authors stated that “Correlation analysis cannot identify causal determinants” and further performed the dominance analysis. However, dominance analysis also does not infer causality.

We apologise for our lack of clarity. Indeed, dominance analysis is not a form of causal inference, and we did not mean to suggest that it is. Rather, our focus was on the additional insight provided by multivariate analysis, beyond what is provided by individual correlations. We have rephrased our explanation in the manuscript accordingly:

“Correlation analysis cannot identify causal determinants of regional changes in identifiability. Nevertheless, multivariate analysis can be helpful to provide insights beyond what is available from multiple individual correlations.”

- There are >900 cognitive concepts listed in the Cognitive Atlas. What criteria or processes were used here to select 123 terms in the final analysis?

We used the same subset of terms overlapping between NeuroSynth and the Cognitive Atlas (Poldrack 2011 *Frontiers*) that were used in several previous studies (Alexander-Bloch et al 2018 *NeuroImage*; Markello & Misic 2021 *NeuroImage*; Hansen et al 2021 *Nature Human Behaviour*).

All this information is reported in the *Methods* section of our manuscript, subsection *Meta-analytic cognitive matching from NeuroSynth*:

“Although more than a thousand terms are catalogued in the NeuroSynth engine, we refine our analysis by focusing on cognitive function and therefore we limit the terms of interest to cognitive and behavioural terms. To avoid introducing a selection bias, we opted for selecting terms in a data-driven fashion instead of selecting terms manually. Therefore, terms were selected from the Cognitive Atlas, a public ontology of cognitive science [96], which includes a comprehensive list of neurocognitive terms. This approach totaled to $t = 123$ terms, ranging from umbrella terms (“attention”,

“emotion”) to specific cognitive processes (“visual attention”, “episodic memory”), behaviours (“eating”, “sleep”), and emotional states (“fear”, “anxiety”) (note that the 123 term-based meta-analytic maps from NeuroSynth do not explicitly exclude patient studies). The Cognitive Atlas subdivision has previously been used in conjunction with NeuroSynth [121-123], so we opted for the same approach to make our results comparable to previous reports”

In our revised manuscript, we have now added a further pointer to this subsection in the main Results section, to draw the reader’s attention (“see Methods section for details of how the NeuroSynth brain maps were selected”).

Please also note that our cognitive matching results are not dependent on the specific choice of meta-analytic terms, or even the specific meta-analytic database used: we replicated our findings using a different set of terms from a different, expert-curated meta-analytic database, BrainMap (see Figure S17 and subsection *Alternative meta-analytic matching from BrainMap*, as well as the corresponding section in the Methods for details).

“This successful replication indicates that our cognitive matching procedure is robust both to the specific choice of which terms to include (which are different between BrainMap and our intersection of NeuroSynth and the Cognitive Atlas (Poldrack 2011 *Frontiers*); and to the choice of meta-analytic database more broadly.”

We now provide the full list of NeuroSynth terms used, and BrainMap terms used, in Supplementary Tables 1 and 2.

Minor:

- Fig 2a, ICC should be in a range of 0-1. It is noted that the ICC matrix from awake to sevoflurane includes negative (blue) values?

We are unsure why the Reviewer believes that ICC should be in the range between 0 and 1, analogously to Pearson correlation. The fact that ICC can take negative values is also evident from Figure 3 of Amico and Goni (2018 *Scientific Reports*).

To avoid confusion, in our revised manuscript we now explicitly specify that the ICC ranges between -1 and +1 (see revised Figure 3 and Methods).

- Many of the figures include the color maps without absolute values on the colorbar. Can the authors provide the absolute values although the comparisons between metrics were relative across regions?

For Figures 2d and Figure S14d, values for the range of the colormap are provided on the y-axis of the scatter-plots directly below each brain map. We have now clarified this in the corresponding figure captions. We have added the values of the colorbars for Figure 3a. All other colormaps display the corresponding values.

Fig2b The authors have merged the edge identifiability for the within-unimodal and uni-transmodal edges. What was the rationale behind combining these two types of edges?

We combined unimodal-unimodal and unimodal-transmodal edges because we wanted to test whether anaesthesia has particularly large effects on edges connecting transmodal regions, versus all other edges. Figure S3 shows the results of combining edges in different ways.

Remove the duplicate “with” in “Replication of cognitive matching with with BrainMap”

Done: thank you for pointing this out.

Reviewer #2:

Luppi et al. utilized fMRI while subjects were under general anesthesia to explore its effects on human brain uniqueness. Their findings indicate that anesthesia reduces the self-similarity and individual distinctiveness in brain function, particularly in areas linked to sensory-association. This phenomenon was observed consistently across various anesthetics, was reversible after the subjects recovered from anesthesia, and led to a closer resemblance between the functional connectivity patterns of human and macaque brains. The study suggests that anesthesia not only blurs the unique features between individual human brains but also makes the human brain functionally more similar to that of other primates, especially in regions that are more developed in humans. My detailed comments are provided below.

1.While the author suggests that sevoflurane anesthesia diminishes the identifiability of individual connectomes, the comparisons used—'recovery vs awake' and 'sevo vs awake'—primarily demonstrate deviations from the awake state rather than a definitive loss of identifiability. To more effectively support this conclusion, it would be beneficial to include a 'sevo vs sevo' comparison alongside the 'awake vs awake' scenario. Additionally, the analysis concerning the ICC could be expanded to compare 'sevo vs sevo' with 'awake vs awake.'

It's important to note, however, that previous research suggests that under anesthesia, brain functional networks might align more closely with structural connections, which remain highly individualized. Therefore, ICC values for 'sevo vs sevo' under anesthesia might not necessarily be low, potentially indicating maintained individual distinctiveness despite the anesthetic state.

Figures S20 and S22 (formerly Fig. S16 and S17 from the original submission) provide a comparison of two awake states (awake and recovery) against two anaesthetised states with different levels of sevoflurane. This is reported in the main *Results* section, subsection *Anaesthesia reduces within-state identifiability of the human functional connectome*:

To empirically demonstrate that these results are not simply due to comparing within-state correlations against between-state correlations (with ‘state’ in this context referring to wakefulness or anaesthesia), we take advantage of the fact that our sevoflurane data include multiple scans obtained under anaesthesia. This allows us to compare two conscious scans (baseline and recovery) and two anaesthetised scans: either vol 2% and 3% sevoflurane, or vol 3% sevoflurane and burst-suppression.

Results for the ICC analysis comparing sevo-vs-sevo (vol 3% vs burst-suppression) with awake-vs-awake (pre-anaesthetic baseline and post-anaesthetic recovery) are provided in the new Figure S21.

Figure S20. Replication of identifiability results using within-state comparisons | (a) Identifiability matrix between wakefulness and post-anaesthetic recovery. (b) Identifiability matrix between vol 2% sevoflurane and vol 3% sevoflurane anaesthesia. Entries along the diagonal, represent self-self similarity (correlation of FC patterns), whereas off-diagonal entries represent self-other similarity. (c) Self-self similarity is significantly higher between two conscious states, than between vol 2% and vol 3% sevoflurane. (d) The difference between self-self correlation and mean self-other correlation (differential identifiability) is significantly higher between two conscious states, than between vol 2% and vol 3% sevoflurane. (e) The regional distribution of contributions to identifiability (change in intra-class correlation coefficient) is plotted on the cortical surface. (f) Identifiability matrix between wakefulness and post-anaesthetic recovery. (g) Identifiability matrix between vol 3% and burst-suppression level of sevoflurane anaesthesia. (h) Self-self similarity is significantly higher between two conscious states, than between vol 3% and burst-suppression level of sevoflurane. (i) The difference between self-self correlation and mean self-other correlation (differential identifiability) is significantly higher between two conscious states, than between vol 3% and burst-suppression level of sevoflurane. (j) The regional distribution of contributions to identifiability (change in intra-class correlation coefficient) is plotted on the cortical surface. *, $p < 0.05$; ***, $p < 0.001$.

Figure S21. Replication of edge-wise and regional identifiability results using within-state comparisons | (a) Matrix of edge-level difference in intra-class correlation coefficient between awake-recovery and vol 2% - vol 3% sevoflurane anaesthesia. (b) Regional distribution of propofol-induced loss of ICC (awake-recovery versus 2%-3% sevoflurane). (c) The sevoflurane-induced regional loss of ICC (awake-recovery versus 2%-3% sevoflurane) is significantly spatially aligned with the archetypal sensory-association axis of cortical organisation; the regional distribution of inter-individual variability of functional connectivity; the regional cortical expansion between macaque and human; and the regional expression of human-accelerated genes pertaining to brain function and development (“HAR-brain genes”). (d) Matrix of edge-level difference in intra-class correlation coefficient between awake-recovery and vol 3% - burst-suppression levels of sevoflurane anaesthesia. (e) Regional distribution of propofol-induced loss of ICC (awake-recovery versus 3% sevoflurane - burst-suppression). (f) The sevoflurane-induced regional loss of ICC (awake-recovery versus 3% sevoflurane - burst-suppression) is significantly spatially aligned with the archetypal sensory-association axis of cortical organisation; the regional distribution of inter-individual variability of functional connectivity; the regional cortical expansion between macaque

and human; and the regional expression of human-accelerated genes pertaining to brain function and development (“HAR-brain genes”). For all scatter-plots, N = 200 regions from the Schaefer atlas.

Figure S22. Distributions of self-self and self-other correlations for different pairs of conditions | (a) Two conscious states: awake versus post-anaesthetic recovery. (b) Vol 2% versus vol 3% sevoflurane. (c) Vol 3% versus burst-suppression levels of sevoflurane.

2. Figure 4 posits a closer resemblance between human and macaque cortical connectivity under anesthesia than in wakefulness. Yet, the methods for this comparison remain unspecified. Moreover, the negative correlation reported for awake connectivity diverges from prior research suggesting homology in these networks. It would be beneficial for the author to introduce additional awake-state data to contextualize and clarify this unexpected correlation.

Comparison was performed by either node-wise or edge-wise correlation, as shown in the original Figure S7 of our initial submission. Specifically, the node-wise comparison was performed by obtaining regional node strength (mean of each region’s functional connections) for each human and for the macaque group-average FC matrix, and then spatially correlating the resulting vectors. This was illustrated in Figure 4a of our initial submission, and its legend.

We have now added Figure S8 illustrating the correspondence between the macaque and human parcellations made by Bezgin et al (2017 Human Brain Mapping).

We also emphasise that although the correlation is negative at the level of nodes, the correlation is instead positive at the level of edges (this was shown in Figure S7 of our initial submission). Both of these relationships between the macaque and human data were replicated in the awake data of our replication dataset (this was shown in Figures S14 and S15 of our initial submission).

However, because these analyses clearly were a source of confusion among reviewers, we have now replaced them with a different approach, projecting the data into the space of the first 2 principal components. This new analysis clearly shows, in a more intuitive manner, that anaesthesia shifts human FC patterns to be closer to macaque FC (both awake and anaesthetised).

From the Methods of the revised manuscript:

We use Principal components analysis (PCA) to obtain a low-dimensional representation of the human and macaque FC in a common space. PCA re-represents the data in terms of linearly uncorrelated (orthogonal) variables called principal components, which are extracted from the data themselves as the axes of maximum variation (Jolliffe 2002). Therefore, PCA is widely used for dimensionality reduction and visualisation of high-dimensional data, because it provides a low-dimensional representation of the data while preserving as much of the original variability as possible.

To obtain the joint PCA space, we begin by vectorising the upper triangular of each FC matrix, for each scan (awake, recovery, and different levels of anaesthesia) of each individual. We also do the same for all available scans in the awake and anaesthetised macaque datasets. The vectorised FC patterns are then concatenated, forming separate columns of a matrix M whose rows correspond to FC edges. The PCA algorithm is then applied to this matrix M , to extract linearly orthogonal principal components of maximum variability (ranked in descending order of variance explained). The algorithm also provides weights that associate each column of M with each of the extracted PCs. We can use these weights to obtain a low-dimensional representation of each column of M (corresponding to FC patterns) as a point in the space of principal components. In our main analyses we use the first two principal components to define the low-dimensional space, but we also replicate our results using the first three principal components.

From the Results of the revised manuscript:

“We can now use Principal Components Analysis (PCA) to project all FC patterns across humans and macaques in a common low-dimensional space (see Methods). PCA is widely used for dimensionality reduction and visualisation of high-dimensional data, because it provides a low-dimensional representation of the data while preserving as much of the original variability as possible. This approach enables us to re-represent each individual's FC as a point in a 2-dimensional plane, where each dimension corresponds to one of the main axes of variation in the data. We can then

follow how the location of individuals' FC changes in this low-dimensional space, as a function of anaesthesia. We clearly see that each condition (awake, recovery, and various levels of sevoflurane anaesthesia) tends to occupy a different region of the space spanned by the first two principal components (Fig.4a). It is also immediately apparent that as the dose of sevoflurane increases, individuals' FC patterns move progressively further away from the initial position of pre-anaesthesia wakefulness data - only to then return closest to their initial awake position upon post-anaesthetic recovery of wakefulness. We formally quantify this shift in terms of Euclidean distance in the 2D space of the first two PCs: the average Euclidean distance between pairs of data-points from wakefulness and from other conditions is significantly lower for recovery than for any anaesthetised condition, and increases with increasing depth of anaesthesia (Fig.4b). Notably, as the human anaesthetised FC patterns move away from wakefulness, they exhibit a clear shift towards the location of the macaque FC datasets -- with burst-suppression (the deepest level of anaesthesia) and the two macaque datasets being similarly distant from human wakefulness FC. [...]

Thus, anaesthesia shifts the functional connectivity of the human brain closer to the functional connectivity of the non-human primate brain, and this shift is reversed upon recovery. This result complements our observation that anaesthetic-induced reduction in regional identifiability is most pronounced in regions of the human brain that are genetically most human-specific.”

Figure 4. Anaesthesia shifts human functional connectivity closer to macaque functional connectivity | (a) Low-dimensional projection of the human and macaque functional connectivity, in the space of the first two principal axes of variation from Principal Components Analysis. Each circle represents the FC from one human, with colour reflecting condition (awake, recovery, or different doses of sevoflurane). Each diamond represents FC from one macaque, with colour representing the dataset (awake or anaesthetised). **(b)** Distribution of Euclidean distances from awake humans' FC patterns, in the 2D space from (a). N = 225 (15×15) pairs of data-points for human data; N = 285 (15×19) pairs of data-points for the awake macaque data; N = 135 (15×9) pairs of data-points for the anaesthetised macaque data. *** (gray), $p < 0.001$ against recovery (FDR-corrected). Box-plots: center line, median; box limits, upper and lower quartiles; whiskers, 1.5× interquartile range. See Supplementary Data 2 for full statistical reporting. Source data are provided as a source data file.

From the Discussion of the revised manuscript:

Furthermore, we found that as anaesthesia deepens, it shifts the position of the human functional connectome closer to the macaque functional connectome in a joint low-dimensional space -- returning close to the initial position upon recovery.

3. The author, through sophisticated analyses, claims that anesthesia reduces the uniqueness of brain functional connectivity among individuals and across species, primarily using a resting-state paradigm. The main finding is that the intrasubject correlation for 'sevo vs awake' is significantly lower than that for 'recovery vs awake.' However, this result are not necessarily specific to the state of unconsciousness; previous study showed that the awake resting state possesses the highest fingerprinting capacity, with task states generally exhibiting weaker correlation than the resting state (Emily et al, 2015). The author should discuss or conduct additional experiments to elucidate the specificity of these results.

Our main result is that upon anaesthetic-induced loss of consciousness, brain identifiability is reduced. However, we never suggested that loss of consciousness is the *only* way to reduce the uniqueness of the functional connectome. In fact, in our Discussion we explicitly addressed the effect of other perturbations of consciousness on fingerprinting - including one that reduces identifiability:

“a recent report suggests that psilocybin increases the idiosyncrasy of functional connectivity, resulting in greater differential identifiability [46] - the opposite of what we found here with different anaesthetics. Of note, decreased idiosyncrasy of FC was recently reported with another psychedelic, ayahuasca [81], in ritualistic users of psychedelics (members of the Santo Daime religious community). This result suggests that psychedelics may be able to modulate FC idiosyncrasy in both directions: increasing distinctiveness among strangers, but increasing similarity among individuals for whom the psychedelic experience is part of a shared, ritualised cultural experience - which is likely to induce a commonality of mental state among individuals”

Likewise, in our Results we also discussed the effect of tasks, addressing evidence that tasks can both increase or decrease identifiability vis-a-vis rest, depending on how tightly controlled they are:

“We find that as anaesthesia deepens, both self-similarity and the difference between self-self and self-other correlations (i.e., identifiability) are progressively reduced, with the two distributions becoming increasingly overlapping (Fig. S22). This pattern is the reverse of what was recently found by [47], who showed that tightly controlled cognitive tasks increase self-other similarity, but increase self-self similarity even more, thereby resulting in an overall increase in identifiability”

Nevertheless, we now explicitly added the following text in our Discussion:

“Indeed, Colenbier et al (2023) and Finn et al (2015) showed that brain identifiability can be modulated by different cognitive tasks. Therefore, although our main result is

that brain identifiability is reduced upon anaesthetic-induced loss of consciousness, this is not a claim that anaesthesia is the only way to reduce the uniqueness of the functional connectome.”

Reviewer #3:

Remarks to the Author:

Luppi et al., showed that anesthesia reduces uniqueness of functional connectivity (FC) across individuals and species. The "fingerprint" of individual FC in resting eye-closed state was diminished under anesthesia and recovered along with return of consciousness. The authors further found that the greatest decrease in identifiability was in transmodal cortex. Furthermore, similarity between fMRI BOLD signal and cognitive map was substantially reduced in anesthesia. I found the question the authors ask, i.e., does anesthesia suppress uniqueness of functional connectivity?, is interesting and the corresponding data were nicely visualized as boxplot, distributions and individual points, which may help readers understand the manuscript. There are a few questions and comments regarding manuscripts:

Thank you for this positive assessment.

1) To my understanding, uniqueness requires both diversity across individuals and persistency over time. For example, if FC configurations are all the same, identifiability will be low even though FC persists over time. On the other hand, although FC configurations are diverse enough across individuals, identifiability will be low if FC varies much over time. Considering this, it is unclear how the identifiability was reduced by anesthetics. Is it because of (a) anesthesia makes all FC configurations very similar each other (reduced diversity)? Or (b) FCs are still diverse but their persistency is decreased by anesthesia?

Indeed, we quantify identifiability as self-self correlation across scans (corresponding to the Reviewer's term of persistency over time) minus self-other correlation (reflecting diversity across individuals). So the envisioned scenario (a) "anesthesia makes all FC configurations very similar each other (reduced diversity)" corresponds to higher self-other correlation; whereas scenario (b) "FCs are still diverse but their persistency is decreased by anesthesia" corresponds to decreased self-self correlation.

We have now added this clarification, including the Reviewer's helpful terminology, at the beginning of our Results section:

“Successful brain fingerprinting requires two conditions. The first is persistency: an individual's FC needs to be consistent over time, in order to be used to identify the individual. The second is diversity: the FCs of distinct individuals need to be different from each other, to avoid confusing individuals. If FC patterns are all the same, identifiability will be low even though FC persists over time. On the other hand, if FC is very variable over time identifiability will be low, even if FC configurations are diverse across individuals. We quantify persistency as self-self correlation across scans

(correlation between FC at time 1 and time 2, for the same individual). We quantify (lack of) diversity as the mean self-other correlation: the mean correlation between an individual's FC at time 1, and every other individual's FC at time 2. Finally, differential identifiability is the difference between self-self correlation (persistence) and self-other correlation (lack of diversity).”

In our results we show that anaesthesia reduces both self-self correlation (persistence; Figure 1c, S1c, S1h, S12c, S13c), and identifiability (self-self correlation minus self-other correlation; Figure 1d, S1d, S1i, S12d, S13d). In particular, Figure S22 shows that diversity is actually increased (lower self-other correlation), but persistence is decreased (lower self-self correlation). The reduction in persistence is greater than the increase in diversity, leading to an overall reduction in identifiability.

Figure S22. Distributions of self-self and self-other correlations for different pairs of conditions | (a) Two conscious states: awake versus post-anaesthetic recovery. (b) Vol 2% versus vol 3% sevoflurane. (c) Vol 3% versus burst-suppression levels of sevoflurane.

2) The Fig S16 suggests that FC in anaesthesia is more diverse and probably less structured ((b) in the above), because the off-diagonal components of the matrix in Fig S16 b and g is low. That is, there may be large interindividual variation in FC configuration under anaesthesia. If so, I have following questions. In EEG/MEG studies, it has been well known that anesthetized brain has stereotypic FC configuration (Cimenser et al., 2011; Purdon et al., 2013; Blain-Moraes et al., 2017). For example,

anesthesia results in a stereotypical FC pattern, so-called "anteriorization". Coherence and power distribution becomes very structured, and moreover, it is consistent across individuals. How would you reconcile your result (Fig. S16 b and g) with this seemingly opposite result of EEG/MEG?

Thank you for bringing these intriguing works to our attention. fMRI and M/EEG reflect different neurobiological processes and operate at different spatial and temporal scales. In particular, the phenomenon of alpha anteriorization mentioned by the Reviewer occurs at a timescale that is several orders of magnitude faster than the BOLD signal fluctuations studied here (8-12 Hz vs 0.008-0.09 Hz). Please also note our toy examples about how two objects can each become more similar to a common template, yet simultaneously reduce their similarity to each other.

We have included a discussion of this point in our revised manuscript:

“From the EEG literature it is also well established that anaesthesia induces so-called “anteriorization” of the distribution of EEG alpha oscillations (8-12 Hz), with the peak of alpha power shifting from occipital to frontal electrodes (Cimenser et al., 2011; Purdon et al., 2013; Blain-Moraes et al., 2017). Although we find that anaesthesia reduces the overall identifiability of the fMRI connectome, it may still seem counterintuitive that individuals’ fMRI FC patterns become less similar to each other under anaesthesia, since the EEG topography is expected to undergo similar reconfigurations across individuals. However, it is essential to realise that it is not inconsistent for two objects A and B to each become more similar to a third object C, while at the same time becoming less similar to each other (see Supplementary Code 1, and also Fig. S.32 for illustrations of this phenomenon in 1D and 2D). Additionally, fMRI and EEG reflect different neurobiological processes and operate at different spatial and temporal scales. M/EEG co-fluctuation patterns of different frequency bands can look very different from each other and from fMRI, and carry different information for fingerprinting (Sareen 2021 *Neuroimage*, Shafiei 2022 *Plos Biology*, da Silva 2021 *Nature Commun*). In particular, the phenomenon of EEG alpha anteriorization occurs at a timescale that is several orders of magnitude removed from the fMRI BOLD signal fluctuations studied here (8-12 Hz vs 0.008-0.09 Hz). Lastly, anteriorisation pertains to the behaviour of regions, whereas fingerprinting is predicated on the interactions between different regions. In Supplementary Code 2, we provide an example of two systems that each undergo the same change in the spatial pattern of amplitude of activity of each element, while simultaneously decreasing their correlation at the level of edges. Altogether, any of these factors could explain the coexistence of our fMRI results about reduced inter-individual and inter-species distinctiveness, with the phenomenon of anaesthetic-induced EEG anteriorisation. Teasing these factors apart with dedicated studies that explicitly investigate M/EEG brain fingerprinting under anaesthesia represents a promising avenue for future work.”

Output of Supplementary Code 2. A and B comprise 82 nodes, each with a temporal signal. At time 1, the FCs (node-by-node correlations) of A and B are correlated with each other, but A and B exhibit different patterns of signal amplitude. At time 2, both A and B have nearly identical patterns of signal amplitude, even though their FCs are now less correlated than at time 1.

3) If anesthesia does not make FC configuration stereotypical (Fig. S16 b and g), how do FCs of most individuals under anesthesia become highly correlated with macaque's FC?

We appreciate that this result can seem counterintuitive. However, it is in fact entirely plausible. Please see the new Figure S32 for simple one-dimensional and two-dimensional examples showing that in fact both facts can be true together, without inconsistency. In both cases, at T=2 objects A and B become each closer to C than at T=1, while the distance between A and B themselves increases. Please see also our MATLAB toy model (Supplementary Code 1), for a toy example in higher dimension.

Figure S32. Simple visual examples of how two objects can become more similar to a third, while also becoming less similar to each other | For both the 1D example (left) and 2D example (right), proximity reflects similarity. At T=2, both A and B are closer to C than they were at T=1; however, the distance between them has increased.

Output of Supplementary Code 1. At time 2, both A and B are more correlated with C than they were at time 1; simultaneously, however, the similarity between A and B at time 2 is lower than it was at time 1.

4) I highly recommend the authors to add an additional figure (a subplot or in supplementary), visualizing the FC configurations in the 2d or 3d space. For example, scatter or schematic plot of individual FCs in low-dimensional 2d or 3d space (after dimensionality reduction by t-SNE or UMAP). By doing so, the readers would be able to understand how anesthesia reconfigures FC structure: e.g., does FC become stereotypical (all individual FC under anesthesia are clustered or distant each other?) by anesthetics?, Why and how identifiability was reduced?, visualization of distances between FC of consciousness vs FC of anesthesia vs FC of macaque, etc.

Thank you for this excellent suggestion. We have now included a plot of the projection of the data onto the first two principal components, which clearly shows how anaesthesia systematically shifts FC to different locations in the 2-dimensional PCA space. As anaesthesia deepens, human FC is shifted closer to the macaque FC, and then the original location is restored upon recovery.

“We can now use Principal Components Analysis (PCA) to project all FC patterns across humans and macaques in a common low-dimensional space (see Methods). PCA is widely used for dimensionality reduction and visualisation of high-dimensional data, because it provides a low-dimensional representation of the data while preserving as much of the original variability as possible. This approach enables us to re-represent each individual's FC as a point in a 2-dimensional plane, where each dimension corresponds to one of the main axes of variation in the data. We can then follow how the location of individuals' FC changes in this low-dimensional space, as a

function of anaesthesia. We clearly see that each condition (awake, recovery, and various levels of sevoflurane anaesthesia) tends to occupy a different region of the space spanned by the first two principal components (Fig.4a). It is also immediately apparent that as the dose of sevoflurane increases, individuals' FC patterns move progressively further away from the initial position of pre-anaesthesia wakefulness data - only to then return closest to their initial awake position upon post-anaesthetic recovery of wakefulness. We formally quantify this shift in terms of Euclidean distance in the 2D space of the first two PCs: the average Euclidean distance between pairs of data-points from wakefulness and from other conditions is significantly lower for recovery than for any anaesthetised condition, and increases with increasing depth of anaesthesia (Fig.4b). Notably, as the human anaesthetised FC patterns move away from wakefulness, they exhibit a clear shift towards the location of the macaque FC datasets -- with burst-suppression (the deepest level of anaesthesia) and the two macaque datasets being similarly distant from human wakefulness FC.

[...]

Thus, anaesthesia shifts the functional connectivity of the human brain closer to the functional connectivity of the non-human primate brain, and this shift is reversed upon recovery. This result complements our observation that anaesthetic-induced reduction in regional identifiability is most pronounced in regions of the human brain that are genetically most human-specific.”

Figure 4. Anaesthesia shifts human functional connectivity closer to macaque functional connectivity | (a) Low-dimensional projection of the human and macaque functional connectivity, in the space of the first two principal axes of variation from Principal Components Analysis. Each circle represents the FC from one human, with colour reflecting condition (awake, recovery, or different doses of sevoflurane). Each diamond represents FC from one macaque, with colour representing the dataset (awake or anaesthetised). (b) Distribution of Euclidean distances from awake humans' FC patterns, in the 2D space from (a). N = 225 (15×15) pairs of data-points for human data; N = 285 (15×19) pairs of data-points for the awake macaque data; N = 135 (15×9) pairs of data-points for the anaesthetised macaque data. *** (gray), $p < 0.001$ against recovery (FDR-corrected). Box-plots: center line, median; box limits, upper and lower quartiles; whiskers, 1.5× interquartile range. See Supplementary Data 2 for full statistical reporting. Source data are provided as a source data file.

5) Regarding page 9 first paragraph (FC vs. SC and reference 63~65).

One study reported that FC of anesthetized brain was dissimilar to structural connectivity (SC), compared to that of conscious brain (Tagliazucchi et al., 2016, "Large-scale signatures of unconsciousness are consistent with a departure from critical dynamics"). Why do you think that there are seemingly opposite results?

Although our study does not directly address the question of structure-function coupling, we note that Tagliazucchi and colleagues (2016) themselves provided a potential explanation for the apparent discrepancy between their own results to the opposite ones observed by Barttfeld et al., (2015 *PNAS*). Namely, Tagliazucchi and colleagues remarked that their effect was localised to a set of frontal and thalamic regions, in contrast with the global effect observed by Barttfeld (and now by others as well). Additionally, Barttfeld's result was found using time-varying functional connectivity, in terms of increased occupancy of more structurally-connected patterns, whereas Tagliazucchi considered the entire scan length.

For completeness, we have now added a reference to the result of Tagliazucchi, in the paragraph in question:

“(but see Tagliazucchi et al (2016) for a report of locally decreased structure-function coupling under propofol anaesthesia)”

6) To my understanding, in Fig. 1d, "self-self minus self-others" represents either or . But it is not clear it is the former or latter. Could you describe in more details about the definition of y-axis in the Fig. 1d?

The y-axis of Fig. 1d reports, for each individual, the difference between self-self correlation and self-other correlation. Self-self correlation is the correlation between the individual's FC at scan 1 and at scan 2; self-other correlation is the average correlation between the individual's FC at scan 1, and every other individual's FC at scan 2. In other words, the y-axis of Fig. 1d reports the differential identifiability measure (*Idiff*, in the terminology of Amico and Goni 2018). We now clarify this in our revised manuscript:

“differential identifiability (the difference between self-self correlation, and the mean correlation between an individual's scan at time t_x and every other individual's scan at t_y)”

7) Typo: Two "with" in the sub-title "Replication of cognitive matching with with Brain-Map" in Results section

Thank you for spotting this. We have corrected the typo.

**Response to reviewer comments for manuscript NATHUMBEHAV-24031050B,
“General anaesthesia reduces the uniqueness of brain functional connectivity
across individuals and across species”**

We thank the Reviewers for their feedback on our manuscript. Please find below our point-to-point responses to each comment. For ease of reading, reviewers' feedback is provided in **bold**; and quoted passages from the revised manuscript are shown as indented text.

Reviewer #1 (Remarks to the Author):

This manuscript examines the effects of general anesthesia agents like sevoflurane and propofol on brain connectivity. Under anesthesia, individual distinctions in human brain connectivity become less pronounced, and there is reduced involvement of cognitive function. The authors have included additional analyses that robustly support this finding, which has made the conclusions clear and compelling.

Only one aspect of the results regarding the analysis comparing the similarity or distance between human and macaque connectivity is somewhat unclear. The authors attempted to conclude that anesthesia shifts human functional connectivity closer to that of macaques. However, in figure 4b, the distance between the macaque functional connectivity and the awake human FC (brown and grey bars) appears to be greater than any distance within human FC states. This observation seems to contradict the authors' conclusion.

We appreciate the opportunity to clarify this point. Our claim that anaesthesia shifts human functional connectivity closer to that of macaques can be restated as follows: the distance between human anaesthetised FC and macaque FC, is smaller than the distance between human awake FC and macaque FC. Formally, if A is human awake FC; S is human FC under anaesthesia; and M is macaque FC; then, our claim is that $distance(S,M) < distance(A,M)$.

We can now see that the situation where $distance(S,M) < distance(A,M)$ (our claim) does not logically require that $distance(A,S)$ (distance within human FC states) must be greater than $distance(A,M)$ (distance from macaque to awake human). This is illustrated in the toy example below.

In this example, the S-M distance is smaller than the A-M distance (so this scenario satisfies our claim). However, the A-S distance is *smaller* than A-M distance. Therefore, it is evident that $distance(S,M) < distance(A,M)$ (our claim) is fully compatible with $distance(A,S) < distance(A,M)$. So there is no contradiction in claiming that the distance between the macaque

functional connectivity and the awake human FC is greater than any distance within human FC states. In other words, anaesthesia can reduce the distance between human FC and macaque FC, regardless of whether the distance within human states (A-S) is greater or smaller than the distance between species (A-M).

This analysis (distance from macaque FC in PCA space) was shown in Supplementary Figures S10 and S11 (currently Figures S11 and S12 in the revised manuscript), which provide evidence that indeed, macaque-to-anaesthetised-human distance (middle three box-plots) is smaller than macaque-to-awake-human distance (first and last box-plots), with a larger effect for larger dose of anaesthetic, which is reversed upon recovery. The previous version of Figure 4b (now Figure 4c in the revised manuscript) provides complementary evidence that increasingly deep anaesthesia brings human FC increasingly further away from awake FC, and this is also reversed upon recovery.

Ultimately, however, we believe that the evidence that anaesthesia shifts human FC closer to that of macaques is best seen from the PCA plot itself, in Figure 4a (see below): it is very clear that human awake and recovery (red and purple dots) are very close to each other; the vol 3% sevoflurane FC (green dots) is further away from them, and closer to the macaque data (diamonds); and finally, the burst-suppression anaesthesia (blue dots) is furthest from wakefulness and closest to the macaque data - to the point of partial overlap. So it is visually apparent from Figure 4a that indeed, macaque-to-human-anaesthesia distance is smaller than macaque-to-human-wakefulness distance. We now clarify this point in the *Results* section of our revised manuscript:

“Altogether, this low-dimensional representation highlights how anaesthesia shifts the functional connectivity of the human brain away from wakefulness and closer to the functional connectivity of the non-human primate brain: the distance between human anaesthetised FC and macaque FC, is smaller than the distance between human awake FC and macaque FC. This phenomenon is reversed upon post-anaesthetic recovery, whereupon human FC moves back near the original position that it occupied at baseline. These results complement our observation that anaesthetic-induced reduction in regional identifiability is most pronounced in regions of the human brain that are genetically most human-specific (Fig. 2d).”

However, we appreciate that the inclusion of the macaque data in the previous version of Figure 4b was a source of confusion, since this comparison is irrelevant to the message of 4b. Accordingly, and in response to the Reviewer’s next point about the importance of PC2 (below), we now provide an updated Figure 4 (shown below). Please see also our response to the next point, for changes in the text.

Figure 4. Anaesthesia shifts human functional connectivity away from wakefulness and closer to macaque functional connectivity | (a) Low-dimensional projection of the human and macaque functional connectivity, in the space of the first two principal axes of variation from Principal Components Analysis. Each circle represents the FC from one human, with colour reflecting condition (awake, recovery, or different doses of sevoflurane). Each diamond represents FC from one macaque, with colour representing the dataset (awake or anaesthetised). (b) Projection of the data from (a) onto PC2. (c) Distribution of Euclidean distances from awake humans' FC patterns, in the human dataset, along PC2 as shown in (b). $N = 225$ (15×15) pairs of data-points in each box-plot *** (gray), $p < 0.001$ against human recovery condition (FDR-corrected). (d) Distribution of Euclidean distances between the human data and awake macaques' FC patterns, along PC2. $N = 285$ (15×19) pairs of data-points. *** (black), $p < 0.001$ against human awake (FDR-corrected). *** (gray), $p < 0.001$ against human recovery (FDR-corrected). (e) Distribution of Euclidean distances between the human data and anaesthetised macaques' FC patterns, along PC2. $N = 135$ (15×9) pairs of data-points. *** (black), $p < 0.001$ against human awake (FDR-corrected). *** (gray), $p < 0.001$ against human recovery (FDR-corrected). Box-plots: center line, median; box limits, upper and lower quartiles; whiskers, $1.5 \times$ interquartile range. See Supplementary Data 2 for full statistical reporting. Source data are provided as a source data file.

In figure 4a, if I understand correctly, a joint PCA was applied by concatenating the human and macaque data (if so, please clarify this in the methods section). It appears that PC2 captures the differences in human states, while PC1 captures state differences in macaques. Given that PCs are orthogonal, does this imply that the distinction between awake and anesthetized states differs between humans and macaques? Could the authors elaborate on how we should interpret this species-state interaction?

The Reviewer is correct that the human and macaque data are concatenated for the PCA. This is described in the Methods section, subsection *Low-dimensional representation with Principal Components Analysis*:

“To obtain the joint PCA space, we begin by vectorising the upper triangular of each FC matrix, for each scan (awake, recovery, and different levels of anaesthesia) of each individual. We also do the same for all available scans in the awake and anaesthetised macaque datasets. The vectorised FC patterns are then concatenated, forming separate columns of a matrix M whose rows correspond to FC edges. The PCA algorithm is then applied to this matrix M, to extract linearly orthogonal principal components”

We have now clarified that the first sentence refers to the human data, specifically:

“To obtain the joint PCA space, we begin by vectorising the upper triangular of each FC matrix, for each scan (awake, recovery, and different levels of anaesthesia) of each individual **in the human dataset.**” (emphasis added)

We have also added clarification about concatenation in the Results section, where the PCA analysis is first introduced:

“We can now use Principal Components Analysis (PCA) to project all **concatenated** FC patterns across humans and macaques in a common low-dimensional space (see *Methods*).”

Pertaining to the difference between the two macaque datasets along PC1, this is difficult to interpret because the human data all come from the same individuals scanned under identical conditions except for the presence of anaesthetic. This is not the case for the macaque datasets, which do not comprise the same animals and were acquired in different sites and with different scanners and acquisition parameters. So although adding the anaesthetised macaque dataset allowed us to show that our inter-species comparisons also hold when using a different macaque dataset, we cannot draw firm conclusions about whether PC1 reflects state differences in macaques (as suggested by the Reviewer) or any other combination of these various differences between the two datasets.

For this reason, we now focus our primary analysis on PC2, because we agree with the Reviewer that this is the most relevant dimension for changes in human states (the previous analyses that used both PC1 and PC2 are now shown as Figures S10 - S12). Focusing on PC2 as shown in the revised Figure 4b, it becomes evident that both macaque datasets are approximately in the same position along this dimension. Along this dimension, anaesthetised human FC is positioned progressively further away from awake human FC, as the dose of anaesthetic increases (revised Figure 4c). At the same time, the human-to-macaque distance becomes smaller under anaesthesia than during wakefulness or recovery, as discussed above (Figure 4d-e). We have revised the Results section of our manuscript accordingly:

“We clearly see that each condition (awake, recovery, and various levels of sevoflurane anaesthesia) tends to occupy a different region of the space spanned by the first two principal components (Fig. 4a). Since PC1 appears to primarily reflect the difference between one of our macaque datasets and all other data, we focus our main analysis on PC2, which captures the differences in human states (similar results are observed when considering both PC1 and PC2; Fig. S10 and Fig. S11). It is

immediately apparent that as the dose of sevoflurane increases, the FC patterns of our human participants move progressively further away along PC2 from their initial position during pre-anaesthesia wakefulness (red circles in Fig. 4a,b)—only to then return closest to their initial awake position upon post-anaesthetic recovery of wakefulness (purple circles in Fig. 4a,b). We formally quantify this shift in terms of Euclidean distance along PC2: we find that the human condition with smallest PC2 distance from the awake human data is recovery, and deeper levels of sevoflurane anaesthesia correspond to further distance away from awake along PC2 (Fig. 4c). At the same time, we see that as the human anaesthetised FC patterns move away from wakefulness, they also reduce their distance to the location of both macaque FC datasets along PC2 (Fig. 4d,e)—with burst-suppression (the deepest level of human anaesthesia) being both furthest away from human wakeful FC (Fig. 4c), and closest to macaque FC along PC2 (Fig. 4d,e). Analogous results are observed when considering the space of both the first two principal components (Fig. S10 and Fig. S11), or when using cosine distance instead of Euclidean distance (Fig. S12); see also Fig. S13 for a representation in the space of the first three principal components instead.”

We hope that these clarifications, new analyses, and revised Figure 4 will remove any lingering doubts.

I noticed a comment regarding the ICC range. In theory, intra-class correlation is defined as the proportion of variance that is attributable to differences between classes. In other words, it is the ratio of between-subject variance to the total variance. Since both the numerator and denominator in this calculation are variances, which are inherently non-negative, the ICC should be non-negative by definition. While in rare cases in practice, the estimation of ICC can fall slightly below 0. Usually, a negative ICC generally suggests model or estimation anomalies. Therefore, I suspect there might be some misunderstanding.

We appreciate the opportunity to clarify this point. In the practical implementation of ICC, the ICC is usually estimated using the difference between sample mean squares (within and between subjects, termed MSW and MSB respectively):

$$\text{ICC} = (\text{MSB} - \text{MSW}) / (\text{MSB} + (k-1) \text{MSW})$$

The rationale is that if there is no group influence, then the variability within groups should be the same as the variability between groups. That is, $\text{MSB} = \text{MSW}$ and the intraclass correlation equals 0. If the variability within groups is less than the variability across groups, then this is evidence of some kind of convergence within the group, and ICC will be greater than 0. However, if the within-group variability MSW is greater than the variability between groups MSB, then the intraclass correlation will be negative. In other words, the intraclass correlation will be negative whenever $\text{MSB} < \text{MSW}$, i.e. the variability within groups exceeds the variability across groups. This is why we can observe negative values. Indeed, we note that negative ICC values with functional MRI data were also observed in previous work, as can be seen for example in Figure 4B, Figure 7B of Amico and Goni (*Scientific Reports*, 2018).

We have added this clarification in the *Methods* section:

“In practice, the ICC is estimated using the difference between sample mean squares:

$$ICC = (MSB - MSW) / (MSB + (k - 1) MSW)$$

with MSB being the variability of the group means from the grand mean; MSW being the variability of the individual scores from their respective group means; and k being the sample size. The rationale is that if group membership has no relevance, then the variability within groups should be the same as the variability between groups. That is, MSB = MSW and the ICC equals 0. However, if there is more variability within groups than between groups, then a negative ICC will be observed.”

Nevertheless, as a further robustness analysis we replicated the edge-wise fingerprinting analysis using only ICC values that are reliably different from zero (i.e. whose confidence interval does not include zero). Two noteworthy results emerge (see the new Figure S3 below). First, all statistically significant ICC values are positive. In other words, we do not observe any ICC values that are reliably negative. However, we see that the number of FC edges that are capable of identifying individuals in a statistically significant way is drastically diminished when comparing awake and anaesthesia. Second, the regional distribution of ICC differences induced by anaesthesia remains virtually unchanged (correlation of $r=0.88$ with the results obtained using all ICC values). These results are replicated in the propofol data as well (as shown in the new Figure S18). Together, these additional analyses clearly affirm that our results are not unduly influenced by the presence of negative ICC values.

Figure S3. Anatomical characterisation of contributions to sevoflurane-induced loss of identifiability, when only including significant ICC values | (a) Significant (p < 0.05) edge-level intra-class correlation coefficients

between awake and recovery. (b) Significant edge-level intra-class correlation coefficients between awake and sevoflurane 3%. (c) Difference in edge-level intra-class correlation coefficients between (a) and (b). (d) Regional distribution of sevoflurane-induced loss of significant ICC, projected onto the cortical surface. (e) The regional pattern of anaesthetic-induced changes in significant ICC is significantly spatially correlated with the corresponding map obtained when including all ICC values: Spearman $\rho = 0.88$, $p\text{-spin} < 0.001$, $N = 200$ regions from the Schaefer atlas.

Figure S18. Anatomical characterisation of contributions to propofol-induced loss of identifiability, when only including significant ICC values | (a) Significant ($p < 0.05$) edge-level intra-class correlation coefficients between awake and recovery. (b) Significant edge-level intra-class correlation coefficients between awake and propofol. (c) Difference in edge-level intra-class correlation coefficients between (a) and (b). (d) Regional distribution of propofol-induced loss of significant ICC, projected onto the cortical surface. (e) The regional pattern of anaesthetic-induced changes in significant ICC is significantly spatially correlated with the corresponding map obtained when including all ICC values: Spearman $\rho = 0.88$, $p\text{-spin} < 0.001$, $N = 200$ regions from the Schaefer atlas.

Reviewer #2 (Remarks to the Author):

Many thanks for the revisions or clarifications regarding all my concerns. I have no further comments.

We thank the Reviewer for this positive feedback.